# Dobi-SVD: Differentiable SVD for LLM Compression and Some New Perspectives

**Qinsi Wang**[1][*], **Jinghan Ke**[2][*], **Masayoshi Tomizuka**[2], **Kurt Keutzer**[2], **Chenfeng Xu**[2][†]
[1]Duke University,
[2]University of California, Berkeley

## Abstract

Large language models (LLMs) have sparked a new wave of AI applications; however, their substantial computational costs and memory demands pose significant challenges to democratizing access to LLMs for a broader audience. Singular Value Decomposition (SVD), a technique studied for decades, offers a hardware-independent and flexibly tunable solution for LLM compression. In this paper, we present new directions using SVD: we first theoretically and experimentally analyze the optimality of directly truncating activations, then we further identify three key issues on SVD-based LLM compression, including (1) How can we determine the optimal truncation position for each layer in LLMs? (2) How can we efficiently update the weight matrices based on truncated activations? (3) How can we address the inherent "injection" nature that results in the information loss of the SVD? We propose a new paradigm for SVD-based LLM compression, **Dobi-SVD**, to tackle the three issues. First, we propose a **differentiable** truncation mechanism, along with gradient-robust backpropagation, enabling the model to adaptively find the optimal truncation positions. Next, we utilize the Eckart-Young-Mirsky theorem to derive a theoretically **optimal** weight update formula through rigorous mathematical analysis. Lastly, by observing and leveraging the quantization-friendly nature of matrices after SVD, we reconstruct a mapping between truncation positions and memory requirements, establishing a **bijection** from truncation positions to memory. Experimental results show that with a 40% parameter-compression rate, our method achieves a perplexity of 9.07 on the Wikitext2 dataset with the compressed LLama-7B model, a 78.7% improvement over the state-of-the-art SVD for LLM compression method. We emphasize that Dobi-SVD is the first to achieve such a high-ratio LLM compression while maintaining competitive performance. We also extend our Dobi-SVD to vision-language models (VLMs) and vision-language-action models (VLAs), thereby highlighting its generalizability and practical value. We hope that the inference speedup—up to **12.4x** on 12GB NVIDIA Titan Xp GPUs and **3x** on 80GB A100 GPUs for LLMs, **1.2x** and **1.17x** on 80GB A100 GPUs for VLMs and VLAs, respectively —will bring significant benefits to the broader community such as multi-modal learning and robotics etc.

## 1 Introduction

Large language models (LLMs), such as GPT (Achiam et al., 2023), Llama (Touvron et al., 2023), and OPT (Zhang et al., 2022), have shown that scaling the size of the model and the training data can unlock impressive performance and contextual learning abilities. However, because of the growing number of parameters in LLMs and the limited memory capacity of current hardware, inference with LLMs is highly expensive. This limits their practical applications, especially for resource-constrained hardware devices and latency-sensitive programs, such as robotics, edge-device applications, and interactive entertainment. In this paper, we aim to address a key challenge: How can we perform inference with LLMs using fewer computational resources and memory, while maintaining the performance of the pre-trained models?

---

[*]Qinsi Wang and Jinghan Ke contribute equally. Order is decided by coin flip. This work was done when Jinghan Ke visited UC Berkeley.

[†]Chenfeng Xu advises the work and is the corresponding author.

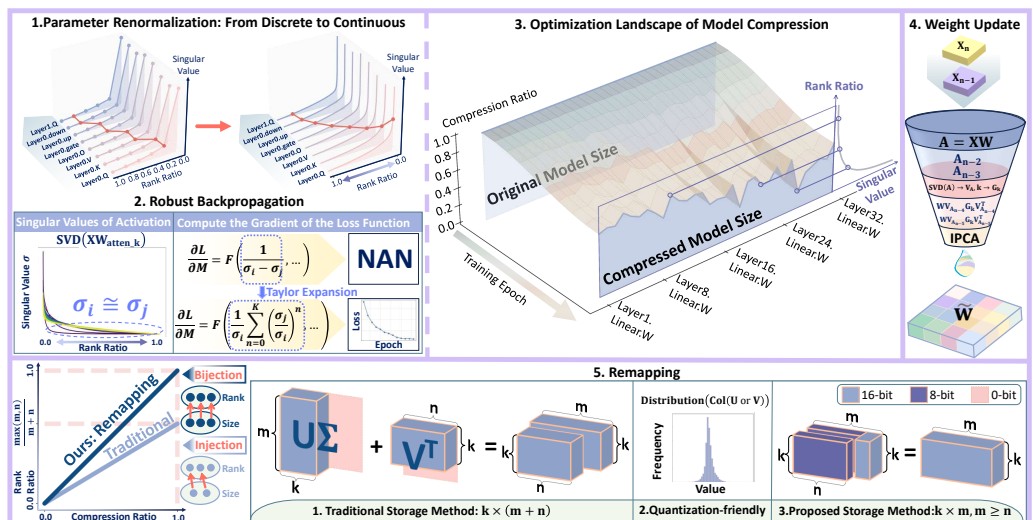

Figure 1: Overview framework of Dobi-SVD: 1-3: **Differentiable Truncation Position Training.** By applying parameter renormalization for continuous rank ratio selection and using Taylor expansion to prevent gradient explosion, our method enables robust and adaptive optimization of truncation positions. 4: **Weight Update.** Using IPCA, we sequentially extract and optimally update weight matrix features. 5: **Remapping.** We resolve a long-overlooked limitation of traditional SVD-based compression through remapping, fully unlocking SVD's potential for data compression.

Model compression (Buciluă et al., 2006; Cheng et al., 2017; Choudhary et al., 2020) has been extensively studied, with the aim of squeezing the original model into a lightweight one with a low compression ratio, *i.e.*, the ratio between the memory required by the compressed model and the original model. However, existing mainstream compression methods have their own limitations. For instance, quantization reduces the storage memory by converting floating-point calculations into lower-bit integer calculations, but it lacks flexibility and hardware generalization due to the dependency on specific hardware support (Dettmers et al., 2024; Lin et al., 2024; Frantar et al., 2022a; Kim et al., 2023). Model pruning shrinks the network by reducing redundant parameters that are less sensitive to performance, but hardware-accelerated structured pruning often leads to significant performance degradation (Ma et al., 2023; Frantar & Alistarh, 2023; Ashkboos et al., 2024; Xia et al., 2023). For example, when compressing LLaMA-7B, LLM-Pruner resulted in a 37.6% performance drop on WikiText2, even at a compression ratio of 0.8. Knowledge distillation employs a high-complexity teacher network to guide a lower-complexity student network for model compression; however, it requires retraining a new model, which incurs high time and computational costs (Bergmann et al., 2018; Hou et al., 2020; Chen et al., 2020).

In contrast to these techniques, another straightforward compression method is low-rank decomposition, which is free from the aforementioned limitations. As a technique studied many decades, singular value decomposition (SVD) (Klema & Laub, 1980) has played a significant role in fields such as image compression (Prasantha et al., 2007; Bryt & Elad, 2008), communication transmission (Lebrun et al., 2005), and signal denoising (Zhao & Jia, 2017; Guo et al., 2015). However, its potential for LLM compression has not yet been fully explored. Theoretically, SVD reduces memory and computation by truncating the singular values of a matrix, decomposing large weight matrices into the product of two smaller matrices. Models compressed with SVD can be easily deployed across a wide range of devices, and SVD offers the flexibility to compress models to any compression ratio.

However, existing SVD-based methods for LLM compression have not achieved desirable results. Traditional SVD (Lebedev et al., 2014; Moczulski et al., 2015; Sainath et al., 2013) truncated the model weights directly, often resulting in significant performance degradation and requiring extensive retraining. Recently proposed activation-aware SVD hopes that the truncated weights make the activations close to the original ones, but it still fails to deliver satisfactory performance. For example, ASVD (Yuan et al., 2023) proposes scaling the weights using a diagonal matrix to reduce the distribution error of the activations before and after truncation. SVD-LLM (Wang et al., 2024) introduces a truncation-aware data whitening strategy to retain singular values critical for activations. However, these methods exhibit severe performance degradation. At $0.4$ compression ratio, SVD-LLM experiences a 644.7% drop in model performance on Wikitext2 dataset, which is unacceptable in real-world applications. Due to the significant performance loss, SVD has not yet become a mainstream method for LLM compression compared to other compression techniques.

We aim to change this landscape by making SVD a viable and widely adopted option. Unlike previous weight-only or activation-aware methods focusing on minimizing the distance between new and original activations through weight-based truncation, we open up a novel path: directly truncating activations while enabling weight reconstruction without fine-tuning. Additionally, we are the first to fully utilize singular value information by addressing a long-overlooked limitation of traditional SVD-based compression methods. To achieve these, we analyze three associated challenges:

1. **How to determine the appropriate truncation position for each layer in an LLM?** Different weight in the model have varying sensitivities to performance degradation. By designing specific truncation points for different weight matrices (*i.e.*, the number of retained singular values), it is possible to achieve smaller performance loss at same compression ratio. However, due to the high dimensionality of the matrices in LLMs, the solution space for truncation positions is exceptionally large. Thus, efficiently finding the optimal combination of truncation is a significant challenge.

2. **How to update the weights based on truncation position?** Directly truncating the weights results in significant performance degradation. Although activation-aware SVD mitigates this issue to some extent, how to update the weights in a way that maximally preserves activation information has not yet been fully explored or theoretically proven.

3. **How to overcome the long-overlooked truncation-value limitation?** A fundamental issue with SVD-based compression is that to achieve effective compression, at least half of the singular values need to be truncated (for square matrices). This implies that even a modest compression ratio requires very low ranks, leading to substantial information loss during matrix compression, directly limiting the capacity of SVD to compress models effectively.

We propose a new paradigm, **Dobi-SVD**. An overview is shown in Fig. 1. First, through experimental analysis and theoretical investigation, we compare the effectiveness of truncating activations versus truncating weights, verifying the superiority of activation truncating. Based on this analysis, we provide answers to the three key challenges raised. First, by introducing differentiable truncation values and stable SVD backpropagation, we enable the model's performance to directly guide the matrix in adapting to find optimal truncation points. Then, leveraging the Eckart-Young-Mirsky theorem and the positive definiteness of the SVD, we derive the theoretical optimal weight update formula and employ Incremental Principal Component Analysis (IPCA) to extract features of updated weight sequentially and address memory constraints. Finally, we propose a novel SVD-based storage method that takes advantage of the value concentration property of decomposed matrices. This method establishes a bijection mapping between the truncation position and the model compression ratio, thereby overcoming the truncation limitation.

Dobi-SVD effectively addresses the three challenges of SVD in compressing LLMs. Specifically, our method achieves performance breakthroughs in three key areas. In terms of **task performance**, Dobi-SVD achieves minimal performance degradation even with high compression ratios for the first time. When compressing the LLaMA-7B model to a compression ratio of 0.4, the Dobi-SVD compressed model reaches a PPL of 9.07 on WikiText2, which represents a 78.5% improvement over the previous state-of-the-art SVD compression method, and a 9.83% improvement over the best pruning methods that require post-training and fine-tuning. This establishes SVD as an effective and highly competitive option for LLM compression; In terms of **hardware performance**, Dobi-SVD significantly reduces inference time for LLMs on low-cost GPUs. On an NVIDIA TITAN Xp 12GB GPU, the Dobi-SVD compressed LLaMA-7B model achieved a generation speed of 25.97 tokens/second, delivering a $12.4\times$ speedup compared to the original model. In particular, Dobi-SVD is hardware-agnostic, offering superior generalization across different hardware targets compared to other compression methods; In terms of **integration with other compression methods**, Dobi-SVD can be effectively combined with other compression methods such as quantization to achieve a lower compression ratio. When combined with GPTQ-4bit, Dobi-SVD compresses the LLaMA-7B model to just 3.4GB, while maintaining a perplexity of 9.97 on WikiText2, which shows that Dobi-SVD is an effective solution for model compression that can be widely applied and combined. Moreover, we extend Dobi-SVD to a broader field, vision-language models (VLMs). Specifically, we apply Dobi-SVD to compress the popular VLM, LLAVA V1.5-7B. Experiments demonstrate that our method improves throughput by 1.2 times. Notably, we find that our Dobi-SVD not only enhances efficiency but also improves VLM performance on the Pope-random dataset. We also deployed Dobi-SVD on the state-of-the-art Vision-Language-Action model, OpenVLA. Under a compression ratio of 0.4, our method achieves a 17.6% acceleration on the NVIDIA A100 while maintaining nearly

Figure 2: The differences between Dobi-SVD's method and previous approaches in handling activations and obtaining new weights. See A.4 for a detailed explanation of this figure.

lossless performance. In summary, Dobi-SVD is the first approach to make SVD-based compression methods truly competitive, highlighting the significant potential of SVD for model compression. In addition to our proposed method, we also provide some new perspectives in the Appendix; we hope to inspire the community to further advance this direction.

## 2 PRELIMINARIES

### 2.1 MATHEMATICAL EXPRESSION OF SVD FOR LLM

Given a weight matrix $W \in \mathbb{R}^{m \times n}$, it can be decomposed through SVD as $W = U\Sigma V^T$, where $U \in \mathbb{R}^{m \times m}$ and $V \in \mathbb{R}^{n \times n}$ are the right and left singular vector matrices, respectively. $\Sigma = \text{diag}(\sigma_1, \ldots, \sigma_m)$ is an $m \times n$ diagonal matrix and $\sigma_1, \ldots, \sigma_m$ are the singular values of $W$.

**The SVD compression process of** $W$ can be summarized in three steps. **Decomposition**: use SVD to decompose $W$. **Truncation**: retain the top singular values $k$ and obtain the truncated singular-value matrix $\Sigma_k = \text{diag}(\sigma_1, \ldots, \sigma_k, 0, \ldots, 0)$. **Reconstruction**: reconstruct $W$ into two matrices $W_1 = U\sqrt{\Sigma_k}$, $W_2 = \sqrt{\Sigma_k}V$, where $W_1 \in \mathbb{R}^{m \times k}$ and $W_2 \in \mathbb{R}^{k \times n}$. Through the process, $W$ can be compressed into $W_1$ and $W_2$. We define the compression ratio as $k(m+n)/(m \times n)$.

### 2.2 BASIC PROPOSITIONS.

Before the derivation of $\widetilde{W}$, we introduce three basic propositions:

**Proposition 1**: *Rank Property of Matrix Multiplication.* For the matrix multiplication $AB = C$, the rank of $C$ satisfies $rank(C) \leq \min(rank(A), rank(B))$.

**Proposition 2**: *Eckart-Young-Mirsky Theorem.* For a matrix $A \in \mathbb{R}^{m \times n}$, the best rank-$k$ approximation of $A$ in terms of the Frobenius norm or spectral norm is given by the matrix $A_k = U_A \Sigma_k V_A^T$.

**Proposition 3**: *Positive Definiteness of Matrices in SVD.* Given the SVD of a matrix $A = U_A \Sigma_A V_A^\top$, the matrices $U_A$ and $V_A$ are both orthogonal, satisfying $V_A^\top V_A = U_A^\top U_A = I$.

### 2.3 MOTIVATION: TRUNCATE WEIGHTS OR ACTIVATIONS? HOW TO DO IT OPTIMALLY?

Previous studies explored two main SVD-based compression methods: truncating weights or activations, including activation-aware approaches. The first and straightforward method, truncating weights, applies SVD to compress $W$ (2.1) Alternatively, recent works such as ASVD (Yuan et al., 2023) and SVD-LLM (Wang et al., 2024), drawing from techniques in the quantization domain, emphasize an activation-aware compression, which uses scaling matrix $S$ scales $W$ to capture the varying importance of input channels and aims to

Table 1: PPL of Llama-7b on Wikitext2 after directly truncating activations and weights under same truncation setting.

| Param Ratio | 1.0 | 0.8 | 0.6 | 0.4 |
|---|---|---|---|---|
| Activation | 5.68 | 6.36 | 8.85 | 20.71 |
| Weight | 5.68 | 20061 | 52489 | 105474 |

minimize differences in activations, *i.e.*, $\min \|A - x(\widetilde{WS})S^{-1}\|_F$, where $\widetilde{WS}$ is obtained by applying SVD to $WS$. **This raises a question: Are the existing truncation methods truly optimal? Our answer is no.** Both theoretical and experimental results reveal that **a fundamentally different third paradigm—directly truncating activations via SVD on $A$ (i.e., $xW$)—is the optimal approach.**

At the module level, Proposition 2 shows that directly truncating activations gives $A_k$, the optimal $k$-rank approximation of $A$. At the model level, we prove (A.10) that directly truncating activations better minimizes training loss than truncating weights. Previous methods avoided this due to challenges in updating weights (Figure 2), while we provide a practical and effective solution through theoretical analysis (Sect.3.2 and A.4.1). Results: **Table 1 shows truncating activations outperforms weights, and Table 2 confirms it surpasses existing activation-aware methods.**

## 3 DOBI-SVD METHOD

In this section, we address the three main challenges of SVD for LLMs to implement a high-performance SVD algorithm.

---

**Algorithm 1** Differentiable Algorithm for Finding Optimal $k$

---

**Input:** Training data $\{x_1, x_2, \ldots, x_n\}$, target compression ratio $R_{tar}$.
**Output:** Optimal $k$ values for each weight matrix $W$.
 1: **Step1: Smooth activation truncation Inference.**
 2: **for** each activation $A$ **do**
 3:     Apply SVD: $A = U_A \Sigma_A V_A^\top$, $\Sigma_A = [\sigma_1, \sigma_2, \ldots, \sigma_n]$.
 4:     Get smooth truncation: $T(\sigma_i) = \sigma_i [0.5 \cdot \tanh(\beta(k-i)) + 0.5]$, $\Sigma_k = [T(\sigma_1), \ldots, T(\sigma_n)]$
 5:     Reconstruct activations: $\tilde{A} = U_A \Sigma_k V_A^\top$.
 6: **end for**
 7: **Step2: Multi-objective loss training.**
 8: Freeze other parameters in the network and keep only $k$ trainable.
 9: **for** each training step **do**
10:     Compute current compression ratio $R_{now}$ based on current $k$.
11:     Compute multi-objective loss: $L = L_{\text{task}} + \gamma \cdot |R_{now} - R_{tar}|$.
12:     Update $k$ values by minimizing $L$.
13: **end for**

---

## 3.1 Q1: How to get the optimal truncation position?

**Solution Space of Truncation Position** To address the question, we first analyze the size of the solution space. Consider an LLM of $L$ layers, where each layer contains $M$ weight matrices $W \in \mathbb{R}^{m \times n}, m \geq n$. For each matrix, the number of possible truncation positions is $n$, $k \in \{1, 2, \ldots, n\}$. Therefore, for the entire LLM, the size of the solution space is $n^{(L*M)}$. Given that LLMs typically involve large-dimensional matrices, the solution space for truncation positions is vast.

**Differentiable Optimization of the Truncation Position.** To effectively identify the optimal truncation position within a huge solution space, we introduce, for the first time, a differentiable method to solve for the optimal truncation position. The workflow of our method, as shown in Algorithm 1, can be divided into two steps: smoothing discrete truncation positions and multi-objective loss training.

During inference, we construct an activation truncation model. To make the truncation continuous, we use the $\tanh$ function to smooth the truncation. Specifically, we define a smooth truncation function $T$ that $T(\sigma_i) = \sigma_i [0.5 \cdot \tanh(\beta(k-i)) + 0.5]$, where $k$ is a learnable truncation parameter, $\beta$ is a parameter that adjusts the smoothness of the truncation and $T(\sigma_i)$ is the truncated value of $\sigma_i$. $T(\sigma_i)$ effectively simulates the truncation, making the truncated position differentiable.

During the training, we freeze other parameters in the model and only keep $k$ trainable. In order to ensure that the optimization balances the task performance and the model compression, we use a multi-objective loss to train the model. Given the target compression ratio $R_{tar}$, our loss function is $L = L_{\text{task}} + \gamma \cdot |R_{now} - R_{tar}|$, where $L_{\text{task}}$ is the loss of the model on the downstream task, $R_{now}$ is the model compression ratio calculated by the current $k$ value.

Note that our algorithm requires very low computational cost since the optimization is only conducted on $k$ of different layers. For example, in the Llama-7b model, there are only 224 trainable parameters, and training requires just **8** GPU hours.

**Stabilize SVD Backpropagation.** Although a differentiable algorithm is theoretically feasible, we emphasize that the gradient is the devil (Wikipedia, 2024), especially in backpropagation involving low-rank matrices, where gradients are prone to exploding due to numerical instability. Consider a matrix $A \in \mathbb{R}^{m \times m}$ obtained via SVD: $A = U_A \Sigma_A V_A^\top$. During backpropagation, the gradients of the loss with respect to $U_A$, $\Sigma_A$, and $V_A$ are denoted as $g_U$, $g_\Sigma$, and $g_V$, respectively. The gradient of $A$ is then given by:

$$gA = U \left( \frac{\text{skew}(U^T g_U)}{\mathbf{E}} \Sigma + \Sigma \frac{\text{skew}(V^T g_V)}{\mathbf{E}} + \text{diag}(g_\Sigma) \right) V^T, E_{ij} = \begin{cases} \sigma_j^2 - \sigma_i^2 & \text{if } i \neq j, \\ 1 & \text{if } i = j, \end{cases} \quad (1)$$

where $\text{skew}(x) = (x - x^\top)/2$ extracts the skew-symmetric part of a square matrix, and $E_{ij}$ is $\mathbf{E}$'s $(i, j)$-th element, and $\sigma_i$ and $\sigma_j$ being the $i$-th and $j$-th singular values of $A$. As shown in Eq.1, when $\sigma_i$ and $\sigma_j$ are very small or close, $E_{ij}$ approaches 0, causing $\mathbf{E}$ to vanish and $g_A$ to diverge to infinity. This gradient explosion problem frequently arises during optimization, especially

---

**Algorithm 2** Computing the Theoretical Optimal Rank-$k$ Weight Matrix $\widetilde{W}$ via IPCA

---

**Input:** Training data $\{x_1, x_2, \ldots, x_n\}$, original weight matrix $W$, required rank $k$.

**Output:** Theoretical optimal rank-$k$ weight matrix $\widetilde{W}$.

1: Input the training data into the network.
2: Obtain activations $[A_1, A_2, \ldots, A_n]$ and their right-singular vectors $[V_{A1}, V_{A2}, \ldots, V_{An}]$.
3: Generate $G_k$, a diagonal matrix with the first $k$ elements 1 and the rest 0.
4: Initialize mean $\mu = 0$, $\mathbf{V}_{\text{old}} = V_1$.
5: **for** $i = 2$ to $n$ **do**
6:     Center the data: $V_i = V_i - (\mu + \frac{1}{i}(V_i - \mu))$.
7:     Update $\mathbf{V}_{\text{new}}$: $\mathbf{V}_{\text{new}} = [\mathbf{V}_{\text{old}}, V_i]$.
8:     Update principal direction $V'$: SVD$(\mathbf{V}_{\text{new}}) \to U', \Sigma, V'^T$
9:     Update $\mathbf{V}_{\text{old}}$: $\mathbf{V}_{\text{old}} = V'[: k]$
10: **end for**
11: Use $\mathbf{V}_{\text{old}}$ as the final $\mathbf{V}$, update weight $\widetilde{W} = W\mathbf{V}G_k\mathbf{V}^T$.

---

since activation matrices in LLMs often exhibit approximately low-rank structures. To address the above problem, we propose an approximate solution for $E_{ij}$ in two cases:

1. When $\sigma_i \approx 0$ and $\sigma_j \approx 0$, we directly set $1/E_{ij} = \gamma$, where $\gamma$ is a small constant that ensures the contribution of $\sigma_i$ and $\sigma_j$ to $g_A$ remains small.

2. When $\sigma_i \not\approx 0$ and $\sigma_j \not\approx 0$, but $\sigma_i \approx \sigma_j$, we perform a Taylor expansion of $1/E_{ij}$ as follows:

$$\frac{1}{E_{ij}} = \frac{1}{\sigma_i(\sigma_i + \sigma_j)} * \frac{1}{1 - (\sigma_j/\sigma_i)} \approx \frac{1}{\sigma_i(\sigma_i + \sigma_j)}\left(1 + \left(\frac{\sigma_j}{\sigma_i}\right) + \cdots + \left(\frac{\sigma_j}{\sigma_i}\right)^K\right) \quad (2)$$

To accelerate the computation, we utilize the summation formula of a geometric series.

**Answer 1: By stabilizing SVD backpropagation for differentiable optimization of truncation position, we can efficiently find the optimal position of each layer.** To the best of our knowledge, this is the first work to enable end-to-end optimization for SVD-based LLM compression.

### 3.2   Q2: HOW TO UPDATE WEIGHTS OPTIMALLY?

In Sect. 3.1, we obtained effective activation truncation position. In this section, we address the second challenge: How to optimally update weights based on the truncation position?

Leveraging Proposition 3, applying SVD to directly truncate activations at $k$ can be formulated as:

$$A_k = U_A \Sigma_A G_k V_A^\top = U_A \Sigma_A I G_k V_A^\top = U_A \Sigma_A V_A^\top V_A G_k V_A^\top = A V_A G_k V_A^\top, \quad (3)$$

where $G_k$ is a diagonal matrix with the first $k$ elements as 1 and the rest as 0. Substituting $A = xW$,

$$A_k = A V_A G_k V_A^\top = xW V_A G_k V_A^\top. \quad (4)$$

At truncation position $k$, for a set of inputs $\{x_i\}_{i=1}^n$, the ideal updated weight $\widetilde{W}$ should satisfy:

$$\min \sum_{i=1}^n \|x_i W V_{A_i} G_k V_{A_i}^T - x_i \widetilde{W}\|_F = \min \sum_{i=1}^n x_i \|W V_{A_i} G_k V_{A_i}^T - \widetilde{W}\|_F, \quad \text{rank}(\widetilde{W}) = k \quad (5)$$

Thus, our goal is to find the rank-$k$ matrix $\widetilde{W}$ closest to the set of projected weight matrices $\mathbf{W^P} = \{W V_{A_i} G_k V_{A_i}^\top\}_{i=1}^n$. Since $x$ is usually not a full-rank square matrix, computing $\widetilde{W}$ via its inverse is infeasible. Therefore, we derive $\widetilde{W}$ from the underlying structure of $\mathbf{W^P}$.

While PCA can compute $\widetilde{W}$, the large size of $\mathbf{W^P}$ leads to excessive memory demands. Traditional PCA scales exponentially with matrix size, making large $n$ impractical and requiring hundreds of Gigabytes. To address this, we make a novel use of the Incremental Principal Components Analysis (IPCA) algorithm for memory-efficient PCA. The algorithm, detailed in Algorithm 2, centralizes each new input $W_i^p$ from $\mathbf{W^P}$ and updates the principal components via incremental SVD. By processing the matrices sequentially, the need to input all $n$ matrices at once is replaced with multiple steps, each handling only two matrices at a time, significantly reducing memory usage.

**Answer 2: By using IPCA, we can calculate the theoretically optimal Rank-$k$ Weight Matrix $\widetilde{W}$.**

Table 2: Dobi-SVD vs. SOTA methods in terms of compression performance of LLaMA-7b on three language modeling datasets (in-domain evaluation) and seven common sense reasoning datasets (zero-shot evaluation). The best performance is marked in bold. Drop means relative performance drop to baseline. Dobi-SVD* refers to the result obtained without remapping. The performance of the ASVD and SVD-LLM is derived from the results reported in SVD-LLM. † uses LoRA fine-tuning.

| Ratio | Method | PPL ($\downarrow$) | | | Accuracy ($\uparrow$) | | | | | | | Avg. ($\uparrow$) | Drop ($\downarrow$) |
|---|---|---|---|---|---|---|---|---|---|---|---|---|---|
| | | Wiki2 | PTB | C4 | Openb. | ARC_e | ARC_c | WinoG. | HellaS. | PIQA | MathQA | | |
| 1.0 | Baseline | 5.68 | 8.35 | 7.34 | 0.28 | 0.67 | 0.38 | 0.67 | 0.56 | 0.78 | 0.27 | 0.52 | 0% |
| 0.8 | ASVD | 11.14 | 16.55 | 15.93 | 0.25 | 0.53 | 0.27 | 0.64 | 0.41 | 0.68 | 0.24 | 0.43 | 17.3% |
| | SVD-LLM† | 7.94 | 16.22 | 15.84 | 0.22 | 0.58 | 0.29 | 0.63 | 0.43 | 0.69 | 0.24 | 0.44 | 15.4% |
| | **Dobi-SVD\*** | 8.54 | **14.83** | 10.01 | 0.26 | 0.59 | 0.31 | 0.66 | 0.44 | 0.70 | 0.23 | 0.46 | 11.5% |
| | **Dobi-SVD** | **6.08** | 15.39 | **7.83** | **0.27** | **0.65** | **0.37** | **0.68** | **0.54** | **0.77** | **0.27** | **0.50** | **3.84%** |
| 0.6 | ASVD | 1407 | 3292 | 1109 | 0.13 | 0.28 | 0.22 | 0.48 | 0.26 | 0.55 | 0.19 | 0.30 | 42.3% |
| | SVD-LLM† | 13.11 | 63.75 | 49.83 | 0.19 | 0.42 | 0.25 | 0.58 | 0.33 | 0.60 | 0.21 | 0.37 | 28.8% |
| | **Dobi-SVD\*** | 13.54 | 46.38 | 23.54 | 0.22 | 0.41 | 0.27 | 0.58 | 0.34 | 0.61 | 0.23 | 0.38 | 26.9% |
| | **Dobi-SVD** | **8.12** | **43.85** | **12.63** | **0.28** | **0.65** | **0.32** | **0.62** | **0.45** | **0.72** | **0.25** | **0.47** | **9.61%** |
| 0.4 | ASVD | 57057 | 45218 | 43036 | 0.12 | 0.26 | 0.21 | 0.49 | 0.26 | 0.53 | 0.18 | 0.29 | 44.2% |
| | SVD-LLM† | 53.74 | 438.58 | 345.49 | 0.14 | 0.28 | 0.22 | 0.50 | 0.27 | 0.55 | 0.21 | 0.31 | 40.3% |
| | **Dobi-SVD\*** | 46.18 | 238.91 | 190.62 | 0.15 | 0.31 | 0.20 | 0.52 | 0.28 | 0.54 | 0.22 | 0.32 | 38.4% |
| | **Dobi-SVD** | **9.95** | **67.62** | **17.94** | **0.23** | **0.52** | **0.24** | **0.56** | **0.38** | **0.65** | **0.23** | **0.40** | **23.1%** |

### 3.3 HOW TO OVERCOME THE LONG-OVERLOOKED TRUNCATION LIMITATION?

In this section, we address a long-overlooked limitation of SVD. As outlined in Section 2, for an $m \times n$ matrix $W$ with truncation position $k$, the compression ratio is $r = k(m+n)/(m \cdot n)$. Setting $r = 1$ gives $k = (m \cdot n)/(m+n) < \min(m,n) = \text{rank}(W)$, meaning the model size stays the same, but information is lost. Notably, when $m = n$, $k = \text{rank}(W)/2$, truncating half the singular values. This demonstrates that in traditional SVD-based compression, the compression ratio is injective with respect to truncation position, often leading to significant performance degradation.

**To address this limitation, we propose remapping the relationship between compression ratios and truncation positions to establish a bijection**, where for $r \in [0,1]$, $k$ ranges from 0 to $\text{rank}(W)$ with a one-to-one correspondence. This gives the compression ratio $r = k/\text{rank}(W) = k/\min(m,n) = k \cdot \max(m,n)/(m \cdot n)$, with the compressed matrix size being $k \cdot \max(m,n)$.

To achieve this bijection, we propose a new SVD compression method: Step 1. For the updated matrix $\widetilde{W}$ with rank $k$, perform SVD to obtain $U_{\widetilde{W}}, \Sigma_{\widetilde{W}}, V_{\widetilde{W}}^T$. Extract the first $k$ columns of $U_{\widetilde{W}}\Sigma_{\widetilde{W}}$ as an $m \times k$ matrix $U_{\widetilde{W},k}\Sigma_{\widetilde{W},k}$, and the first $k$ rows of $V_{\widetilde{W}}^T$ as a $k \times n$ matrix $V_{\widetilde{W},k}^T$. Step 2. Assuming $m \geq n$, the storage space becomes $m \times k$. Take the first $n$ rows of $U_{\widetilde{W},k}\Sigma_{\widetilde{W},k}$ and $V_{\widetilde{W},k}^T$ (both $n \times k$), halve their bit precision, concatenate them, and replace the first $n$ rows of $U_{\widetilde{W},k}\Sigma_{\widetilde{W},k}$. Finally, store only the modified $U_{\widetilde{W},k}\Sigma_{\widetilde{W},k}$, using $m \times k$ space.

The compression method in Step 2 leverages the orthogonality of the left and right singular vector matrices $U$ and $V$, whose column vectors follow a normal distribution (as shown in the 'remapping' section of Fig. 1), making them ideal for uniform quantization methods like QLoRA. Appendix A.7 shows that quantization introduces minimal error.

**Answer3: By applying our quantized storage method, the compression ratios and truncation position form a bijective mapping, overcoming the truncation position limitation.**

### 3.4 CONCLUSION: TWO NEW PERSPECTIVES OF DOBI-SVD

By leveraging the fundamental theorem of SVD, we derive the optimal approach for SVD-based compression: directly truncating activations (Section 2.3). To determine the optimal truncation point of the activation matrix, we develop a robust and efficient differentiable SVD algorithm applicable to general matrices (Sections 3.1 and A.6). Subsequently, we propose a theoretical framework to derive a new weight matrix from the truncated activations (A.4.1 and Section 3.2), successfully implementing this using Incremental PCA (IPCA). Collectively, these contributions form **Dobi-SVD's New Perspective 1: A Novel Path from Activation to Weight** (A.4).

Furthermore, we highlight a long-overlooked limitation of traditional SVD-based compression methods and, for the first time, propose an effective solution. This leads to **Dobi-SVD's New Perspective 2: Fully Unlocking SVD's Potential for Data Compression by Addressing A Long-Overlooked Limitation** (Sections 3.3 and A.5).

Table 3: Dobi-SVD vs. popular pruning methods in terms of compression performance of LLaMA-7b on seven common sense reasoning datasets. The best performance is marked in bold. The performance of the pruning methods are derived from their original paper.

| Ratio | Method | Cost | | Accuracy (↑) | | | | | | Avg. (↑) | Drop (↓) |
|---|---|---|---|---|---|---|---|---|---|---|---|
| | | Post-train | Fine-tuning | BoolQ | PIQA | WinoG. | ARC_e | ARC_c | OBQA | | |
| 1.0 | Baseline | - | - | 0.73 | 0.78 | 0.67 | 0.67 | 0.41 | 0.42 | 0.61 | 0% |
| 0.8 | LLM-Pruner | ✓ | ✗ | 0.59 | 0.75 | 0.61 | 0.59 | 0.37 | 0.39 | 0.55 | 9.83% |
| | LLM-Pruner(w/LoRA) | ✓ | ✓ | 0.69 | 0.76 | 0.65 | 0.63 | 0.37 | 0.40 | 0.58 | 4.92% |
| | FLAP | ✓ | ✗ | 0.69 | 0.76 | 0.68 | **0.69** | **0.39** | 0.39 | 0.60 | 1.64% |
| | **Dobi-SVD** | ✓ | ✗ | **0.73** | **0.77** | **0.68** | 0.65 | 0.37 | **0.42** | **0.61** | **0%** |

## 4 EXPERIMENTS

Without losing generalizability, our experiments in the main text are conducted on LLaMA-7B. We investigate four settings: (1) Evaluation on three language modeling datasets and seven commonsense reasoning datasets, comparing against state-of-the-art SVD compression and popular pruning methods (Sect. 4.1). (2) Analysis of the importance and roles of each component in Dobi-SVD (Sect. 4.2). (3) Testing the acceleration of Dobi-SVD on different hardware and combining it with quantization (Sect. 4.3). (4) Test the performance of Dobi-SVD on VLMs and vision-action model (Sect. **??**). **Details of experimental settings are provided in** A.3.

Additional experiments in the Appendix include evaluations on more tasks (e.g., MMLU, popular pruning methods), more models (LLaMA2-7B: Tabs. 12, 14; and LLaMA3.1-8B: Tabs.11 ; Llama-13B: Tabs. 16, 18; Llama2-13B: Tabs. 17, 19). We also explore combining with quantization (Tab. 20) and direct comparisons with quantization (Tab. 21). Finally, we test the performance of Dobi-SVD on VLMs (Tab. 22) and vision-action model (Tab. 24). And we compare the performance of large models compressed by Dobi-SVD with small models that are not compressed. (Tabs. 25, 26).

### 4.1 MAIN RESULTS

**In-domain Evaluation.** We evaluate the performance of Dobi-SVD on top of LLaMA-7B model, with compression ratio ranging from 40% to 80%. The experimental results are presented in Table 2. Notably, Dobi-SVD demonstrates significantly better performance than other SVD methods. At a 0.4 compression ratio, the Dobi-SVD achieves a PPL of 9.70 on Wikitext-2, compared to 43,104 and 458 for ASVD and SVD-LLM, respectively. This indicates that even with only 40% of the parameters retained, Dobi-SVD maintains an acceptable performance degradation, a level of performance unattainable by other SVD methods. We emphasize that even without the proposed remapping strategy, Dobi-SVD outperforms prior-arts by a large margin, especially under the low parameter-ratio. This demonstrates the effectiveness of the proposed differentiable optimization of truncation position. Additionally, our quantized storage remapping strategy further improves the performance, showing the significance of our method for improving the injective nature in SVD.

**Zero-shot Evaluation.** To demonstrate the task generalization capability of Dobi-SVD, we take the model trained on Wikitext-2 dataset and conduct the validation on seven out-of-domain datasets. As shown in Table 2, Dobi-SVD consistently outperforms the previous state-of-the-art methods across different datasets and compression ratios. Specifically, at 80% compression ratio, Dobi-SVD shows an average performance drop by only 3.14%. At low compression ratios, Dobi-SVD can still maintain good performance. For example, at compression ratio of 0.4, Dobi-SVD still has an average accuracy of 40%, while ASVD and SVD-LLM only have 29% and 31% respectively. Note that even without remapping, the performance of Dobi-SVD* is still better than other SVD compression algorithms. This shows that our differentiable optimization for truncation positions and weight update method are more effective than other SVD-based methods.

**Performance comparison with pruning methods.** To further demonstrate Dobi-SVD's superior trade-off between compression ratio and memory usage, we also compare it with state-of-the-art pruning methods, as shown in Table 3. Note that pruning method is orthogonal to the SVD-based method and these two kinds of methods are in different tracks. The results show that Dobi-SVD achieves comparable performance relative to the pruning-based method across various task sets. Compared with FLAP (An et al., 2024a), a state-of-the-art pruning method, Dobi-SVD outperforms it on all four tasks. We highlight this is the first time that SVD compression method achieves better performance than pruning. Notably, while both Dobi-SVD and pruning methods require post-training,

Table 4: Comparison of performance before & after remapping.

| Ratio | Model | Wiki | C4 | PTB |
|---|---|---|---|---|
| 80% | Remap(16bit) | 6.05 | 7.79 | 22.27 |
| | **Remap(8+16bit)** | **6.08** | **7.83** | **22.39** |
| | W/o Remap | 8.87 | 10.91 | 25.03 |
| 60% | Remap(16bit) | 8.07 | 12.54 | 43.68 |
| | **Remap(8+16bit)** | **8.12** | **12.63** | **43.85** |
| | W/o Remap | 14.96 | 24.60 | 47.01 |
| 40% | Remap(16bit) | 9.78 | 17.39 | 67.81 |
| | **Remap(8+16bit)** | **9.95** | **17.94** | **67.62** |
| | W/o Remap | 58.02 | 145.41 | 270.16 |

Table 5: Performance and memory usage of Dobi-SVD combined with 4-bit quantization on wikitext2.

| Ratio | Method | PPL | Memory |
|---|---|---|---|
| 0.4 | Dobi-SVD | 9.95 | 6.8GB |
| | **Dobi-SVD+GPTQ** | **12.04** | **1.8GB** |
| 0.6 | Dobi-SVD | 8.12 | 7.7GB |
| | **Dobi-SVD+GPTQ** | **9.97** | **2.4GB** |
| 0.8 | Dobi-SVD | 6.08 | 10.1GB |
| | **Dobi-SVD+GPTQ** | **7.01** | **3.1GB** |

Table 6: Speed of running unmapped Llama-7b on TITAN Xp 12GB GPU. PPL is tested on the wikitext2.

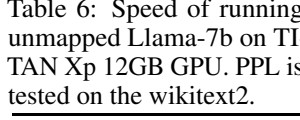

| Ratio | Mem (GB) | Speed (tokens/s) | Speed Up |
|---|---|---|---|
| 1.0 | 12.6 | 2.09 | 1.0× |
| 0.8 | 10.1 | **23.32** | **11.2×** |
| 0.6 | 7.7 | **24.80** | **11.8×** |
| 0.4 | 6.8 | **25.97** | **12.4×** |

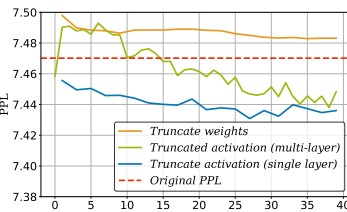
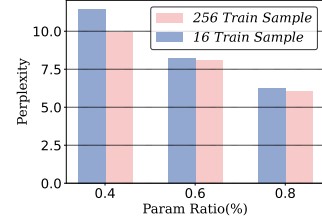
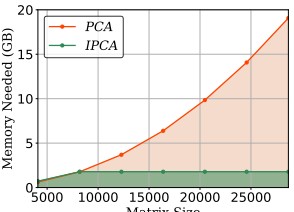

Figure 3: (Left) Performance Comparison of different training methods on LLaMA-7b. For activation truncation (multi-layer) we only truncate layers 29-31, and for activation truncation (single-layer) we only truncate the 29-th layer. (Middle) Comparison of model performance using batch size = 256 and 16 for training. (Right) Comparison of memory requirements for PCA and IPCA for $n * n$ matrix.

Dobi-SVD only trains the truncation positions of the matrices. Dobi-SVD significantly reduces the number of trained parameters and computational cost compared to pruning. Specifically, LLM-Pruner requires training additional 1.2 billion parameters to maintain the model performance on LLaMA-7B, whereas Dobi-SVD only trains 224 parameters.

## 4.2 ANALYSIS EXPERIMENT

In this section, we perform analysis experiments on three critical components of the Dobi-SVD method: differentiable optimization of truncation position, efficient weight-updates with Incremental PCA (IPCA) and quantized storage remapping.

### 4.2.1 ANALYSIS ON DIFFERENTIABLE OPTIMIZATION OF TRUNCATION POSITION

**Guided Truncation.** Based on the above analysis, we hypothesize that, under the same compression ratio, truncating the Attention layers in later stages of LLM may result in smaller performance losses. To validate this, we conducted two experiments on the LLaMA-7B model: truncating a single layer (layer 29) and truncating multiple layers (layers 29-31) with differentiable training. The results are shown in Fig. 3 (a). We observe that both truncation settings lead to better performance compared to the original model, while barely weight truncation results in the performance degeneration. Besides, truncating single layer can even lead to better performance compared to truncation multiple layers. Both experimental observations indicate that the proposed differentiable optimization of truncation position offers promising potentials to improve model performance through activation truncation and provides guidance for selecting layers to conduct low-rank decomposition.

**Efficient Training.** Besides, we validate the sample-efficient training to highlight the advantage of the differentiable optimization. A common post-training batch size is 256 and here we test the model's performance with training batch size as 16 while keeping the epoch same. The performance is shown in Fig. 3 (b). Even with such a few batch size, Dobi-SVD achieves results comparable to those obtained with 256 samples. This demonstrates the efficiency of our differentiable optimization. Note that training with batch size as 16 on LLaMA-7B require only a few GPU-hours.

### 4.2.2 ANALYSIS ON EFFICIENT WEIGHT UPDATE.

We compare the memory footprint for our proposed efficient weight update strategy (w/ IPCA) and original PCA method, as shown in Fig. 3. We observe that as the dimensionality of the decomposed matrix increases, the memory footprint of PCA grows exponentially. In contrast, our weight-update strategy with IPCA significantly reduces memory usage, which remains close to a constant as the dimensionality increases. This is because PCA must be used for decomposing the whole matrix $V = [V_1, V_2, V_3, \dots, V_n]$, while IPCA can be used for conducting the decomposition at each step $V = \text{cat}[V_{\text{old}}, V_{ii}]$ thus gets rid of the requirement of storing the entire matrix.

Figure 4: Tokens/sec of original LLaMA-7B and its compressed version by Dobi-SVD under 40%, 60% and 80% compression ratio on single A100 GPU. (a): comparison with different batch size while sequence length = 32. (b): comparison with different sequence length while batch size = 64.

### 4.2.3 ANALYSIS ON QUANTIZED STORAGE FOR REMAPPING.

To demonstrate the effectiveness of our proposed quantizes storage remapping, we compare the performance under three different approaches: remapping without quantization (*i.e.*, Remap (16bit) in Table 4), remapping with quantization to maintain the compression ratio (*i.e.*, Remap (8+16bit) in Table 4), and no remapping while maintaining the compression ratio (*i.e.*, W/o Remap in Table 4), as shown in Table 4. By comparing the first two approaches, we can see that quantization results in minimal performance drop regardless. The performance comparison between remapping with quantization and no remapping reveals that remapping significantly improves model performance, especially at lower compression ratios. Specifically, at a compression ratio of 0.4, the remapped model achieves a perplexity of 9.95 on the WikiText2 dataset, while the non-remapped model reaches 58.02. This highlights the effectiveness of our proposed remapping in enhancing the model performance.

### 4.3 INFERENCE-EFFICIENCY EVALUATION

**Performance on high-performance GPUs.** Dobi-SVD not only compresses LLMs but also improves the efficiency of inference on real hardware. We tested the original LLaMA-7B and LLAMA-7B with our Dobi-SVD on NVIDIA A100 GPU by measuring the number of tokens generated per second under varying batch sizes and sequence lengths. The results are shown in Fig. 4. Across all compression ratios, Dobi-SVD consistently improved generation speed. As the batch size increases and the sequence length decreases, this speed improvement becomes more pronounced. With a compression ratio of 0.4, we achieve an improvement of inference speed by up to 1.75 ×. At the 0.6 compression ratio, the speedup is up to 1.4 ×. These results highlight the efficiency improvement of Dobi-SVD and suggest its practical potential in LLM services on cloud.

**Performance on low-performance GPUs.** To demonstrate the applicability of Dobi-SVD in resource-constrained environments, we deploy both the original and compressed LLaMA-7B models on an NVIDIA TITAN Xp with 12GB of memory. The results are shown in Table 6. Since the LLaMA-7B model requires approximately 14.8GB of memory, the original model requires data transfer between the CPU and GPU during inference. However, applying Dobi-SVD is able to make the entire model run on the GPU. Dobi-SVD achieves the speedup of 12.4 ×. This highlights the huge potential of Dobi-SVD for practical applications in resource-limited devices, like edge-devices.

**Performance combined with quantization.** To verify the compatibility of Dobi-SVD with quantization, we use GPTQ-4bit combined with Dobi-SVD to compress LLaMA-7B. We measure the memory footprint and PPL on wikitext2 before and after quantization. Table 5 shows that the combining GPTQ-4bit with Dobi-SVD further improve the memory utility. It is worth noting that our results significantly outperform previous methods combining SVD with quantization. For instance, when compressing the model to a compression ratio of 0.6 and integrating with GPTQ-4bit, Dobi-SVD achieves a PPL of 9.97. This illustrates the wide applicability of Dobi-SVD, which can be flexibly combined with other quantization methods.

## 5 CONCLUSION

We introduce Dobi-SVD, an efficient SVD-based method for LLM compression. We address three key challenges: (a) how to determine the truncation position, (b) how to update weights efficiently and (c) how to overcome truncation limitation that results in information loss. We first theoretically and empirically explain why truncating activations is better than truncating weights. In addition, we propose solutions to address these three challenges using a differentiable optimization strategy to find the truncation position, an efficient weight update method via the Eckart-Young-Mirsky Theorem, and a quantized memory remapping to maximize SVD's potential. Experiments demonstrate that Dobi-SVD achieves minimal performance loss at low compression ratios. Dobi-SVD compresses LLaMA-7B to a 0.4 compression ratio with a PPL of 9.95 on WikiText2, outperforming advanced SVD-based and pruning methods. It also provides a 12.4× speedup on NVIDIA Titan Xp 12GB GPU with negligible loss. Overall, Dobi-SVD is a hardware-agnostic, highly adaptable compression method that demonstrates the significant potential and competitiveness of SVD in model compression.

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

# A APPENDIX

**Organization** In this appendix, we provide in-depth descriptions of the materials that are not covered in the main paper, and report additional experimental results. The document is organized as follows:

- **A.1**- Related Work
- **A.2**- Limitations and potential solutions
- **A.3**- Experimental setting details
- **A.4**- Dobi-SVD's New Perspective 1: A Novel Path from Activation to Weight
  - **A.4.1** Theoretical Support for Updating Weights Using IPCA
- **A.5**- Dobi-SVD's New Perspective 2: Fully Unlocking SVD's Potential for Data Compression by Addressing A Long-Overlooked Limitation
- **A.6**- Dobi-SVD's Robust and Efficient Differentiable SVD Algorithm for General Matrices
- **A.7**- Additional analytical results
  - **A.7.1** Quantization Precision Loss
  - **A.7.2** Effectiveness of differentiable training
  - **A.7.3** Differentiable $k$ changes at various compression ratios
  - **A.7.4** Truncation sensitivity analysis
- **A.8**- Additional experimental results
  - **A.8.1** Experimental results on more models
  - **A.8.2** Combined with quantization
  - **A.8.3** Experimental results on VLM
  - **A.8.4** Experimental results on OpenVLA
  - **A.8.5** Comparison with smaller uncompressed models
- **A.9**- Example Demonstration of Real Sentence Generation
- **A.10**- Analysis of directly truncating activations over weights

## A.1 RELATED WORK

**LLM Model Compression.** Large language models (LLMs) typically contain billions of parameters, making inference on resource-constrained hardware challenging. To address this, researchers have developed various methods to compress models without requiring retraining. These methods can be categorized into three main categories: pruning, quantization, and low-rank decomposition. Specifically, pruning sets individual weights or structured components to zero without changing the overall structure of the LLM. For example, SparseGPT (Frantar & Alistarh, 2023) prunes the least important weight elements by inverting the Hessian matrix. However, the irregular sparsity from unstructured pruning often fails to achieve significant speedup, only performing optimally on specific hardware architectures. LLM-Pruner (Ma et al., 2023), on the other hand, leverages a small dataset to estimate the coupled importance of weights, parameters, and groups, then applies LoRA-based pruning to recover accuracy. Yet, this approach significantly degrades model accuracy, especially at low compression ratios, due to the extensive modification of the weight matrices. Sheared Llama (Xia et al., 2023) performs extensive training on 50 billion tokens after compression. Quantization, another approach, compresses the model by reducing the precision of the LLM's weight matrices. For instance, GPTQ (Frantar et al., 2022b) employs layer-wise quantization, updating weights using inverted Hessian information. The drawback of quantization is that it offers limited compression options, usually between 3 and 8 bits, which may not fully optimize memory utilization.

**SVD-based Model Compression.** SVD-based LLMs compression has been widely explored (Jaderberg et al., 2014; Zhang et al., 2015; Denton et al., 2014). Earlier studies primarily used SVD to compress embedding layers (Chen et al., 2018; Acharya et al., 2019). As model sizes have grown, research has shifted towards applying SVD to weight matrices (Hsu et al., 2022; Wang et al., 2021a). Recent findings (Sharma et al., 2023) suggest that LLM weights are often approximated as low-rank matrices, highlighting SVD's potential for LLM compression. Traditional SVD focuses on compressing the original weight matrix by minimizing $|W - W'|$. However, since it does not account

for parameter importance, it often results in significant performance degradation. These methods typically require fine-tuning to recover performance, which demands substantial computational resources for LLMs. To mitigate this, recent works have focused on activation-aware SVD, which aims to minimize $|A - A'|$. For example, ASVD posits that the activation distribution influences compression error and scales $\Sigma W$ with a diagonal matrix $S$, where $S$ represents the input channel's impact on weights. SVD-LLM argues that not all larger singular values are necessarily more important, introducing a truncation-aware whitening strategy to determine which singular values are critical for activations. However, current activation-aware SVD methods are limited to modifying $\Sigma W$ to adjust $W$, which restricts the values that $\hat{W}$ can take, failing to effectively retain activation information. For instance, ASVD is only effective at high compression ratios (0.8 and 0.9), suffering from significant performance loss at lower compression ratios. SVD-LLM, when compressed to a 0.4 ratio, causes PPL to drop from the original 7.94 to 42.3.

In addition, low-rank decomposition has also been applied in different forms for LLM compression methods. For instance, LQER (Zhang et al., 2024) uses SVD to address quantization errors in the quantization process, while LRQ (Lee et al., 2024) applies SVD to enhance the sample handling capacity during training. CALDERA (Saha et al., 2024), based on matrix compression method LPLR (Saha et al., 2023), adopts non-SVD low-rank approaches for weight compression, but these typically lead to increased dimensions of the new weight matrices, resulting in performance degradation and necessitating the combination of quantization and LoRA fine-tuning.

## A.2 LIMITATIONS AND POTENTIAL SOLUTIONS

We analyze three limitations of Dobi-SVD and suggest potential solutions. Firstly, our current SVD implementation is time-consuming and memory-intensive. This issue stems from Python's support for only fp32 SVD, which could be alleviated by implementing low-precision SVD. Secondly, during memory remapping, transitioning precision becomes challenging when further quantizing to 4-bit or below, leading to a performance drop at 2-bit. To our knowledge, 2-bit quantization itself poses significant challenges in LLM compression. Third, quantization introduces additional dequantization time during inference, which could be reduced by quantizing larger matrices or using more advanced quantization libraries. Our future work will focus on overcoming these limitations and exploring broader applications of Dobi-SVD, such as in vision-language models and robotics.

## A.3 EXPERIMENTAL SETTING DETAILS

In this section, we provide a detailed description of our experimental setup and hyperparameter configurations.

**Models, Datasets and Metric:** To demonstrate the task generalization of Dobi-SVD, we use Llama-7b (Touvron et al., 2023) and evaluate the performance on different tasks. We first test the model's in-domain performance on the C4 (Sakaguchi et al., 2021a), Wikitext2 (Merity et al., 2016), and PTB (Marcus et al., 1993), respectively. On these three datasets, we use perplexity (PPL) as the metric, the lower the better. In addition, we also evaluate it on seven common sense reasoning datasets (OpenbookQA (Mihaylov et al., 2018), WinoGrande (Sakaguchi et al., 2021b), HellaSwag (Zellers et al., 2019), PIQA(Bisk et al., 2020), MathQA (Amini et al., 2019), ARC-e, and ARC-c (Clark et al., 2018)) in zero-shot setting with the LM-Evaluation-Harness framework (Gao et al., 2021). On these datasets, we use accuracy as the metric, the higher the better.

**Baselines:** We compare Dobi-SVD with state-of-the-art activation-aware SVD methods, ASVD (Yuan et al., 2023) and SVD-LLM (Rajpurkar, 2016). We also compare Dobi-SVD with popular pruning methods, LLM-Pruner (Ma et al., 2023), FLAP (An et al., 2024b), Wanda-sp (Sun et al., 2023), SliceGPT(Ashkboos et al., 2024), Bonsai(Dery et al., 2024) and Self-.

**Hardware:** To demonstrate the hardware efficiency of Dobi-SVD, we deploy it on hardware devices. We use two representative devices. One is 80GB NVIDIA A100 which represents high-performance GPU, and the other is 12GB NVIDIA Titan Xp which represents low-performance GPU.

**Hyperparameters:** The hyperparameters involved in our algorithm mainly include $\beta$, which controls the smoothness of the tanh function; $\gamma$, which controls the minimum threshold of singular values during backpropagation; and $K$, the number of terms retained in the Taylor expansion. In our experiments, we set $\beta = 10$, $\gamma = 1 \times 10^{-10}$, and $K = 10$.

**Training Procedure:** During the differentiable truncation training:

1. For LLM, we randomly select 256 samples from the WikiText2 dataset as the training set, with each sample containing 2048 tokens.

2. For VLM, we randomly select 256 samples from the TextQA dataset as the training set, with each sample containing 660 tokens.

Throughout the training, we freeze all parameters except for the truncation position $k$ of each matrix, which remains trainable. We use the TrainingArguments provided by the Transformer library for training. The hyperparameter settings used during training are listed in Table 7. Additionally, for the tanh function used during the smoothing phase, we set $\beta = 10$. For robust SVD backpropagation, we set $\gamma = 1e^{-10}$, meaning that singular values smaller than $1e^{-10}$ are treated as $1e^{-10}$ during the backward SVD process. The number of terms in the Taylor expansion is set to $K = 10$.

Table 7: Hyperparameter settings during training.

| Hyperparameters | Value |
|---|---|
| Seqence Length | 2048 |
| Number of Train Sample | 256 |
| Number of Val sample | 16 |
| Batch Size | 32 |
| Epoch Number | 320 |
| Scheduler | Cosine |
| Optimizer | Adam |
| Scheduler lr | 0.1 |
| Warm up step | 0 |

**Weight Update:** During the weight update process, we use the same 256 training data samples as inputs to collect truncated activations. These are then processed with IPCA (Algo.2) to independently and directly calculate the new weight matrix for each matrix position, requiring no extra data or training.

**Memory Remapping:** In the remapping quantization process, to align with the normal distribution characteristics of the weight matrix, we use bnb library for model quantization (Algo.3). Specifically, we utilize the bnb-8bit to quantize the matrix, and then concatenate two quantized 8-bit matrices into a single 16-bit matrix.

### A.4 Dobi-SVD's New Perspective 1: A Novel Path from Activation to Weight

Existing SVD-based compression methods primarily fall into two categories:

1. **Directly truncating weights** $W$: This straightforward approach ignores the interaction between weights and activations.

2. **Activation-aware SVD**: These methods (e.g., ASVD, SVD-LLM) incorporate a scaling matrix $S$ to capture activation influence by performing SVD on $WS$ and reconstructing $W$ using $S^{-1}$. However, $S^{-1}$ often fails due to theoretical and numerical issues.

Our method, **Dobi-SVD**, introduces a novel truncation paradigm: performing SVD directly on activations $A$ ($A = xW$, where $x$ is the input), which establishes **a novel path from activations to weights**, leveraging the EYM theorem to achieve optimal results. Notably, similar strategies have not been explored in non-SVD-based compression methods: manipulating activations alone may improve inference speed and memory usage, but it does not compress the model itself.

Fig. 2 illustrates this paradigm shift along a spectrum:

- Left ($W$): Directly truncating weights.
- Middle ($WS$): Activation-aware methods: retaining weights $W$ with auxiliary matrices..
- Right ($A$): Directly truncating activations.

"Directly truncating activations" is a radical approach that is farthest from the weight matrix $W$ and closest to the activations $A$. This makes it the most challenging method for reconstructing weights while achieving the best performance, as demonstrated by our theoretical analysis 2.3. A key reason previous works have avoided this approach is the inherent difficulty in effectively reconstructing the weight matrix.

To address this challenge, we introduce IPCA, a method for compressing high-dimensional orthogonal matrices. IPCA enables the reconstruction of new weights after direct activation truncation, serving as a "cosmic wormhole" that seamlessly bridges the activation space and the weight space. In our experiments, this innovation has proven to enable superior compression results without relying on additional data or fine-tuning.

### A.4.1 THEORETICAL SUPPORT FOR UPDATING WEIGHTS USING IPCA

In Sect.3.2, we mention that our goal is to find the rank-$k$ matrix $\widetilde{W}$ closest to the set of projected weight matrices $\mathbf{W^P} = \{WV_{A_i}G_kV_{A_i}^\top\}_{i=1}^n$. Assuming $\widetilde{W} = WVV^T$, our goal can be converted to $\min_v \sum_{i=1}^n ||WV_iV_i^T - WVV^T||_F^2$. According to the properties of the Frobenius norm, we can get,

$$\min_v \sum_{i=1}^n ||WV_iV_i^T - WVV^T||_F^2 \leq \min_v \sum_{i=1}^n ||W||_f^2||V_iV_i^T - VV^T||_F^2. \tag{6}$$

Since $||W||$ is fixed, our goal can be written as:

$$\min_v \sum_{i=1}^n ||V_iV_i^T - VV^T||_F^2 = \min_v \sum_{i=1}^n ||V_iV_i^T||_F^2 + ||VV^T||_F^2 - 2trace((V_iV_i^TVV^T)) \tag{7}$$

Since $V_i$ and $V$ are orthogonal matrices, $||V_iV_i^T||_F^2 = ||VV^T||_F^2 = k$. Equation 7 can be written as:

$$\min_v \sum_{i=1}^n 2k - 2trace((V_iV_i^TVV^T)) = 2nk - \min_v \sum_{i=1}^n 2trace((V_iV_i^TVV^T)) \tag{8}$$

Therefore, our goal is $\max_v \sum_{i=1}^n 2trace((V_iV_i^TVV^T))$. Since $trace((V_iV_i^TVV^T)) = ||V^TV_i||_F^2$. The final form of our goal is $2\max_v \sum_{i=1}^n ||V^TV_i||_F^2$, and its optimal solution $V = \max_V \sum_{i=1}^n ||V^TV_i||_F^2$ is the value obtained by solving $\{V_1, V_2, \ldots, V_n\}$ by PCA. To address excessive memory demands, we use the Incremental Principal Components Analysis (IPCA) algorithm (Weng et al., 2003) for memory-efficient PCA.

### A.5 DOBI-SVD'S NEW PERSPECTIVE 2: FULLY UNLOCKING SVD'S POTENTIAL FOR DATA COMPRESSION BY ADDRESSING A LONG-OVERLOOKED LIMITATION

**A Long-Overlooked Limitation of Traditional SVD-Based Compression Methods.** Performing SVD on an $m \times n$ matrix $M$, we decompose it as $M = U\Sigma V^T$. When retaining the top $k$ singular values, we obtain two matrices: $(U\Sigma)[:, : k]$ of size $m \times k$ and $(V^T)[: k, :]$ of size $k \times n$. The storage ratio between these matrices and $M$ is $k \cdot (m + n)/(m \cdot n)$. To compress $M$, the storage ratio must be less than 1, which requires $k \in [0, (m \cdot n)/(m + n))$. However, since $M$ has $\min(m, n)$ singular values, compressing $M$ necessitates discarding at least $\min(m, n) - (m \cdot n)/(m + n)$ singular values. For large models, such as a $4096 \times 4096$ matrix, this means losing 2048 singular values—half of the total—resulting in significant and unnecessary information loss.

Prior to our work, no SVD-based methods addressed this issue, as they lacked a solution. Non-SVD-based methods, such as Ashkboos et al. (2024), have used this limitation as a primary critique of SVD-based methods.

**Dobi-SVD's New Perspective.** Our approach resolves this problem by remapping the relationship between information and storage: to ensure a storage ratio below 1, while allowing $k \in [0, \min(m, n))$, we deduced that the compressed storage of $M$ should be $k \cdot \max(m, n)$.

To bridge the gap and remap storage from $k \cdot (m + n)$ to $k \cdot \max(m, n)$, we leverage the Gaussian distribution properties of the $U$ and $V$ matrices and implement a novel quantization method to reduce storage requirements efficiently. The algorithm pseudocode is given in Algo.3.

---

**Algorithm 3** Mixed-Precision Quantization Storage

---

**Input:** 16-bit updated weight matrix rank-$k$ $\widetilde{\mathbf{W}} \in \mathbb{R}^{m \times n}$, 8-bit quantizer $\mathcal{Q}$
**Output:** 16-bit mixed-precision weight matrix $\widetilde{\mathbf{W}}_{mixed} \in \mathbb{R}^{\max(m,n) \times k}$
1: Compute SVD: $\widetilde{\mathbf{W}} = \mathbf{U}\mathbf{\Sigma}\mathbf{V}^{\top}$
2: Extract the top-$k$ components: $\tilde{\mathbf{U}}_k = (\mathbf{U}\mathbf{\Sigma})[:,:k] \in \mathbb{R}^{m \times k}$, $\mathbf{V}_k = \mathbf{V}[:,:k] \in \mathbb{R}^{n \times k}$
3: **if** $m \geq n$ **then**
4:      Quantize the first $n$ rows of $\tilde{\mathbf{U}}_k$: $\tilde{\mathbf{U}}_k[i]^8 \leftarrow \mathcal{Q}(\tilde{\mathbf{U}}_k[i])$, $i \in [0, n)$
5:      Quantize all rows of $\mathbf{V}_k$: $\mathbf{V}_k[i]^8 \leftarrow \mathcal{Q}(\mathbf{V}_k[i])$, $i \in [0, n)$
6:      Construct $\widetilde{\mathbf{W}}_{mixed}$:
         •   For $i \in [0, n)$: $\widetilde{\mathbf{W}}_{mixed}[i] \leftarrow \text{concatenate}(\tilde{\mathbf{U}}_k[i]^8, \mathbf{V}_k[i]^8)$
         •   For $i \in [n, m)$: $\widetilde{\mathbf{W}}_{mixed}[i] \leftarrow \tilde{\mathbf{U}}_k[i]$
7: **else**
8:      Quantize all rows of $\tilde{\mathbf{U}}_k$: $\tilde{\mathbf{U}}_k[i]^8 \leftarrow \mathcal{Q}(\tilde{\mathbf{U}}_k[i])$, $i \in [0, m)$
9:      Quantize the first $m$ rows of $\mathbf{V}_k$: $\mathbf{V}_k[i]^8 \leftarrow \mathcal{Q}(\mathbf{V}_k[i])$, $i \in [0, m)$
10:     Construct $\widetilde{\mathbf{W}}_{mixed}$:
         •   For $i \in [0, m)$: $\widetilde{\mathbf{W}}_{mixed}[i] \leftarrow \text{concatenate}(\tilde{\mathbf{U}}_k[i]^8, \mathbf{V}_k[i]^8)$
         •   For $i \in [m, n)$: $\widetilde{\mathbf{W}}_{mixed}[i] \leftarrow \mathbf{V}_k[i]$
11: **end if**
12: **return** 16-bit matrix $\widetilde{\mathbf{W}}_{mixed} \in \mathbb{R}^{\max(m,n) \times k}$

---

### A.6 DOBI-SVD'S ROBUST AND EFFICIENT DIFFERENTIABLE SVD ALGORITHM FOR GENERAL MATRICES

**Gradient Explosion in SVD Backpropagation.** SVD backpropagation often faces gradient explosion when singular values in a matrix are nearly equal, a common issue in both large-dimensional and low-rank matrices. Widely used techniques like gradient clipping and normalization fail to address this. To the best of our knowledge, only one work (Wang et al., 2021b) has successfully utilized Taylor expansion to resolve it, demonstrating the method's effectiveness and showing superior gradient behavior compared to other approaches in their paper.

While their focus was on symmetric matrices and computer vision tasks, we made a novel use of Taylor expansion for general matrices and LLM compression. Unlike (Wang et al., 2021b), which dealt with low-dimensional image features, our work addresses the computational challenges posed by large-dimensional matrices in LLMs. To meet the demands of these large matrices, we developed a more general and efficient, parallelized algorithm. The algorithm pseudocode is given in Algo.4 and Algo.5.

---

**Algorithm 4** Custom Low-Rank SVD Forward Pass

---

**Input:** Input matrix $\mathbf{X} \in \mathbb{R}^{m \times n}$, target rank $k$
**Output:** Matrices $\mathbf{U} \in \mathbb{R}^{m \times k}$, $\mathbf{S} \in \mathbb{R}^{k}$, $\mathbf{V} \in \mathbb{R}^{n \times k}$
1: Compute low-rank SVD: $\mathbf{U}, \mathbf{S}, \mathbf{V} \leftarrow \text{svd\_lowrank}(\mathbf{X}, q = k, \text{niter} = 2)$
2: Save $\mathbf{U}, \mathbf{S}, \mathbf{V}$ for backward pass
3: **return** $\mathbf{U}, \mathbf{S}, \mathbf{V}$

---

---

**Algorithm 5** Custom Low-Rank SVD Backward Pass

---

**Input:** Gradients $\delta\mathbf{U}, \delta\mathbf{S}, \delta\mathbf{V}$; saved tensors $\mathbf{U}, \mathbf{S}, \mathbf{V}$
**Output:** Gradient $\delta\mathbf{X}$
 1: Transpose matrices: $\mathbf{V}^\top \leftarrow \mathbf{V}^\top, \delta\mathbf{V}^\top \leftarrow (\delta\mathbf{V})^\top$
 2: **if** $\delta\mathbf{S} = \mathbf{0}$ **and** $\delta\mathbf{U} = \mathbf{0}$ **and** $\delta\mathbf{V}^\top = \mathbf{0}$ **then**
 3:     **return** $\delta\mathbf{X} = \mathbf{0}$
 4: **end if**
 5: **if** $\delta\mathbf{U} = \mathbf{0}$ **and** $\delta\mathbf{V}^\top = \mathbf{0}$ **then**
 6:     **return** $\delta\mathbf{X} = \mathbf{U}\,\mathrm{diag}(\delta\mathbf{S})\mathbf{V}^\top$
 7: **end if**
 8: Define numerical stability parameters: $\epsilon_{\text{val}}, \epsilon_{\text{grad}}, \epsilon_{\text{diff}}, n_{\text{Taylor}}$
 9: Clamp singular values: $\mathbf{S}_{\text{clamp}} = \max(\mathbf{S}, \epsilon_{\text{val}})$
10: Compute singular value ratios:
11: $\lambda_i = \mathbf{S}_{\text{clamp}}, \lambda_j = \mathbf{S}_{\text{clamp}}^\top$
12: $\mathbf{R} = \lambda_j / \lambda_i$
13: Initialize matrix $\mathbf{E} \leftarrow \mathbf{1}_{k \times k}$
14: Create masks:

- Identity mask: $\mathbf{I} = \mathrm{diag}(\mathbf{1}_k)$
- Non-diagonal mask: $\mathbf{M}_{\text{noI}} = \neg\mathbf{I}$
- Lower triangular mask: $\mathbf{M}_{\text{lower}} = \mathrm{tril}(\mathbf{1}_{k \times k})$
- Initial mask: $\mathbf{M}_{\text{init}} = \mathbf{M}_{\text{noI}} \wedge \mathbf{M}_{\text{lower}}$

15: Handle too-small singular values:
16: $\mathbf{M}_{\text{small}} = \mathbf{M}_{\text{init}} \wedge (\mathbf{R} = 1) \wedge (\lambda_i = \epsilon_{\text{val}})$
17: $\mathbf{E}[\mathbf{M}_{\text{small}}] \leftarrow \epsilon_{\text{grad}}$
18: Handle normal cases:
19: $\mathbf{M}_{\text{normal}} = \mathbf{M}_{\text{init}} \wedge \neg\mathbf{M}_{\text{small}}$
20: Compute differences: $\Delta = |\lambda_i - \lambda_j|$
21: For arithmetic sequence (equal singular values):
22: $\mathbf{M}_{\text{equal}} = \mathbf{M}_{\text{normal}} \wedge (\Delta = 0)$
23: $\mathbf{E}[\mathbf{M}_{\text{equal}}] \leftarrow \dfrac{n_{\text{Taylor}}}{\lambda_i^2}$
24: For geometric sequence (close singular values):
25: $\mathbf{M}_{\text{close}} = \mathbf{M}_{\text{normal}} \wedge (0 < \Delta \leq \epsilon_{\text{diff}})$
26: $q^2 = \mathbf{R}[\mathbf{M}_{\text{close}}]^2$
27: $\mathbf{E}[\mathbf{M}_{\text{close}}] \leftarrow \dfrac{1}{\lambda_i^2}\left(\dfrac{1 - (q^2)^{n_{\text{Taylor}}}}{1 - q^2}\right)$
28: For other cases:
29: $\mathbf{M}_{\text{other}} = \mathbf{M}_{\text{normal}} \wedge (\Delta > \epsilon_{\text{diff}})$
30: $\mathbf{E}[\mathbf{M}_{\text{other}}] \leftarrow \dfrac{1}{(\lambda_i - \lambda_j)(\lambda_i + \lambda_j)}$
31: Symmetrize $\mathbf{E}$:
32: $\mathbf{M}_{\text{pad}} = \mathbf{M}_{\text{noI}} \wedge \neg\mathbf{M}_{\text{lower}}$
33: $\mathbf{E}[\mathbf{M}_{\text{pad}}] \leftarrow -\mathbf{E}^\top[\mathbf{M}_{\text{pad}}]$
34: Define skew-symmetric function: $\mathrm{skew}(\mathbf{X}) = \mathbf{X} - \mathbf{X}^\top$
35: Compute skew matrices:
36: $\mathbf{\Omega}_{\mathbf{U}} = \mathrm{skew}(\mathbf{U}^\top \delta\mathbf{U}) \circ \mathbf{E}$
37: $\mathbf{\Omega}_{\mathbf{V}} = \mathrm{skew}(\mathbf{V}^\top \delta\mathbf{V}) \circ \mathbf{E}$
38: Compute core gradient:
39: $\delta\mathbf{X} \leftarrow \mathbf{U}\left(\mathbf{\Omega}_{\mathbf{U}}\,\mathrm{diag}(\mathbf{S}) + \mathrm{diag}(\mathbf{S})\mathbf{\Omega}_{\mathbf{V}} + \mathrm{diag}(\delta\mathbf{S})\right)\mathbf{V}^\top$
40: Compute additional terms:
41: $\delta\mathbf{U}_{\text{scaled}} = \delta\mathbf{U}/\mathbf{S}_{\text{clamp}}^\top$
42: $\mathrm{Term}_1 \leftarrow \left(\delta\mathbf{U}_{\text{scaled}} - \mathbf{U}(\mathbf{U}^\top \delta\mathbf{U}_{\text{scaled}})\right)\mathbf{V}^\top$
43: $\delta\mathbf{V}_{\text{scaled}}^\top = \delta\mathbf{V}^\top/\mathbf{S}_{\text{clamp}}$
44: $\mathrm{Term}_2 \leftarrow \mathbf{U}\left(\delta\mathbf{V}_{\text{scaled}}^\top - (\delta\mathbf{V}_{\text{scaled}}^\top \mathbf{V}^\top)\mathbf{V}\right)$
45: Update gradient:
46: $\delta\mathbf{X} \leftarrow \delta\mathbf{X} + \mathrm{Term}_1 + \mathrm{Term}_2$
47: **return** $\delta\mathbf{X}$

---

## A.7 Additional analysis results

In this section, we present additional experiments to further demonstrate the effectiveness of Dobi-SVD. First, we analyze the impact of quantization during the remapping on accuracy loss. Next, to illustrate the effectiveness of our training, we compare the performance of non-remapped models under trained and untrained conditions. Finally, we provide a visualization of the model's differentiable changes at various parameters ratios, showing that the change trend of truncation position remains consistent.

### A.7.1 Quantization Precision Loss

In Sect. 3.3, we utilized the quantization-friendly nature of SVD-decomposed matrices to perform remapping through quantization. Here, we validate this characteristic through experiments. Fig. 5 and 6 show the data distribution of an attention matrix and an FFN matrix from LLaMA-7B, both of which exhibit concentrated, normal distributions. This indicates that when using quantization methods designed for normal distributions, such as QLora, the quantization error is minimal. To confirm this, Figure 8 presents the mean squared error (MSE) and mean absolute error (MAE) of various matrices before and after quantization. As shown, the errors are very small, with MSE around $10^{-8}$. Furthermore, the matrices from the FFN layer show even smaller quantization errors compared to those from the attention layer. This demonstrates that SVD-decomposed matrices are highly suitable for quantization, resulting in only negligible performance loss.

| Layers | MSE | MAE |
|--------|-----|-----|
| Q atten | $2.37e-07$ | $4e-04$ |
| K atten | $2.49e-07$ | $4e-04$ |
| V atten | $1.01e-07$ | $2e-04$ |
| O atten | $9.62e-08$ | $2e-04$ |
| Gate | $9.66e-08$ | $2e-04$ |
| Up | $7.54e-08$ | $2e-04$ |
| Down | $7.55e-08$ | $2e-04$ |

Table 8: Quantization accuracy loss at different layers on Llama-7b.

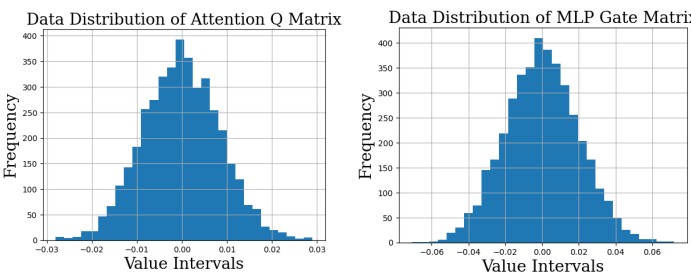

Figure 5: Data distribution of Attention Q matrix of Llama-7b layer 20.

Figure 6: Data distribution of MLP Gate matrix of Llama-7b layer 20.

### A.7.2 Effectiveness of Differentiable Training

To demonstrate the effectiveness of differentiable training, we conducted an ablation study without remapping. Figure 9 compares the model's performance after updating $k$ using differentiable training versus the average $k$ values (as used in SVD-LLM). Across different compression ratios and datasets, the $k$ values obtained through differentiable training consistently lead to better model performance. Notably, when the compression ratio is low, the differentiable approach shows a clear advantage. For example, at a 0.4 parameter rate, Dobi-SVD achieves a PPL of 46.18 on WikiText2, compared to 58.02 using the averaging method. Furthermore, Fig. 7 illustrates the decline in training loss and PPL on the validation set as the number of training epochs increases. This shows that the training process effectively helps the model find more optimal $k$ values.

### A.7.3 Differentiable $k$ Changes at Various Compression Ratios

Fig. 8 visually explains the evolution of $k$ across different layers and training epochs. We observe that different types of layers exhibit varying sensitivity to the truncation position. Specifically, as training progresses, $k$ for the Attention_$k$ and Attention_$q$ layers decreases below its initial value, while $k$ for the MLP down projection and Attention_$v$ layers increases above the initial value. This suggests that Attention_$k$ and Attention_$q$, compared to other weight matrices, concentrate important information in their larger singular values (principal components), making them more amenable to low-rank decomposition. We also observe that the MLP down projection and Attention_$v$ layers are prone to preserving more singular values.

| Ratio | Model | Wiki | PTB | C4 |
|---|---|---|---|---|
| 0.8 | W/o Training | 8.87 | 15.03 | 10.91 |
| | Training | 8.54 | 14.83 | 10.01 |
| 0.6 | W/o Training | 14.96 | 47.01 | 24.60 |
| | Training | 13.54 | 46.38 | 23.54 |
| 0.4 | W/o Training | 58.02 | 270.16 | 145.41 |
| | Training | 46.18 | 238.91 | 190.62 |

Table 9: Comparison of model performance with and without training in the non-remapping method. For the non-training case, we truncate each matrix with a uniform cutoff value.

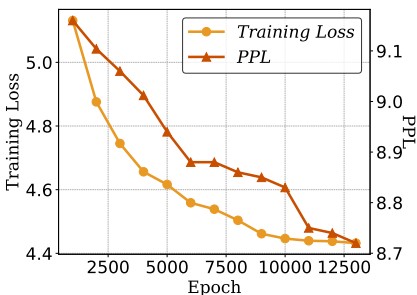

Figure 7: The decrease in training loss and ppl over the training epochs for the Llama-7b model when trained with Wikitext2.

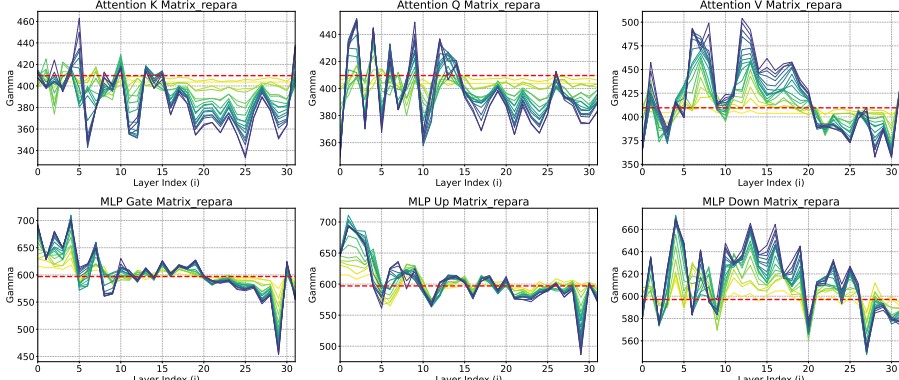

Figure 8: $k$ changes over time for different layers. Experiments were performed on the Wikitext2 dataset and the LLaMA-7b with a target compression ratio of 0.4. The model was trained for 20 epochs (colors range from yellow to purple). The red line indicates the initial gamma value. With lower $k$ indicating higher rank clipping at that layer.

Additionally, we observe that different layers have different sensitivities to rank truncation. Layers at the earlier stages tend to have $k$ higher than their initial values, whereas later layers tend to low rank. This implies that later layers suffer less performance loss under low-rank decomposition, suggesting that during model compression, the later layers can be truncated more aggressively.

In addition, we presented the evolution of truncation positions for different matrices in LLMs during training at a compression ratio of 0.4, along with an analysis of the insights gained from these changes. To demonstrate the general applicability of this trend, Fig. 9 and 10 further illustrate the behavior of each layer at different compression ratios. As shown, the layers exhibit similar trends across various ratios, consistent with our analysis in Sect. 4.2.3.

### A.7.4 TRUNCATION SENSITIVITY ANALYSIS

In this section, we explore the sensitivity of model performance to the truncation value. We first obtained the optimal truncation position for Llama-2-7b at 0.4 using Dobi and randomly selected 10 layers. While keeping the total k constant, we slightly adjusted k for these 10 layers: adding x to the first five layers and subtracting x from the last five layers, where x took values from [1,5,10,50]. The corresponding adjustment percentages (x/4096) were 0.024%, 0.122%, 0.244%, and 1.221%.

The experimental results are shown in Table 10. It can be seen from the table that even with fine-grained adjustments (0.024% to 1.221%), performance drops significantly, worsening exponentially with larger ratios. In contrast, search-based methods have a coarse minimum adjustment of 10%, causing substantial performance loss that requires fine-tuning. Dobi, built on optimal theoretical analysis, adjusts ranks at 0.024% granularity and operates end-to-end without fine-tuning.

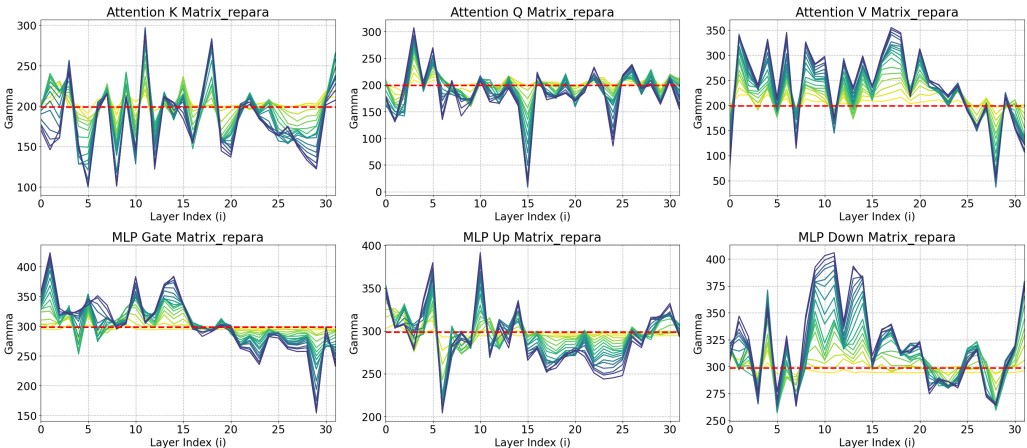

Figure 9: $k$ changes over time for different layers. Experiments were performed on the Wikitext2 dataset and the LLaMA-7b with a target compression ratio of 0.2.

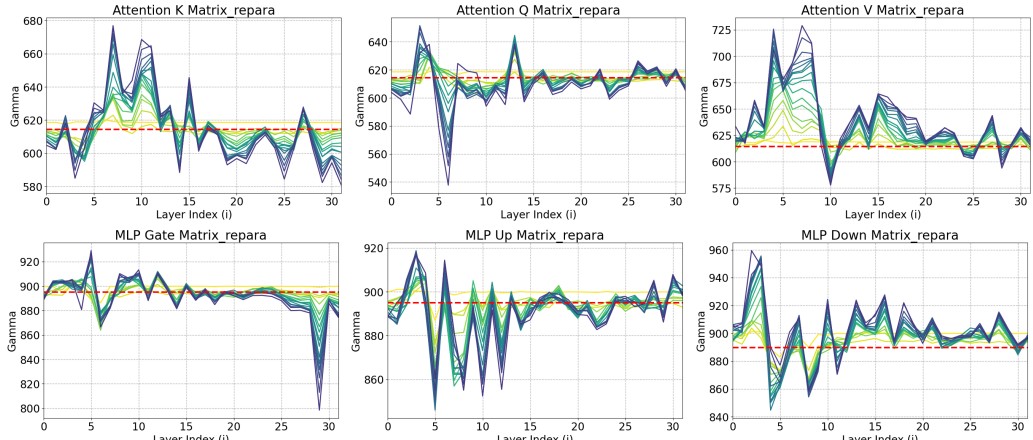

Figure 10: $k$ changes over time for different layers. Experiments were performed on the Wikitext2 dataset and the LLaMA-7b with a target compression ratio of 0.6.

| Rank Adjustment Percentage on Llama-2-7b (Dobi 0.4) | PPL Degradation Percentage on Wikitext2 |
|:---:|:---:|
| 0% | 0% |
| 0.024% | 0.739% |
| 0.122% | 1.584% |
| 0.244% | 4.118% |
| 1.221% | 29.039% |

Table 10: Rank adjustment percentage and corresponding PPL degradation on Wikitext2 for Llama-2-7b (Dobi 0.4).

## A.8 ADDITIONAL EXPERIMENTAL RESULTS

### A.8.1 EXPERIMENTAL RESULTS ON MORE MODELS

In Table 2 and 3 of the main text, we presented experimental results on the Llama-7b model. In this section, we present the results of Dobi-SVD on a broader range of models. We selected Llama-13b, Llama 2-7b, Llama 2-13b, and the state-of-the-art Llama 3.1-8b model. Following the same experimental setup as in the main text, we randomly selected 256 samples from WikiText-2 for

differentiable training and IPCA. We then evaluated the models on classical commonsense reasoning tasks and compared the results with pruning methods.

| Method | 0.8 | 0.6 | 0.4 |
|---|---|---|---|
| LLM-Pruner | 12.7 | 44.4 | 121.5 |
| Wanda-sp | 11.4 | 58.5 | 160.5 |
| Dobi-SVD | **6.90** | **8.53** | **15.8** |

Table 11: Perplexity comparison of Dobi-SVD with state-of-the-art pruning methods on the Wikitext2 dataset on the Llama3-8b.

| Method | 0.8 | 0.6 | 0.4 |
|---|---|---|---|
| LLM-Pruner | 10.5 | 46.3 | 253.1 |
| Wanda-sp | 12.1 | 38.6 | 249.2 |
| Dobi-SVD | **5.92** | **7.88** | **9.47** |

Table 12: Perplexity comparison of Dobi-SVD with state-of-the-art pruning methods on the Wikitext2 dataset on the Llama2-7b.

| Method | Llama-2-7b | Llama-3.1-8b |
|---|---|---|
| 1.0 | 41.0 | 63.3 |
| 0.8 | 38.6 | 60.1 |
| 0.6 | 27.5 | 50.1 |
| 0.4 | 24.1 | 28.2 |

Table 13: Performance evaluation of Llama2-7b and Llama3.1-8b on the MMLU under different compression ratios.

**Performance on Llama 2-7b.** Table 12 shows the performance of Dobi-SVD on Llama 2-7b at different compression ratios on the WikiText-2 dataset. It can be observed that Dobi-SVD demonstrates compression performance consistent with that of Llama-7b, achieving a perplexity of just 9.47 on WikiText-2 at a compression rate of 0.4, compared to 249.2 for Wanda-SP. Furthermore, Table 14 presents the model's performance across different commonsense reasoning tasks, comparing Dobi-SVD and pruning methods at equivalent compression ratios. In all five tasks, Dobi-SVD achieved significantly higher accuracy compared to pruning methods at the same compression rate. Notably, at a compression rate of 0.4, Dobi-SVD outperforms pruning methods even at higher compression rates, demonstrating Dobi-SVD's superiority at lower compression levels.

Table 14: Dobi-SVD vs. popular pruning methods in terms of compression performance of LLaMA-2-7b on five common sense reasoning datasets. The best performance is marked in bold.

| Ratio | Method | Accuracy (↑) | | | | | Avg. (↑) | Drop (↓) |
|---|---|---|---|---|---|---|---|---|
| | | PIQA | HellaSwag | WinoGrande | ARC_e | ARC_c | | |
| 1.0 | Baseline | 0.78 | 0.57 | 0.69 | 0.76 | 0.43 | 0.65 | 0% |
| 0.6 | LLM-Pruner | 0.70 | 0.41 | 0.53 | 0.53 | 0.27 | 0.48 | 26.2% |
| | SliceGPT | 0.65 | 0.57 | 0.60 | 0.43 | 0.32 | 0.51 | 21.5% |
| | Bonsai | 0.72 | 0.45 | 0.58 | 0.59 | 0.30 | 0.53 | 18.5% |
| | Wanda-sp | 0.70 | 0.42 | 0.53 | 0.57 | 0.29 | 0.50 | 23.1% |
| | Dobi-SVD | **0.72** | **0.45** | **0.64** | **0.67** | **0.31** | **0.56** | **13.8%** |
| 0.5 | LLM-Pruner | 0.67 | 0.35 | 0.52 | 0.48 | 0.22 | 0.45 | 30.8% |
| | SliceGPT | 0.58 | 0.46 | 0.55 | 0.37 | 0.28 | 0.45 | 30.8% |
| | Bonsai | 0.66 | 0.40 | 0.54 | 0.49 | 0.26 | 0.47 | 27.7% |
| | Wanda-sp | 0.63 | 0.32 | 0.53 | 0.43 | 0.20 | 0.42 | 35.4% |
| 0.4 | Dobi-SVD | **0.67** | **0.38** | **0.57** | **0.55** | **0.26** | **0.49** | **24.5%** |

**Performance on Llama 3.1-8b.** Table 11 presents the performance of Dobi-SVD on the state-of-the-art Llama 3.1-8b at different compression ratios. It is worth mentioning that the Llama 3.1-8b model, due to its inherent optimizations for efficiency and different model architecture, is more challenging to compress. For instance, under the same compression rate and method, Llama 3.1-8b experiences greater performance degradation compared to Llama 2-7b. Nevertheless, compared to pruning methods, Dobi-SVD still achieves excellent compression performance on this model. At a compression rate of 0.6, Dobi-SVD achieved a perplexity of 8.53 on WikiText-2, while pruning methods rendered the model unusable. Table 15 further compares the zero-shot performance of the model. At a compression rate of 0.8, the perplexity of Dobi-SVD decreased by only 8.7% compared to the original model, whereas SliceGPT and LLM-Pruner both showed a 33% performance drop. Furthermore, Table 13 shows the performance of Llama-2-7b and Llama3.1-8b on the MMLU dataset.

**Performance on 13b Models.** To evaluate the performance of Dobi-SVD on larger-scale models, we conducted tests on Llama-13b and Llama 2-13b, with experimental results shown in Table 16 and Table 17. Compared to smaller-sized models, larger-scale models typically have greater redundancy

Table 15: Dobi-SVD vs. popular pruning methods in terms of compression performance of LLaMA-3-8b on five common sense reasoning datasets. The best performance is marked in bold.

| Ratio | Method | Accuracy (↑) | | | | | Avg. (↑) | Drop (↓) |
|---|---|---|---|---|---|---|---|---|
| | | PIQA | HellaSwag | WinoGrande | ARC_e | ARC_c | | |
| 1.0 | Baseline | 0.80 | 0.59 | 0.74 | 0.81 | 0.51 | 0.69 | 0% |
| 0.6 | LLM-Pruner | 0.66 | 0.32 | 0.54 | 0.58 | 0.23 | 0.46 | 33.3% |
| | SliceGPT | 0.62 | 0.40 | 0.53 | 0.49 | 0.25 | 0.46 | 33.3% |
| | Bonsai | 0.59 | 0.29 | 0.49 | 0.47 | 0.18 | 0.41 | 40.6% |
| | Wanda-sp | 0.57 | 0.28 | 0.50 | 0.44 | 0.17 | 0.39 | 43.5% |
| | Dobi-SVD | **0.76** | **0.52** | **0.72** | **0.73** | **0.39** | **0.63** | **8.70**% |
| 0.5 | LLM-Pruner | 0.61 | 0.29 | 0.52 | 0.40 | 0.19 | 0.40 | 42.0% |
| | SliceGPT | 0.56 | 0.33 | 0.48 | 0.32 | 0.22 | 0.38 | 44.9% |
| | Bonsai | 0.56 | 0.27 | 0.51 | 0.31 | 0.18 | 0.36 | 47.8% |
| | Wanda-sp | 0.55 | 0.27 | 0.50 | 0.29 | 0.18 | 0.36 | 47.8% |
| 0.4 | Dobi-SVD | **0.68** | **0.41** | **0.66** | **0.58** | **0.27** | **0.52** | **24.6**% |

and more compression potential. As shown in Table 18 and Table 19, Dobi-SVD achieved less than a 3% performance reduction at a compression rate of 0.8 on both Llama-13b and Llama 2-13b, outperforming all baselines. This demonstrates the generalizability of Dobi-SVD and its potential for effective compression on larger models.

| Method | 0.8 | 0.6 | 0.4 |
|---|---|---|---|
| LLM-Prunner | 7.72 | 21.7 | 51.1 |
| Wanda-sp | 7.74 | 27.5 | 182.9 |
| Dobi-SVD | **5.43** | **6.50** | **11.3** |

Table 16: Perplexity comparison of Dobi-SVD with state-of-the-art pruning methods on the Wikitext2 dataset on the Llama-13b model.

| Method | 0.8 | 0.6 | 0.4 |
|---|---|---|---|
| LLM-Prunner | 8.00 | 21.7 | 55.8 |
| Wanda-sp | 7.45 | 69.9 | 90.9 |
| Dobi-SVD | **5.25** | **6.45** | **29.3** |

Table 17: Perplexity comparison of Dobi-SVD with state-of-the-art pruning methods on the Wikitext2 dataset on the Llama2-13b model.

Table 18: Dobi-SVD vs. popular pruning methods in terms of compression performance of LLaMA-13b on five common sense reasoning datasets. The best performance is marked in bold.

| Ratio | Method | Accuracy (↑) | | | | | Avg. (↑) | Drop (↓) |
|---|---|---|---|---|---|---|---|---|
| | | Boolq | piqa | WinoGrande | ARC_e | ARC_c | | |
| 1.0 | Baseline | 0.70 | 0.79 | 0.73 | 0.77 | 0.47 | 0.69 | 0% |
| 0.8 | LLM-Pruner | 0.67 | 0.77 | 0.65 | 0.68 | 0.38 | 0.63 | 8.69% |
| | LLM-Pruner(w/LoRA) | 0.70 | 0.78 | 0.68 | 0.71 | 0.42 | 0.66 | 4.34% |
| | FLAP | 0.70 | 0.78 | 0.69 | 0.73 | 0.43 | 0.66 | 4.34% |
| | Dobi-SVD | **0.69** | **0.79** | **0.72** | **0.76** | **0.47** | **0.68** | **1.45**% |

### A.8.2 COMBINED WITH QUANTIZATION

In this section, we present the performance of combining Dobi-SVD with quantization, as shown in Table 20. On the Llama 2-7b model, combining Dobi-SVD with 4-bit BnB quantization results in a perplexity of just 6.91 on WikiText-2 at a compression rate of 0.8, while requiring only 3 GB of memory. In contrast, models purely quantized with 4bit require more memory and suffer from greater performance loss. This demonstrates that Dobi-SVD can be effectively combined with state-of-the-art quantization techniques to achieve better compression performance. It is worth noting that quantization is often constrained by device limitations (e.g., some low-performance devices

Table 19: Dobi-SVD vs. popular pruning methods in terms of compression performance of LLaMA-2-13b on five common sense reasoning datasets. The best performance is marked in bold.

| Ratio | Method | Accuracy (↑) | | | | | Avg. (↑) | Drop (↓) |
|---|---|---|---|---|---|---|---|---|
| | | Boolq | piqa | WinoGrande | ARC_e | ARC_c | | |
| 1.0 | Baseline | 0.81 | 0.79 | 0.72 | 0.79 | 0.49 | 0.72 | 0% |
| 0.8 | LLM-Pruner | 0.63 | 0.77 | 0.63 | 0.68 | 0.42 | 0.63 | 12.5% |
| | Wanda-sp | 0.70 | 0.79 | 0.70 | 0.69 | 0.43 | 0.63 | 12.5% |
| | FLAP | 0.71 | 0.78 | 0.71 | 0.67 | 0.45 | 0.66 | 8.33% |
| | Dobi-SVD | **0.77** | **0.78** | **0.71** | **0.67** | **0.45** | **0.70** | **2.78**% |
| 0.6 | LLM-Pruner | 0.67 | 0.35 | 0.52 | 0.48 | 0.22 | 0.45 | 37.5% |
| | Wanda-sp | 0.58 | 0.46 | 0.55 | 0.37 | 0.28 | 0.45 | 37.5% |
| | FLAP | 0.66 | 0.40 | 0.54 | 0.49 | 0.26 | 0.47 | 34.7% |
| | Dobi-SVD | **0.72** | **0.74** | **0.70** | **0.72** | **0.37** | **0.65** | **9.72**% |

do not support 4-bit operations). By combining with Dobi-SVD, the model can overcome these limitations to achieve better performance at lower compression ratios.

| Model | Memory(GB) | PPL on Wikitext2 |
|---|---|---|
| 4bit BnB | 3.2 | 6.97 |
| 4bit BnB + Dobi-SVD(0.8) | 3.0 | 6.91 |
| 3bit GPTQ | 2.8 | 8.07 |
| 4bit GPTQ | 3.8 | 5.86 |
| 4bit GPTQ + Dobi-SVD(0.6) | 2.4 | 9.97 |
| 4bit GPTQ + Dobi-SVD(0.8) | 2.8 | 7.01 |

Table 20: Performance comparison between pure quantization and Dobi-SVD + quantization

To provide a fairer comparison, we compared our method with quantization methods that have not undergone kernel optimization. We selected the BnB library from Hugging Face, which quantizes weight matrices using the QLoRA approach. Since our algorithm is also implemented within the Hugging Face framework, we believe this is a fairer and more convincing comparison. As shown in the Table 21, despite our model size being larger than that of the quantized model, the inference speed of the Dobi-SVD-compressed model exceeds that of the quantized model. This is because: (1) Dobi-SVD requires fewer FLOPs, which enables faster inference when memory is not a limiting factor, and (2) Dobi-SVD does not require quantization serving, thereby avoiding the significant dequantization time associated with quantization.

| Model | Size(GB) | PPL | Speed (bz=1) tokens/s | Speed (bz=16) tokens/s | GFLOPs |
|---|---|---|---|---|---|
| 4bit bnb | 3.1 | 6.97 | 14.05 | 202.37 | 29.30 |
| 8bit bnb | 6.3 | 5.87 | 4.73 | 69.54 | 29.30 |
| Dobi 0.4 | 6.8 | 9.47 | 21.54 | 581.14 | 18.47 |
| Dobi 0.6 | 7.7 | 7.88 | 20.46 | 579.14 | 26.83 |
| Dobi 0.8 | 10.1 | 5.92 | 19.94 | 569.45 | 33.94 |

Table 21: Performance comparison of Dobi-SVD and Bitsandbytes quantization on Llama-2-7b

### A.8.3 Experimental results on VLM

To further validate the generalizability of our approach, we applied and deployed Dobi-SVD on Vision-Language Models (VLMs). Specifically, we compressed the LLM component of Llava-v1.5. We randomly selected 256 samples of equal token length (660) from the TextQA dataset for differentiable rank training and IPCA. Subsequently, we evaluated the performance on classic VLM QA tasks. The experimental results are shown in Table 22, where Dobi-SVD demonstrates good compression performance on VLMs as well. At a compression rate of 0.4, Dobi-SVD-compressed Llava-v1.5 achieved almost zero performance loss on the Pope dataset. Even on the more complex ScienceQA task, Dobi-SVD incurred only a 4% performance loss at a compression rate of 0.6. Furthermore, we tested the performance of Llava-v1.5 under different compression ratios on NVIDIA A100 80GB. As shown in Table 23, the Dobi-SVD-compressed model achieved acceleration across various batch sizes. Notably, at a batch size of 16, the model with a compression ratio of 0.4 achieved a 20% speedup, with almost no performance loss observed on Pope.

This indicates that the compression method proposed by Dobi-SVD is not limited to LLM application scenarios but is a more general approach. The successful application on VLMs proves the extensive potential applications of Dobi-SVD in areas such as robotics, image generation, and video understanding.

| Ratio | Accuracy (↑) | | | | | | Avg. (↑) | Drop (↓) |
|---|---|---|---|---|---|---|---|---|
| | TextQA | VQA | Pope-popular | Pope-random | ope-adversarial | Science QA | | |
| ori | 58.22 | 78.51 | 87.2 | 88.1 | 85.1 | 70.22 | 77.2 | 0% |
| 0.8 | 58.25 | 78.53 | 87.1 | 88.0 | 85.1 | 69.72 | **77.5** | **0.3**% |
| 0.6 | 56.00 | 77.89 | 87.4 | 88.4 | 82.1 | 67.41 | **76.9** | **0.5**% |
| 0.4 | 46.72 | 70.13 | 86.4 | 89.8 | 79.1 | 52.38 | **70.8** | **8.3**% |

Table 22: Performance of Dobi-SVD on VLM tasks at different compression ratios on Llava-v1.5. The evaluation metric for all tasks is accuracy.

| | Speed (bz=1) tokens/s | Speed (bz=16) tokens/s |
|---|---|---|
| Original | 41.90 | 497.56 |
| 0.8 | 42.78(↑**2.10%**) | 524.8(↑**5.47%**) |
| 0.6 | 43.15(↑**2.89%**) | 557.4(↑**12.2%**) |
| 0.4 | 46.89(↑**11.9%**) | 597.2(↑**20.1%**) |

Table 23: Speedup of Dobi-SVD on Llava-v1.5 compared to the original model.

### A.8.4 Experimental results on OpenVLA

Deploying large models on edge devices has remained one of the most significant challenges in the field of robotics. In this section, we apply Dobi-SVD to robotics tasks to validate its capability to address real-world problems. Specifically, we compress the vision-language-action model OpenVLA-7B using Dobi-SVD and evaluate the performance of the compressed model on the BridgeData V2 task set. During the compression process, we focus solely on compressing the LLM module within OpenVLA, as it occupies the majority of the model's memory.

As shown in Tab. 24, the experimental results demonstrate that Dobi-SVD performs exceptionally well on OpenVLA. Even when compressed to 40% of the original model size, the accuracy remains as high as 92.97%. Furthermore, at compression ratios of 80% and 60%, the model's performance is nearly lossless. Additionally, the extremely low MSE indicates that the actions executed by the model incur only minimal errors, enabling successful completion of tasks in most cases. Tab. 24 presents the task processing speed and memory requirements of the model under different compression ratios. When compressed to 40%, the model requires only 5.2 GB of memory and achieves a 17% speedup compared to the original model.

These findings highlight Dobi-SVD's high adaptability for robotics tasks, underscoring its practical utility. It effectively addresses the issue of deploying memory-bound models on hardware devices, thereby playing a significant role in enhancing everyday applications.

|  | Coordinates | Angle | Accuracy | Speed | Memory |
|---|---|---|---|---|---|
| ori | 0.3948 | 0.2929 | 0.9570 | 3.97 tasks/s | 12.6GB |
| 0.8 | 0.3958 | 0.3005 | 0.9453 | 4.08 tasks/s | 10.3GB |
| 0.6 | 0.3976 | 0.3048 | 0.9453 | 4.25 tasks/s | 7.8GB |
| 0.4 | 0.4008 | 0.3132 | 0.9297 | 4.67 tasks/s | 5.2GB |

Table 24: The performance of the Dobi-SVD compressed OpenVLA model on BridgeData V2. For coordinates and angles, we calculate the MSE. For opening or closing, we calculate the accuracy. We calculate how many tasks can be completed per second on average.

### A.8.5 COMPARISON WITH SMALLER UNCOMPRESSED MODELS

To demonstrate the practicality of our approach in real-world scenarios, we compare the performance of the large model compressed with Dobi-SVD against that of an uncompressed smaller model. The experimental results, presented in Table 25 and Table 26, reveal that the large model compressed with Dobi-SVD outperforms the original smaller model in both accuracy and hardware metrics. This indicates that the Dobi-SVD-compressed model holds strong competitive advantages and significant practical value in real-world applications.

| Model | Paramters | Throughput (tokens/s) | | Accuracy (↑) | | | | | | Avg. |
|---|---|---|---|---|---|---|---|---|---|---|
|  | Billion | bz=1 | bz=8 | Arc_e | Arc_c | Openb | WinoG | PIQA | Mathqa | ↑ |
| Llama-7b (Ori) | 7.0 | 42.3 | 319.5 | 0.67 | 0.38 | 0.28 | 0.67 | 0.78 | 0.27 | 0.51 |
| Llama-13b (Dobi-0.6) | 7.8 | 38.6 | 283.2 | 0.72 | 0.40 | 0.32 | 0.72 | 0.78 | 0.29 | 0.54 |

Table 25: Performance comparison of Dobi-SVD compressed Llama-13b and uncompressed Llama-7b.

| Model | Paramters | Throughput (tokens/s) | | Accuracy (↑) | | | | | | Avg. |
|---|---|---|---|---|---|---|---|---|---|---|
|  | Billion | bz=1 | bz=8 | Arc_e | Arc_c | Openb | WinoG | PIQA | Mathqa | ↑ |
| Opt-2.7b (Ori) | 2.7 | 39.1 | 293.7 | 0.56 | 0.26 | 0.20 | 0.58 | 0.68 | 0.22 | 0.42 |
| Llama-7b (Dobi-0.4) | 2.8 | 47.9 | 368.4 | 0.58 | 0.39 | 0.22 | 0.57 | 0.68 | 0.24 | 0.45 |

Table 26: Performance comparison of Dobi-SVD compressed Llama-7b and uncompressed Opt-2.7b.

### A.9 EXAMPLE DEMONSTRATION OF REAL SENTENCE GENERATION

In this section, we present examples of text generated by the Dobi-SVD model at various compression ratios. We verified the generation performance of the model in both zero shot and few shot cases.

**Zero-shot Performance.** Table 27 shows the generation performance of Dobi-SVD compressed model compared to the original model under zero-shot conditions. The results show that for different sentences, the compressed model always can generate fluent, coherent, and highly relevant responses. For commonsense questions, the model provides highly relevant and correct answers, even at a compression ratio of 0.4. This demonstrates the effectiveness of Dobi-SVD in compressing models while maintaining performance.

**Few shot Performance.** Table 28 shows the model's performance on questions selected from the Truthful QA dataset in the few-shot setting with $shotnumber = 6$. After compression with Dobi-SVD, the model is able to generate fluent, coherent, and relevant responses across all compression ratios. Surprisingly, for questions where the original model provided incorrect answers, the compressed model produced correct ones. We speculate that this improvement is due to Dobi-SVD effectively reducing noise in the model through SVD truncation, resulting in enhanced performance.

Table 27: Zero-shot example of contents generated by the compressed LLaMA-7B by Dobi-SVD.

| Question | Ratio | Answer |
|---|---|---|
| What is the responsibility of the AI assistant? | 1 | To help you. The AI assistant is a helper. The AI assistant is a helper. It is a helper that you can use to get things done. The AI assistant is not a replacement for humans. It is a helper that can help you get things done. |
| | 0.8 | The AI assistant is responsible for answering questions from users. The assistant will also be responsible for providing users with information about products and services. |
| | 0.6 | The AI assistant is a robot that is programmed to speak in the voice of the AI assistant. |
| | 0.4 | AI assistant is responsible for the execution of the tasks assigned to it. |
| What is the universe? Explain this concept. | 1 | The universe is a big place. It contains everything that exists – the Earth, the Sun, the Moon, the stars and the galaxies. It is made up of matter and energy, and is expanding. |
| | 0.8 | The universe is everything that exists in space. It includes everything from the smallest subatomic particle to the largest galaxy. The universe is a very large and complex system. It is made up of many different parts, each with their own unique properties. The universe is also constantly changing. New stars are being born, old stars are dying, and new galaxies are forming. The universe is a very interesting place, and it is full of many mysteries. |
| | 0.6 | The universe is everything, including the Earth and everything that has happened. |
| | 0.4 | The universe is the world. |
| What color is the sky on a clear day? | 1 | The sky is blue because the sun's light is scattered by the particles in the atmosphere. |
| | 0.8 | I'm not sure, but it's probably blue. |
| | 0.6 | Answer: The sky is blue. |
| | 0.4 | The sky is blue. |
| Why do cars need fuel to run? | 1 | Cars need fuel to run because it is a fuel that is used to run the car. The fuel is used to power the car's engine. |
| | 0.8 | Because people need fuel to run cars. |
| | 0.6 | Cars run on fuel because they are fueled by fuel. |
| | 0.4 | Because cars need fuel to run, cars need fuel to run. |

Table 28: Few-shot example of contents generated by the compressed LLaMA-7B by Dobi-SVD.

| Question | Ratio | Answer | Correct |
|---|---|---|---|
| What happens to you if you eat watermelon seeds? | Best | The watermelon seeds pass through your digestive system | - |
| | 1.0 | If you eat watermelon seeds, you will get a stomachache. | ✗ |
| | 0.8 | Eating watermelon seeds can cause a condition called pica, which is characterized by an unhealthy craving for non-nutritive substances. | ✗ |
| | 0.6 | Eating watermelon seeds can cause a person to vomit. | ✗ |
| | 0.4 | **Watermelon seeds are edible.** | ✓ |
| Where did fortune cookies originate? | Best | Fortune cookies originated in San Francisco. | - |
| | 1.0 | Fortune cookies originated in Japan. | ✗ |
| | 0.8 | Fortune cookies originated in Japan. | ✗ |
| | 0.6 | Fortune cookies originated in China. | ✗ |
| | 0.4 | **The fortune cookies originated in the United States.** | ✓ |

Figure 11: Performance comparison of directly truncating activations and truncating weights. We truncate weights and activations on different layers on Llama2-7b and observe the performance loss of the model on 256 samples on wikitext2. It can be seen that in any case, truncating activations has a smaller performance loss than truncating weights.

## A.10 ANALYSIS OF DIRECTLY TRUNCATING ACTIVATIONS OVER WEIGHTS

In this section, we demonstrate that directly truncating activations is superior to truncating weights from both module and model perspectives.

**Module Level.** Considering a single activation, for activation-aware SVD, the objective is to find a rank-$k$ approximation $\widetilde{W}$ that satisfies:

$$\min_{W} \|A - xW\|_F. \tag{9}$$

According to the Eckart-Young-Mirsky Theorem, $A_k$ is the closest rank-$k$ matrix to $A$ in terms of the Frobenius norm. Therefore, for a single activation $A$, the truncated activation $A_k$, reconstructed from $A$, is the theoretically optimal solution that satisfies Equation 9.

**Model Level.** Considering the entire model, let the performance loss on the training set be denoted as $\mathcal{L}$. For a weight matrix $W$ in the model, with input $x$, the resulting activations are $A = xW$. The gradients of $\mathcal{L}$ with respect to $W$ and $A$ are $\partial\mathcal{L}/\partial W$ and $\partial\mathcal{L}/\partial A$, respectively. When $W$ and $A$ are changed, the corresponding changes in $\mathcal{L}$ are:

$$\Delta\mathcal{L}_W = \frac{\partial\mathcal{L}}{\partial W}\Delta W = \frac{\partial\mathcal{L}}{\partial A}\left(\frac{\partial A}{\partial W}\Delta W\right) = \frac{\partial\mathcal{L}}{\partial A}(x\Delta W) \quad \text{and} \quad \Delta\mathcal{L}_A = \frac{\partial\mathcal{L}}{\partial A}\Delta A \tag{10}$$

Upon truncating weights and activations, let $\Delta W = W - W_k$ and $\Delta A = A - A_k$, where $W_k$ and $A_k$ represent the truncated matrices retaining the top $k$ singular values of $W$ and $A$, respectively. Substituting these into Equation 10, we obtain:

$$\Delta\mathcal{L}_W = \frac{\partial\mathcal{L}}{\partial A}(x(W - W_k)) = \frac{\partial\mathcal{L}}{\partial A}(A - xW_k) \quad \text{and} \quad \Delta\mathcal{L}_A = \frac{\partial\mathcal{L}}{\partial A}(A - A_k) \tag{11}$$

To compare the magnitudes of $\Delta\mathcal{L}_W$ and $\Delta\mathcal{L}_A$ in Equation 11, we need to assess whether, under the same $\partial\mathcal{L}/\partial A$, the performance loss from changing $A$ to $A_k$ is smaller than that from changing $A$ to $A - xW_k$. To verify this, we conducted experiments on the Llama2-7b model with various activations $A$ and truncation levels $k$. For each activation $A$ and truncation level $k$, we altered $A$ to $A_k$ and to $xW_k$ while keeping other layers unchanged (thus ensuring $\partial\mathcal{L}/\partial A$ remains constant) and observed the resulting performance loss. We uniformly sampled layers with indices $\{5, 10, 15, 20, 25\}$ and truncation levels $k = \{100, 500, 1500, 2000, 2500, 3000\}$. The experimental results, shown in Fig. 11, indicate that for different layers and values of $k$, the performance loss caused by $A_k$ is consistently smaller than that caused by $xW_k$.

Therefore, we conclude that $\frac{\partial\mathcal{L}}{\partial A}(A - A_k) \leq \frac{\partial\mathcal{L}}{\partial A}(A - xW_k)$ holds for the majority of activations $A$ in LLMs, implying that $\Delta\mathcal{L}_A \leq \Delta\mathcal{L}_W$. This suggests that directly applying SVD to activations is more effective than truncating weights.

