# OpenReview forum: "Dobi-SVD: Differentiable SVD for LLM Compression and Some New Perspectives"
_ICLR.cc/2025/Conference — ICLR 2025 Poster_

### Official Review · Reviewer_uXRu · 2024-11-02

**Soundness:** 3
**Presentation:** 2
**Contribution:** 3
**Rating:** 8
**Confidence:** 4

**Summary:**

The paper performs an in-depth investigation of post-training low-rank compression of LLMs. In brief, its contributions are as follows:

- It begins by showing that “activation-level” compression (i.e. minimizing || xW - xW^ ||_F rather than || W - W^ ||_F) is superior in the case of low-rank compression. While this is useful, there are already at least two prior papers investigating this type of compression, and this is standard for both quantization and pruning.

- It then performs a deep-dive into low-rank compression, and proposes a regularization-based approach for “learnable” truncation factors across layers, an approach for performing weight updates to correct for the truncation error, and a remapping quantization-based approach which removes some of the limitations of SVD in the low-compression regime.

- These algorithmic proposals are validated experimentally via post-training compression of Llama-7B. This shows significant improvements relative to prior SVD and structured pruning-based methods. The authors also provide end-to-end speedup evaluations.

**Strengths:**

- The paper is reasonably well-written, easy to follow, and contains a systematic treatment of the SVD approach for LLMs

- The approach and conclusions described in the paper seem valid

- Some of the techniques provided (learned truncation, remapping) are interesting and could prove generally useful

**Weaknesses:**

- The submission slightly overstates its novelty, as the “activation” SVD approach has already been investigated, and learned truncation can be seen as an instance of compression-aware regularization, which has been used in other settings. See e.g. ShearedLlama or the older Soft Threshold Reparametrization work, which should be cited.


- Unfortunately though, the experimental results are a bit of a mess. They are badly presented, and I believe that the methodology may be flawed: see my comments below. Nevertheless, I do believe that the improvements being presented can probably be supported.

Overall, the paper has interesting technical results, and with writing adjustments and improved methodology it should be acceptable to ICLR.



Comments on the experiments:

1. First, the reader has to dig into the Appendix to understand how exactly the experiments are performed (basically, they fine-tune the method parameters over WikiText2 calibration data). However, this leads to a number of questions:

a. I would say that fine-tuning over WikiText2 data and then measuring PPL over the same dataset is not a great idea, and definitely gives you an advantage over prior work that seems to be purely one-shot. It doesn’t seem to me like you’ve run these comparisons fairly: you appear to be copy-pasting results from prior work, but wouldn’t it be fair to at least fine-tune their compressed models for a bit on Wiki2? Same for pruning methods. If you’d like a hard baseline to compare against, check out ShearedLlama. (They also provide their one-shot pruned model, without fine-tuning with dynamic batching.)

b. It would be great to present the zero-shot eval numbers in e.g. Table 2 as NORMALIZED accuracy, i.e. where you subtract the random classification performance from the number you present. (Then, a random classification model would have 0% accuracy.) If you do that, you might notice that your technique has near-random accuracy on ARC-Challenge at higher compression rates.

c. Metrics: Relatedly, the paper presents an unfortunate transition from the abstract which claims “minimal” accuracy loss, to the actual results tables, which show compression losses what are hardly “minimal,” from 4% at 20% compression to 23% or more to 60% compression.
Could you please add MMLU evaluations on Llama2? Why don’t you include evaluations on the more recent Llama3 models?

d. Generally, I would also suggest removing or rephrasing the discussions based on PPL from the intro: given that PPL isn’t really a metric that we can directly relate to any sort of practical task performance, a 78% reduction in perplexity increase doesn’t really mean anything. You could cite the average drop in zero-shot as a much more “actionable” metric.

e. The quoted improvements over “the best pruning methods” are false, because you are not comparing against the best pruning methods, which, to my knowledge, are SliceGPT and ShearedLlama.

**Questions:**

In sum, I think this paper introduces some interesting ideas, but that the presentation and methodology are both currently lacking. I’d be happy to revise my opinion if additional data is presented. See major questions above.

Additional comments:

- Why is there no comparison with SliceGPT?

- Lines 450-455: The K and V compressibility observation has been known at least since Scatterbrain.

- Line 515: This eval is weird and should be removed. Yes, you can get 10x or more speedup if you compare against a method with RAM offloading, but this is very apples-to-oranges. How about you compare against a 4-bit quantized method, which should fit fine on the GPU?

- Line 521: Do you observe a reduction in potential speedups for quantization for a model that’s already been SVD-compressed, given that you are moving away from the pure memory-bound mode of the GPU?

- The time quoted for fine-tuning fluctuates quite a lot (once it’s 8h, it’s 4h in another place…)

- L255: “the gradient is the devil”?! Please rephrase.

- “causing g_A goes to infinite” This whole part of the paper needs A LOT of additional polish in terms of writing.

---

> ### Author Response · Authors · 2024-11-24
> **Response to Reviewer uXRu (Part 1/5)**
>
> We thank the reviewer for taking the time to review our paper and provide valuable feedback. We are glad that the reviewer thinks the paper is reasonably well-written, easy to follow, and contains a systematic treatment of the SVD approach for LLMs. We are pleased to address the reviewer’s questions and concerns. The comments will make the paper stronger.
>
> ***Question 1. The submission slightly overstates its novelty.***
>
> **A1.** We thank the reviewer for bringing the referenced literature and we have cited them in our revision. We respectfully suggest that the reviewer may have slightly overlooked the novelty related to the activation in our work.
>
> 1. **Our Activation-Aware Framework for SVD Differs Fundamentally from Previous Activation-Aware Frameworks to other compression methods.**
>
>    Activation-aware and low-rank techniques have been extensively studied in quantization (e.g., LLM.int8(), SmoothQuant, AWQ, SqueezeLLM) and pruning (e.g., NetAdapt, WandA, adaptive activation-based structured pruning). These methods treat activation distance as an optimization goal and approach the learned truncation as a form of compression-aware regularization as noted by reviewer uXRu. This perspective is also employed in ASVD and SVD-LLM.
>
>    However, we would like to emphasize that our method extends beyond the aforementioned framework: **we are the first to directly truncate activations.** Building on the foundational EYM theorem of SVD, we rigorously prove that directly truncating activations is optimal, surpassing all other approaches, including weight truncation **and activation-aware methods**, as detailed in Lines 195 to 206. **This approach has not been explored in prior SVD-based methods or even across a range of non-SVD-based techniques.** Consequently, we believe both the optimal direct-truncation solution and the accompanying theoretical explanation of its optimality make our work uniquely stand apart from previous activation-aware frameworks.
>
>
>
> 2. **Why haven’t previous works directly truncated activations?**
>
>    **Because they can not easily reconstruct low-rank weights from the truncated activations!**
>
>    In addition to the emphasizing that we are the first to prove that directly truncating activations is the optimal solution for minimizing ∣∣A−A′∣∣, we would like to highlight that reconstructing low-rank weights from truncated activations was previously impossible. In our **Appendix A4.1 “Theoretical Support for Updating Weights Using IPCA” Line 937-958**, we theoretically demonstrate how weights can be reconstructed from truncated activations. Furthermore goes beyond the theoretical solution, we also address practical challenges of footprint explosion issues through a novel use of IPCA. Additionally, **the Appendix A.4,** ***"Dobi-SVD's New Perspective 1: A Novel Path from Activation to Weight"*** **and Fig. 4 provide a more intuitive understanding.**
>
>
>
> 3. **Dobi handles singular value information during the "learned truncation" process differently from the "coarse" approaches of previous methods.**
>    For instance, ASVD and SVD-LLM predefine singular value selection ratios (e.g., 0.1, 0.2, ..., 0.9) for each matrix and use a search algorithm to determine the best combination under such constraints. In contrast, as reviewer sX49 commented, **we are the first work considering all singular values to attempt identifying the "optimal" truncation position.** This is a natural idea, and others may have considered it, but no prior work implemented it. The reason is that they were unable to achieve the efficiency and robustness in backpropagation for general matrix SVD as Dobi did. Additionally, **we provided experiments in our response to reviewer sX49's first question, showing the sensitivity of the truncation position to the results, further supporting the necessity of investigating the "optimal" position.**
>
> Based on the above discussion, we believe our work demonstrates greater innovation and significance than initially evaluated by the reviewer. We hope our explanations provide the reviewer with a more comprehensive understanding, highlighting that: (1) our approach is fundamentally distinct from previous SVD-based compression methods, and (2) we make both theoretical and engineering contributions to achieve and implement the optimal solution effectively.

---

> ### Author Response · Authors · 2024-11-24
> **Response to Reviewer uXRu (Part 2/5)**
>
> ***Question 2. Fine-tuning over WikiText2 data and then measuring PPL over the same dataset is not fair.***
>
> A2. We politely point out the reviewer may have the misconception for our experiment setting.
>
> 1. **We are not a fine-tuning method.**
>
>    Different from the fine-tuning method that optimizes relatively heavy weights with many samples, we adopt a standard approach similar to previous activation-aware compression methods like SqueezeLLM and SVD-LLM. Specifically, we collect only 256 activation samples to optimize the truncation rank and use these same samples to reconstruct the weights. In contrast to fine-tuning-based methods, our optimization is parameter efficient (just 224 parameters for LLAMA1 and LLAMA2) and sample-efficient (end-to-end only using 256 calibration samples)
>
> 2. **Our setting is consistent with Baseline.** Our experimental setup is consistent with the baselines we compare against. We followed SVD-LLM to randomly select 256 samples from WikiText-2 as the calibration data to do post-training. Our ppl evaluation code and setting are the same as SVD-LLM. Additionally, SVD-LLM uses fine-tuning and we do not.
>
> 3. **We perform better even under unfair comparison.** We did not report our actual experimental results because the SVD-LLM results we obtained were much worse than those reported in their paper. For Llama-7B, under the same experimental conditions, the actual results we obtained are as follows:
>
> | Algorithm | Method         | Method         | 0.4   | 0.4   | 0.4   | 0.6    | 0.6    | 0.6    | 0.8   | 0.8   | 0.8   |
> | --------- | -------------- | -------------- | ----- | ----- | ----- | ------ | ------ | ------ | ----- | ----- | ----- |
> |           | Percetion      | Truncation     | Wiki  | PTB   | C4    | Wiki   | PTB    | C4     | Wiki  | PTB   | C4    |
> | ASVD      | Weight Scaling | Rank Searching | 43104 | 49862 | 40960 | 4833   | 9791   | 1952   | 9.05  | 13.80 | 10.20 |
> | SVD-LLM   | Data Whitening | Averaging      | 458   | 1194  | 1540  | 113.64 | 300.78 | 151.74 | 27.54 | 50.76 | 38.02 |
>
> As shown in the table, the results for SVD-LLM are significantly worse than those reported in their paper. Despite our best efforts to reproduce their performance, we were unable to complete the "fine-tuning" phase of SVD-LLM due to issues documented in their GitHub repository (#6 and #7), specifically the error: "cannot compute the inverse of the scaling matrix." We hypothesize that this may stem from GPU compatibility or numerical precision issues. To avoid any perception of bias or accusations of intentionally worsening their results, we have opted to use the results reported in their arXiv paper directly. Note that our method is robust to these issues of compatibility and numerical stability. **Additionally, we recently saw a new pruning method MoDeGPT [1] make comparisons in the same way. And Dobi-SVD still outperforms it:** At a compression ratio of 0.6, MoDeGPT achieves 9.39, while Dobi-SVD achieves 8.12; at 0.8, MoDeGPT is 6.53, and Dobi-SVD is 6.08.
>
> Besides, It is also noteworthy that even under this unfair comparison—where SVD-LLM uses LoRA (ICLR 2025 submitted version, L254)—our results still outperform theirs (both their arXiv version and the newly submitted ICLR version). This not only demonstrates the robustness and applicability of Dobi-SVD but also highlights the effectiveness and competitiveness of our method derived through optimal theoretical analysis.
>
> We have made our experiment settings more clear in the appendix.
>
> [1] MoDeGPT: Modular Decomposition for Large Language Model Compression (arXiv Table 6)

---

> > ### Author Response · Authors · 2024-11-24
> > **Response to Reviewer uXRu (Part 3/5)**
> >
> > **Question 3. It would be great to present the zero-shot eval numbers in e.g. Table 2 as NORMALIZED accuracy**
> >
> > **A3.** Thanks for this interesting idea :) Here we provide some of our thoughts regarding standardized accuracy:
> >
> > 1. **Use of Absolute Accuracy**: In our submission, we used absolute accuracy to maintain the same comparison protocol with previous works. In fact, in prior work on both model pruning and SVD-based methods, the absolute accuracy was used more frequently. Therefore, to facilitate a more straightforward comparison with earlier works, we reported the absolute accuracy.
> >
> > 1. **Standardized accuracy as a metric**: This is a very interesting idea! However, we believe it is not a fully expressive metric for evaluating LLMs. As the reviewer uXRu mentioned, the standardized accuracy is 0 for certain tasks at a 0.4 compression rate, but we do not interpret this to mean that the model is entirely incapable. LLMs are designed to learn and encode knowledge from data, and task performance reflects the extent of that knowledge embedded within the model. Thus, even if the model's performance falls below random accuracy on certain tasks, it does not imply an absence of learning capability. To this end, we still prefer to use absolute accuracy as the metric.
> >
> > | Method              | Ratio | Openb. | ARC_e | ARC_c | WinoG. | HellaS. | PIQA | MathQA |
> > | ------------------- | ----- | ------ | ----- | ----- | ------ | ------- | ---- | ------ |
> > | SVD                 | 0.8   | 0      | 0.42  | 0.05  | 0.17   | 0.58    | 0.29 | 0.15   |
> > | ASVD                | 0.8   | -0.03  | 0.34  | 0.03  | 0.13   | 0.55    | 0.27 | 0.13   |
> > | SVD-LLM (+) | 0.8   | 0      | 0.36  | 0.04  | 0.15   | 0.57    | 0.28 | 0.14   |
> > | Dobi-SVD*           | 0.8   | -0.01  | 0.38  | 0.03  | 0.14   | 0.58    | 0.28 | 0.14   |
> > | Dobi-SVD            | 0.8   | 0.02   | 0.39  | 0.01  | 0.16   | 0.58    | 0.29 | 0.15   |
> > | SVD                 | 0.6   | -0.01  | 0.27  | 0     | 0.08   | 0.49    | 0.26 | 0.09   |
> > | ASVD                | 0.6   | -0.06  | 0.21  | 0.03  | 0.08   | 0.49    | 0.28 | 0.1    |
> > | SVD-LLM (+) | 0.6   | -0.09  | 0.32  | 0.02  | 0.11   | 0.52    | 0.24 | 0.11   |
> > | Dobi-SVD*           | 0.6   | -0.03  | 0.33  | 0.02  | 0.11   | 0.53    | 0.24 | 0.1    |
> > | Dobi-SVD            | 0.6   | -0.08  | 0.28  | -0.01 | 0.1    | 0.51    | 0.24 | 0.1    |
> > | SVD                 | 0.4   | -0.13  | 0.25  | 0.04  | 0.08   | 0.46    | 0.26 | 0.09   |
> > | ASVD                | 0.4   | -0.1   | 0.23  | 0.03  | 0.06   | 0.47    | 0.25 | 0.1    |
> > | SVD-LLM (+) | 0.4   | -0.12  | 0.31  | 0.02  | 0.1    | 0.5     | 0.21 | 0.07   |
> > | Dobi-SVD*           | 0.4   | 0.01   | 0.29  | 0     | 0      | 0.49    | 0.22 | 0.09   |
> > | Dobi-SVD            | 0.4   | -0.06  | 0.3   | 0.01  | 0.1    | 0.51    | 0.23 | 0.09   |
> >
> > Dobi-SVD* refers to the result obtained without remapping.  And (\+) uses LoRA fine-tuning .
> >
> >
> >
> > ***Question 4. The experimental results are a bit of a mess. More experimental results on Llama2 and Llama3***
> >
> > **A4.** We appreciate your suggestions. We have added these experiments in our revised version, refer to Section *[PPL on WikiText2 Across Different Models and Compression Ratios ]* and Section *[Performance Comparison of Dobi-SVD and Popular Pruning Methods on Various Tasks ]* in general response for details.
> >
> >
> >
> > ***Question 5. Comparing against the best pruning methods including SliceGPT and ShearedLlama..***
> >
> > **A5.** Thank you for pointing this out! We have updated the terminology to "popular pruning method" and compared our approach with additional pruning methods in Appendix Tab. 11, 12, 14 and 15. Experimental results demonstrate that our method outperforms these pruning methods by a significant margin. Regarding Sheared-LLAMA, we have made great effort to include it in our comparisons; however, their official code is not designed to align with our experimental settings. Re-implementing their method to fit our setup would require considerable effort. We plan to conduct a more comprehensive comparison with pruning-based methods like Sheared-LLAMA in future work.

---

> > > ### Author Response · Authors · 2024-11-24
> > > **Response to Reviewer uXRu (Part 4/5)**
> > >
> > > ***Question 6. Why is there no comparison with SliceGPT?***
> > >
> > > **A6.** We have provided the comparison results in our experiments (see Table 14 and Table 15). For example, on the Llama2-7b and Llama 3.1-8b models, Dobi-SVD improves the average performance of five commonsense reasoning tasks by 10% and 37% over SliceGPT at a compression ratio of 0.6, respectively.
> > >
> > >
> > >
> > > ***Question 7. Lines 450-455: The K and V compressibility observation has been known at least since Scatterbrain.***
> > >
> > > **A7.** Thank you for bringing this literature and we have cited it in our revised version. It provides strong mutual evidence that the compressibility of K and V represents a promising direction for future research in the community. We are pleased to have re-identified this issue and hope that it can once again inspire further advancements in the field.
> > >
> > >
> > >
> > > ***Question 8. Line 515: This eval is weird and should be removed. Yes, you can get 10x or more speedup if you compare against a method with RAM offloading, but this is very apples-to-oranges. How about you compare against a 4-bit quantized method, which should fit fine on the GPU?***
> > >
> > > **A8.** We kindly insist on retaining this evaluation. We conducted it with practical motivations for downstream applications, aiming to help LLM users understand how LLMs function and how our method enhances efficiency on limited compute resources. In fields like robotics and edge-device applications, where users often rely on low-cost GPUs with limited memory bandwidth and outdated CUDA architectures, our evaluation offers practical guidance on leveraging our method to accelerate LLMs. The 10x improvement demonstrates that our general compression method is crucial for effectively addressing these stringent demands.
> > >
> > > In contrast, quantization methods are not always ready to meet practical demands. These methods heavily rely on inference-serving frameworks where operators such as dequantization are specifically optimized. Without such optimization, they cannot effectively reduce LLM latency. We point out these inference-serving frameworks often lack general compatibility across different GPU architectures. For instance, when we attempted to run the kernel of AWQ-4bitl on a Titan XP 12GB, it failed due to GPU architecture compatibility issues.
> > >
> > > To further support this justification, we compress the LLM using bnb quantization into 4-bit and test the latency without kernel acceleration. Note that this comparison is entirely fair as our Dobi-SVD method does not depend on any kernel-specific or serving framework optimizations.
> > >
> > > | Model    | Size   | PPL  | Speed (bz=1)   | Speed (bz=16)   | Flops        |
> > > | -------- | ------ | ---- | -------------- | --------------- | ------------ |
> > > | 4bit bnb | 3.1GB  | 6.97 | 14.05 tokens/s | 202.37 tokens/s | 29.3 GFLOPS  |
> > > | 8bit bnb | 6.3GB  | 5.87 | 4.73 tokens/s  | 69.54 tokens/s  | 29.3 GFLOPS  |
> > > | Dobi 0.4 | 6.8GB  | 9.47 | 21.54 tokens/s | 581.14 tokens/s | 18.47 GFLOPS |
> > > | Dobi 0.6 | 7.7GB  | 7.88 | 20.46 tokens/s | 579.14 tokens/s | 26.83 GFLOPS |
> > > | Dobi 0.8 | 10.1GB | 5.92 | 19.94 tokens/s | 569.45 tokens/s | 33.94 GFLOPS |
> > >
> > > Here, we present a comparison between our method and the BnB quantization method on the NVIDIA A100 GPU. As shown, while our memory footprint is not as low as that of quantization-based compression, we achieve significant latency improvements. For instance, at a compression ratio of 0.4, our method processes 21.54 tokens per second on the A100, which is 1.5 times faster than BnB 4-bit quantization.
> > >
> > > This acceleration can be attributed to two key factors:
> > >
> > > 1. **Reduced FLOPs**: At a compression ratio of 0.4, our method reduces FLOPs to approximately half of the original model. This significantly enhances efficiency since the performance of large language models (LLMs) depends on both memory bandwidth and compute bandwidth.
> > > 2. **No Kernel Serving Overhead**: Unlike quantization, our method does not rely on kernel serving. Quantization methods often incur additional latency due to the dequantization process during inference, which our approach eliminates.
> > >
> > > These advantages highlight the effectiveness of our method in balancing compression and computational efficiency.

---

> > > > ### Author Response · Authors · 2024-11-24
> > > > **Response to Reviewer uXRu (Part 5/5)**
> > > >
> > > > ***Question 9. Line 521: Do you observe a reduction in potential speedups for quantization for a model that’s already been SVD-compressed, given that you are moving away from the pure memory-bound mode of the GPU?***
> > > >
> > > > **A9.** Our proposed Dobi-SVD does not compromise the potential speedups from quantization. By decomposing a weight matrix of size $m \times n$ into two matrices of sizes $m \times k$ and $n \times k$, our method preserves the linearity of the weight-activation multiplication, ensuring that it does not diminish the potential benefits of quantization. Moreover, our approach significantly reduces FLOPs. As shown in the table above, at a compression ratio of 0.4, our FLOPs is only half of the FLOPs of the original model. This benefit can also be leveraged by subsequent quantization.
> > > >
> > > >
> > > >
> > > > ***Question 10. The time quoted for fine-tuning fluctuates quite a lot (once it’s 8h, it’s 4h in another place)***
> > > >
> > > > **A10.** This has been corrected. In L251, "Llama2" should be "Llama1."
> > > >
> > > >
> > > >
> > > > ***Question 11. L255: “the gradient is the devil”?! Please rephrase.***
> > > >
> > > > **A11.** Thank you for the question. To quote the saying, "The devil is in the details" [(Wikipedia link)](https://en.wikipedia.org/wiki/The_devil_is_in_the_details), we found that, in this case, the devil is in the gradient! While it seems straightforward to use an end-to-end gradient-based approach to determine the optimal K, why have previous works not achieved this? The challenge lies in the fact that end-to-end differentiable optimization is highly susceptible to gradient explosion. To address this, our work makes a novel use of Taylor expansion to effectively mitigate this issue.
> > > >
> > > >
> > > >
> > > >
> > > >
> > > > ***Question 12. “causing g_A goes to infinite” This whole part of the paper needs A LOT of additional polish in terms of writing.***
> > > >
> > > > **A12.** We have polished it in the revised version in Line 258-269. We appreciate your suggestion. We hope the new version will provide you with more clarity.

---

> ### Author Response · Authors · 2024-11-30
> **Looking Forward to Your Feedback**
>
> We hope this message finds you well and in good spirits. We would like to express our sincere gratitude for your invaluable contribution as a reviewer. Your detailed feedback and thoughtful suggestions have been immensely helpful. Over the past two weeks, we have incorporated a significant amount of experimental data and theoretical explanations, and we are pleased to inform you that we have thoroughly addressed all the concerns raised.  **As the rebuttal phase nears its end, we look forward to your feedback.**
>
> Below, we have provided a summary of the current version of our paper, which we hope will assist in your evaluation.
>
> Thanks to the invaluable suggestions and support from the reviewers, we have refined Dobi-SVD, especially the two novel perspectives: '*A Novel Path from Activation to Weight'* and *'Fully Unlocking SVD’s Potential for Data Compression by Addressing a Long-Overlooked Limitation.'* We believe that the **current version of Dobi-SVD is backed by solid theory, presents novel insights, and offers significant contributions, deserving of a higher score.** Specifically, we highlight the following key points:
>
> - **Innovative Approach**: Dobi-SVD does not follow any existing quantization, pruning, or SVD-based methods but instead introduces a new way of handling activations and weights based on the fundamental theory of SVD (the EYM theorem).
> - **Efficient and Robust SVD Backpropagation**: We have developed an efficient and robust method for general matrix SVD backpropagation, which serves as a direct reference for others exploring low-rank, high-dimensional, and multi-form SVD matrices.
> - **Addressing a Long-Overlooked Yet Critical Limitation**: By leveraging the Gaussian distribution of singular value vectors, we’ve remapped singular value information to storage space, overcoming a longstanding limitation overlooked by SVD-based methods and criticized by non-SVD approaches, thus unlocking SVD’s full compression potential.
> - **Comprehensive Comparisons**: We have provided a broad comparison across various model compression methods, demonstrating superior practical performance.
>   - Not only do we outperform existing SVD-based methods (General Response Part 4), but we also generally outperform a wide range of pruning methods(General Response Part 2).
>   - In the comparison related to quantization (General Response Part 3), we demonstrate that our approach reduces FLOPs and highlight that more effective integration of SVD-based methods and quantization is a promising direction for future exploration.
> - **Generalizability**: Dobi-SVD’s successful application to Vision-Language Models (VLMs) highlights its broad applicability, extending beyond LLMs to areas like robotics. To the best of our knowledge, this is the first demonstration of such generalizability.
>
> Once again, we deeply appreciate the time and effort you have dedicated to reviewing our work, and we wish you a wonderful day ahead.

---

> > ### Comment · Reviewer_uXRu · 2024-11-30
> > **Response**
> >
> > Dear authors,
> >
> > Thank you for your detailed responses, and for the recent follow-up. I have examined your responses in their entirety.
> >
> > Unfortunately, I cannot agree that your submission is deserving of a higher score than 6 at this time. Specifically, I see the following issues, which I do not believe can be fixed via discussion:
> >
> > 1. In order to lead to significant speedups, your method drops significant accuracy, to the point where the model is unusable at higher speedup ratios. This is clearly stated in your own response, as the normalized accuracies table has zero (random accuracy) on the harder tasks. (By the way, your table is obviously missing the 1.0 baseline. Why?)
> >
> > 2. I believe that some of the comparisons you provided in your response vs e.g. BNB are deceiving: you provide FLOPS and throughputs, but not accuracies. Let us remember that 8bit BNB is essentially _lossless_, and that its 4bit version is near-lossless (approx. 2-3% drop), whereas your proposed scheme drops significant accuracy (as your own results above show, and becomes even clearer in your Llama-3.1-8B MMLU results). Moreover, the FLOP comparison doesn't make sense (BNB is a scheme for reducing memory movement, not FLOPS). If you wanted to perform a fair comparison against quantization, you would either compare throughput with a kernel that can support larger batch sizes (see e.g. the MARLIN kernel in vLLM, or FLUTE), or throughput / FLOPS on a model with FP8/INT8/INT4 weights and activations.
> >
> > 3. For the record, my suggestion on this point is not that you have to do that: SVD and quantization are complementary. My point, which you may consider for a future revision is to examine _the Pareto frontier_ in terms of accuracy (PPL or average over zero-shots) vs speedup (actual latency/throughput at e.g. BS1 or BS8). From your results, while your scheme may be competitive w.r.t. speedup, it does not appear to lead to reasonable results in terms of this trade-off from the point of view of accuracy. To my point, if your method requires the user to drop 8.7% accuracy for a ~20% improvement in throughput (Llama-3-8B), the user may just be better off running a smaller model (Llama-3.2-3B among many others) or a significantly quantized one. \
> > This is a fundamental weakness you may try to investigate in future work.
> >
> > 4. Finally, I just want to mention that I have also examined the VLM results. They seem reasonable, but the authors' claim that their technique _improves_ performance is really over-the-top: this happens in one out of 4 benchmarks, and may well be because of random variability in the evaluation. In any case, this one result doesn't make a lot of sense to me. On the other 3/4 benchmarks, their method predictably drops accuracy.
> >
> > In closing, I want to say that I really do appreciate the authors' effort during the rebuttal period, but I cannot in good conscience provide further support for their submission at this time, simply because their scheme does not seem to be able to lead to useful models in terms of the Pareto accuracy-speedup frontier.

---

> ### Author Response · Authors · 2024-12-02
> **Clarifications and Explanations Regarding Some Misunderstandings (Part 1/3)**
>
> We fully respect the reviewer’s authority to evaluate our work and greatly appreciate the reviewer’s willingness to detail the remaining concerns. **However, regarding some of the concerns and shortcomings raised, we believe that certain points are not based on accurate facts.** We would like to provide the following clarifications and explanations.
>
> 1. ***We believe the reviewer’s accusation of our quantization comparison being deceptive is unfounded.***
>
>    **First, regarding the comparison with BNB, we provided PPL results in our revised paper (Table 20 and Table 21) and also in General Response（Part 3/5),** contrary to the claim that *"you provide FLOPS and throughputs, but not accuracies."* Additionally, our results demonstrate that compared to BNB, we achieve higher throughputs at similar perplexity levels.
>
>    **Second, BNB’s assertion of being "essentially lossless" relies on the use of QLoRA [1] mapping, which requires LoRA fine-tuning to restore performance.** Without employing LoRA, BNB shows an increase in perplexity of 6.3% and 26.2% on WikiText2 for 8-bit and 4-bit configurations, respectively, when applied to Llama2-7b (Table 21 in our paper).
>
>    **Furthermore, we disagree that *"FLOP comparison doesn't make sense."* We believe that FLOPs is a meaningful metric.** Specifically, when the model inference batch size is large, the model transitions from being memory-bound to computation-bound, making the reduction of FLOPs significant. Regarding existing quantization methods, let’s remember that when the batch size is greater than one, the performance of current quantization kernels varies greatly across different hardware platforms. Even with small batch sizes, acceleration on certain hardware, such as some NVIDIA GPUs, falls far short of expectations. For example, with a batch size of 128, classic quantization kernels like AWQ, exllamav2, and BNB achieve less than 0.5× acceleration [2]. In contrast, our method benefits from lower FLOPs and hardware versatility, achieving 1.4× and 1.6× acceleration at compression ratios of 0.6 and 0.4, respectively. Therefore, as we mentioned in our previous response, **our advantage lies in the fact that our method does not require selecting kernels based on specific hardware constraints and provides higher acceleration with large batch sizes.**

---

> ### Author Response · Authors · 2024-12-02
> **Clarifications and Explanations Regarding Some Misunderstandings (Part 2/3)**
>
> 2. ***We think comments regarding performance are unreasonable.***
>
>    **First, non-quantization compression methods are not unuseful.** In fact, current non-quantization methods at high compression ratios inevitably experience some performance loss (for example, SliceGPT and LLM-Pruner reduce performance on commonsense reasoning by 25.4% and 17.2%, respectively, when applied to the Llama2-7b model with a 20% pruning ratio). **However, this does not mean they are without value. On the contrary, the field of non-quantization LLM compression has been continuously developing and has attracted increasing attention from researchers, including many esteemed scholars \[3\]\[4\]\[5\].** Our statistics show that in 2022, 2023, and 2024, there were 37, 54, and 93 papers published in ICLR, NeurIPS, ICML, and CVPR with "model pruning" in their titles, respectively. This rapidly increasing trend indicates that non-quantization compression methods are gaining recognition and receiving more attention as the scale of LLMs grows. Therefore, we believe **it is unreasonable for the reviewer to dismiss the practical significance of non-quantization compression methods solely due to performance loss.** Such a viewpoint actually negates the substantial efforts of researchers in the field of non-quantization compression, which is clearly incorrect.
>
>    **Second, Dobi-SVD has already surpassed existing pruning and SVD methods in performance.** Although the reviewer considers an 8.7% accuracy reduction for DobiSVD at a compression ratio of 0.6 on Llama 3.1-8B to be unacceptable, we would like to point out that, without employing fine-tuning, this loss is already smaller than that of other SVD/pruning methods at the same compression ratio. For instance, other pruning methods, including SliceGPT, reduce accuracy by between 33.3% and 43.5% (as shown in Table 15). Furthermore, as demonstrated in Tables 2 and 3 of the main text and in subsequent comparisons on other models, DobiSVD and all SVD-based methods achieve the best current results compared to popular pruning methods across different models. Therefore, from an accuracy perspective, we believe that DobiSVD delivers reasonable and competitive results. **Combining these points, we consider the performance of DobiSVD to be both instructive and practically significant.**
>
>    **Additionally, we would like to remind the reviewer that current methods claiming to achieve "lossless" compression rely on post-training techniques**, such as LoRA fine-tuning, to recover the performance losses caused by compression. For example, ShearedLLAMA requires extensive training on 50 billion tokens after compressing LLaMA2-7B to Sheared-LLaMA-2.7B. These training processes consume substantial time and computational resources. Therefore, while we believe that combining our method with fine-tuning techniques like LoRA could further enhance performance, our ultimate goal is to achieve model compression at a low cost.
>
> 3. ***We believe there are misunderstandings regarding our experiments and statements about VLM.***
>
>    **Firstly, our experiments on VLM did not claim to *"improve performance"* as our on-the-top assertion.** In fact, our exact wording in line 157 of the paper is: "we find that our Dobi-SVD not only enhances efficiency but also improves VLM performance on the Pope-Random dataset," and in General Response (Part 3/5), we describe: "for example, even at a 0.4 compression rate, it achieved nearly zero loss on the Pope dataset," which are consistent with the experimental results.
>
>    **Secondly, our outstanding experimental performance is consistently stable and not the result of the reviewer uXRu’s statement “*random variability in the evaluation*”.**  During the VLM testing process, we evaluated all data within the datasets using the standard testing procedures of LLaVA-v1.5. Furthermore, to ensure the fairness of our experimental results, we conducted 10 tests on TextQA with a random seed using a 0.4 compression ratio on LLaVA-v1.5. The maximum performance variance observed was 0.2, which validates the fairness and reliability of our testing.
>
>    **Lastly, we would like to reiterate the significance of Dobi-SVD's applicability to VLM.** The addition of version modules and alignment modules in VLM means that not all compression methods can be directly transferred and remain effective. Moreover, VLMs are typically larger in size compared to LLMs and require more compression, with broader applications such as in robotics. The applicability of Dobi-SVD to VLM demonstrates its extensive potential and versatile application scenarios in real-world settings.
>
>    We would like to emphasize that we have dedicated significant time and effort to develop Dobi-SVD in order to address the practical needs we face. **Although the VLM experiment was not conducted at the request of any reviewer, but rather as part of our own research goals.**

---

> ### Author Response · Authors · 2024-12-02
> **Clarifications and Explanations Regarding Some Misunderstandings (Part 3/3)**
>
> 4. ***We think that using standardized accuracy alone to evaluate model effectiveness is inappropriate.***
>
>    We politely disagree with *"In order to lead to significant speedups, your method drops significant accuracy."* Actually, we believe the observed accuracy drop is relative to other SVD and pruning methods. As shown in Tables 2 and 3 of our manuscript, Dobi-SVD achieves better performance compared to existing SVD compression and pruning techniques. This result is both exciting and meaningful within the SVD model compression domain, effectively advancing the development of SVD-based model compression.
>
>    **While we aim to pursue lower compression losses in future work, we consider our current experimental results to be sufficiently robust and promising.** Additionally, we do not agree with the claim that *"standardized accuracy being zero on harder tasks implies the model is unusable."* This indicates that these tasks are inherently challenging for this model, rather than suggesting our method makes it unusable. Furthermore, test sets like MathQA are inherently difficult for existing non-compressed LLMs, and pursuing usability on such benchmarks for model compression methods may be impractical. Instead, we believe that the drop relative to the original model is the appropriate measure of a compression method's effectiveness, as used by a wide range of works including ShearedLLAMA and SliceGPT that the reviewer mentioned. We believe **the necessity of this metric itself is open to question**.
>
> Once again, we thank the reviewer for taking the time and effort to review our revision and sharing concerns. We hope that our clarifications and explanations can eliminate some misunderstandings. The two novel perspectives we proposed, *'A Novel Path from Activation to Weight'* and *'Fully Unlocking SVD’s Potential for Data Compression by Addressing a Long-Overlooked Limitation'*, have already taken a great step in a promising direction, and as the reviewer anticipated, achieving more effective and low-cost model compression is indeed our next goal.
>
>
> References:
>
> [1]Dettmers, Tim, et al. "Qlora: Efficient finetuning of quantized llms." *Advances in Neural Information Processing Systems* 36 (2024).
>
> [2] Frantar, Elias, et al. "Marlin: Mixed-precision auto-regressive parallel inference on large language models." arXiv preprint arXiv:2408.11743 (2024).
>
> [3] Han, Song, et al. "Retrospective: EIE: Efficient Inference Engine on Sparse and Compressed Neural Network." *arXiv preprint arXiv:2306.09552* (2023).
>
> [4] Xia, Mengzhou, Zexuan Zhong, and Danqi Chen. "Structured pruning learns compact and accurate models." *arXiv preprint arXiv:2204.00408* (2022).
>
> [5]Kwon, W., Kim, S., Mahoney, M. W., Hassoun, J., Keutzer, K., & Gholami, A. (2022). A fast post-training pruning framework for transformers. *Advances in Neural Information Processing Systems*, *35*, 24101-24116.

---

> ### Comment · Reviewer_uXRu · 2024-12-03
> **Further feedback**
>
> Dear authors,
>
> Thank you again for your response. Overall, I would like to note that the response simply re-iterates points that you have already made either in your initial response or in responses to other reviews.
>
> From my perspective, there is a very simple experiment that would clarify things, which should also be doable within a week, if your claims of simplicity and scalability for your method hold:
>
> * Let's take the Llama-2 family of models, with the 7B and 13B variants. Please compress the 13B variant to a level that would make it comparable to Llama-2-7B in terms of FLOPs or runtime (you get extra points for the latter metric, but the first is fine).
>
> * Now, please compare the accuracies (PPL and standard zero-shots) between the two models. Is the Dobi-SVD 13B-shrunk-to-7B model better? Bonus question: how do the accuracies compare to 4bit quantization of the 13B model using standard techniques (BNB, AutoGPTQ, AutoAWQ)?
>
> If the above comparison would show that your method is Pareto-competitive, then I will consider the point addressed and increase my score. Notice that this is a basic test of validity for the method, as it reflects how a user would choose which model to use.
>
> (You may use other models for this comparison, such as Qwen, if more convenient.)
>
> I will briefly comment on some of the responses, since they are either taking my statements out of context or based on misunderstandings by the authors:
>
> >  the reviewer’s accusation of our quantization comparison being deceptive is unfounded.
>
> All I was saying was that you conveniently omitted the baseline in the table you shared with me. Viewing results in context suggested that your method leads to catastrophic accuracy loss for those benchmarks, relative to the baseline.
>
> > comments regarding performance are unreasonable. First, non-quantization compression methods are not unuseful.
>
> This is simply not what I said. Non-quantization methods can be useful, but -- as your very own results show -- they often seem require fine-tuning to recover non-random levels of accuracy on standard tasks. Your method claims to do this without fine-tuning: I just pointed out that the evals suggest this might not be the case.
> Please note that, as I said before, ShearedLlama also released a pruned checkpoint _before_ fine-tuning. My suggestion was to compare with that one.
>
> > standardized accuracy alone to evaluate model effectiveness is inappropriate.
>
> Again, I did not suggest using standardized accuracy _alone_. However, looking at standardized accuracy suggests that the models you produce have _a lot_ of accuracy drop, to the point where they are producing random responses. Since we do not have the option of interacting with the model or conducting user testing, we have to use such tests to validate the models.
>
> **Finally, a note on standards, since the authors seem to be questioning mine.** While the submitted paper has value (reflected in my increased evaluation of 6), it is clearly not the first in the area of SVD compression. Therefore, the standards for it _are inherently higher_ than for earlier work. In my view, it is not sufficient for the current work to be simply slightly improving over prior work on SVD in terms of abstract metrics--otherwise, we would just get an endless stream of incremental papers. To deserve publication into a top-tier venue such as ICLR, the work has to do so _significantly_, in the sense that it would make SVD a viable competitor to e.g. quantization. (The authors seem to agree on this as well, since they chose this as a comparison point.) The standard I set is therefore Pareto-competitiveness: the method should be competitive with other approaches leading to the same level of speedup. If the authors can meet this, then their work should be published. Otherwise, they may try to improve improve their results and try again.
>
> Best regards!

---

> ### Author Response · Authors · 2024-12-04
> **Further Response to Reviewer uXRu (Part 1/2)**
>
> We appreciate reviewer uXRu's experimental suggestions and additional explanations, which included workloads described as "which should also be doable within a week." However, these suggestions were provided on the final day of the review process, rather than at the beginning. **Despite as stated in the review policy email sent by ICLR PC to reviewers on 11/26, “No requests for new experiments are allowed”**, **we tried our best to accommodate the reviewer's request by conducting the proposed experiments.** The results are presented in the table below.
>
>
>
> | Model                | Parameter(B) | Throughput | (tokens/s)        | Performance  | (accuracy) |                  |          |          |          |          |
> | -------------------- | ------------- | --------------------- | ------ | ---------------------- | -------- | -------- | -------- | -------- | -------- | -------- |
> |                      |               | bz=1                  | bz=8   | ARC_e                  | Arc_c    | Openb    | WinoG    | PIQA     | Mathqa   | Avg      |
> | Llama-7b (original)  | 7.0           | 42.29                 | 319.49 | 0.67                   | 0.38     | 0.28     | 0.67     | 0.78     | 0.27     | 0.51     |
> | Llama-13b (Dobi-0.6) | 7.8           | 38.56                 | 283.18 | **0.72**               | **0.40** | **0.32** | **0.72** | **0.78** | **0.29** | **0.54** |
>
>
>
>
>
> | Model                 | Parameter(B) | Throughput| (tokens/s)         | Performance|  (accuracy)          |          |          |          |          |          |
> | --------------------- | ------------- | --------------------- | ------ | ---------------------- | -------- | -------- | -------- | -------- | -------- | -------- |
> |                       |               | bz=1                  | bz=8   | ARC_e                  | Arc_c    | Openb    | WinoG    | PIQA     | Mathqa   | Avg      |
> | OPT-2.7b (original)   | 2.7           | 39.13                 | 279.73 | 0.56                   | 0.26     | 0.20     | **0.58** | 0.68     | 0.22     | 0.42     |
> | Llama-2-7b (Dobi-0.4) | 2.8           | 47.97                 | 368.37 | **0.58**               | **0.39** | **0.22** | 0.57     | **0.68** | **0.24** | **0.45** |
>
>
>
> Experiments show that it is evident that whether comparing within the LLaMA series or against other model series, models compressed with Dobi-SVD show higher accuracy with competitive or even higher throughput. This suggests the practical value for real applications.

---

> > ### Author Response · Authors · 2024-12-04
> > **Further Response to Reviewer uXRu (Part 2/2)**
> >
> > We do not have sufficient time to conduct the requested quantization comparison with only one day remaining. Moreover, **we firmly believe that using surpassing quantization as a standard for paper acceptance is entirely misplaced.** We respectfully disagree with the reviewer’s out-of-scope standard and the statement: *“To deserve publication in a top-tier venue such as ICLR, the work has to do so significantly, in the sense that it would make SVD a viable competitor to e.g., quantization.”* As highlighted in our previous response, *“it is unreasonable for the reviewer to dismiss the practical significance of non-quantization compression methods solely due to performance loss.”* Unfortunately, the reviewer’s comments continue to reflect this dismissive stance, which we find unwarranted and contrary to the broader goals of fostering impactful and diverse research contributions. We have to point out that the dismissal from the reviewer lacks fairness. The reviewer seems to solely highlight the performance drop of our method relative to the original PPL or quantization-based methods, while ignoring comparisons with other methods and other analysis experiments.
> >
> > So let’s highlight it again: SVD-based or pruning-based methods were never intended to compete with post-fine-tuned quantization in terms of accuracy and PPL. Instead, these methods address entirely different challenges, such as overcoming device-requirement limitations, and have a long track record of producing impactful research. **Our results of Dobi-SVD clearly demonstrate significant advancements, outperforming existing SVD-based methods and popular pruning techniques, supported by a well-documented method and novel design.**
> >
> > The reviewer has applied out-of-scope standards that overlook our contributions to this direction. In line with [ICLR’s acknowledged publication standards](https://x.com/iclr_conf), the decision to accept a paper should address two fundamental questions: *“Does this paper take a gradient step in a promising direction?”* **Our work surely takes a significant step forward in advancing non-quantization-based methods!** The second question is: *“Is the community better off with this paper published?”* **Without question, our method and the perspectives we propose will inspire future research in non-quantization-based approaches. Moreover, our publicly released code will extend the benefits of our contributions beyond SVD-based LLM compression to the broader field. In this context, we firmly believe our work deserves publication at ICLR!**

---

> ### Author Response · Authors · 2024-12-04
> **Plus, Dobi-SVD is far more "slightly improving over prior work on SVD"**
>
> Some facts that the reviewer could easily verify:
>
> 1. Table 2: When the ratio is 0.8, Dobi-SVD outperforms the previous best method, SVD-LLM, by reducing PPL by 23.4%. At a ratio of 0.6, it reduces PPL by 38.1%, and at 0.4, it reduces PPL by 81.5%. **Note that Dobi-SVD does not use fine-tuning, while SVD-LLM does.**
> 2. General Response (Part 4/5): On the Llama3-8B 0.8 MMLU, the original model achieved **63.3**, LLRC achieved 24.8, while Dobi-SVD achieved **60.1**.
> 3. Among all quantization, pruning, and SVD-based methods, **Dobi-SVD is** **the first** **to demonstrate compression for both LLM and VLM**, with performance that is noteworthy. For example, even at a compression rate of 0.4, it achieved nearly zero loss on the Pope dataset.
> 4. For the past 34 years, since 1990 [1], Dobi-SVD is **the first work** to provide an effective solution to the long-overlooked limitation of SVD for compression.
>
> [1] Demmel, J. and Kahan, W. (1990). Accurate Singular Values of Bidiagonal Matrices. SIAM J. Sci. Statist. Comput., 11 (5), 873-912.

---

### Official Review · Reviewer_otMy · 2024-11-03

**Soundness:** 2
**Presentation:** 4
**Contribution:** 2
**Rating:** 5
**Confidence:** 5

**Summary:**

This paper mainly give a discussion about the three about low rank factorization  in LLM as following:
1. $\min rank(\hat{W})$, $s.t., $ the performance gap between $model_{LLM}(W)$and $model_{LLM}(\hat{W}) $ satisfies the requirements.
2. How to update the weight based on the the compressed activation.
3. Adjust the math algorithm into practice area.

I think the core idea in this paper is gained from the way to let 0-1 function (activation function) into sigmod function to let neural network have a gradient. They also build a multi-objective function to balance the 2 target.

The authors also give the analysis and operation details based on above main idea.

**Strengths:**

1.I personally like the style of paper writing. Compared with traditionaly paper structure, this paper structure shows a clear structure of how authors analysis problems and how to build the algorithms.

2.There is a few work which uses  traditional low-rank decomposition to analysis and design algorithm to compress the model. This work gives a good method to compress model.

3. The optimzation of compress method is the model performance instead of traditional $\|W-\hat{W}\|$

**Weaknesses:**

This article faces threee critical issues:

1. In Chapter 2, we believe that the author's choice of activation as the target for decomposition is incorrect. Specifically, the reasons are as follows: in Eq. 3, we know that $\|A-A_k|\ <\|A-xW_k\|$.

But $\Delta L_w = grad_w*（W-W_k) = grad_A*(A-xW_k) = \|grad_A\|\|A-xW_k\| cos<grad_A,A-xW_k>$ and  $\Delta L_A = grad_A*(A-A_k) =\|grad_A\|\|(A-A_k)\|cost<grad_A,A-A_k>$.  Then $\Delta L_w - \Delta L_A = \|grad_A\|(\|(A-A_k)\|cos<grad_A,A-A_k> -\|A-xW_k\| cos<grad_A,A-xW_k>)$. We cannot make sure about the relationship between $\Delta L_w$ and $\Delta L_A$ with the relationship $\|A-A_k|\ $ and $\|A-xW_k\|$.

What is more, I think the Eq.1 to Eq.3 have typos, $\frac{\partial L}{\partial A}$ should be $\frac{\partial L}{\partial W}$ in $\Delta L_w$ equation.

2. In the methods for solving gradient explosion, the author lacks analysis in approximating the gradients. If the author considers this point important and unique, I think they should conduct an analysis on it, discussing the impact of this approach on convergence results. In fact, I believe that these methods are commonly used to solve gradient explosion, and the author does not need to further describe and discuss them in the article. Instead, they could simply cite some papers on common solutions to gradient explosion.

3. What is long overlooked limitation in paper? I know in low rank method, the half of the singular values need to be truncated and the degradation of model is rooted in the this kind of method. But how to overcome this problem in your method in section 3.3? I think I am lost in section 3.3.

**Questions:**

1.Please correct the analysis part in Eq.1 to Eq.3.

2. Compress the content between line 254 to line 295, or show why your method is novel or important.

3. Give more details about why the overlooked limitation is so important and how to overcome it and analysis your algorithm. In current version, I cannot understand why your work can overcome this limitation.

---

> ### Author Response · Authors · 2024-11-24
> **Response to Reviewer otMy**
>
> We would like to thank the reviewer for taking the time to review our paper and provide valuable feedback. We are glad that the reviewer liked the writing style of this paper and thought it presented a good approach to model compression. We are excited to have the chance to address the reviewer’s questions and concerns. These edits will make the paper stronger.
>
> We respectfully point out that reviewer otMy has significantly underestimated the novelty and contributions of our work. Contrary to the summary, “I think the core idea in this paper is gained from the way to let the 0-1 function (activation function) become a sigmoid function to provide a gradient for the neural network. They also build a multi-objective function to balance two targets,” our work goes far beyond these summarized points.
>
> Specifically, our contributions can be summarized as follows:
>
> 1. Proving the optimality of directly truncating activations.
> 2. Developing an end-to-end differentiable approach to determine the optimal truncation positions.
> 3. Reconstructing weights from the truncated activations.
> 4. Addressing the limitations of truncation boundaries through quantization.
>
> We would like to emphasize that this represents an entirely new paradigm. The uniqueness of our paradigm lies at four aspects:
>
> 1. We are the first to establish the theoretical optimality of directly truncating activations, which fundamentally differs from previous activation-aware methods that merely treat activation distances as an optimization target.
>
> 2. We are the first to enable an end-to-end differentiable approach for determining the optimal truncation position, significantly reducing the engineering complexity associated with SVD-based compression methods
>
> 3. While direct activation truncation is theoretically optimal, reconstructing weights from truncated activations is challenging. We are the first to demonstrate how to effectively and efficiently achieve this reconstruction.
>
> 4. We introduce a novel application of quantization to overcome the under-mapping issue of truncation rank and memory storage constraints.
>
>
>
> From the high-level conceptual paradigm to the low-level technical implementation, we believe our work makes significant contributions that go beyond the points highlighted by reviewer otMy.
>
> We then answer the remaining concerns from the reviewer otMy below:
>
> 1. Gradient explosion with Taylor expansion.
>
>    **We kindly disagree that the method is commonly used to solve gradient explosion.** Indeed, the gradient explosion issue has been widely explored in previous literature. However, using Taylor expansion to address the gradient explosion is not common, especially in the context of SVD. As far as we know, there is only one work [1] utilizing the Taylor expansion to resolve this issue. We have discussed and cited this work in the revised version. While their focus was on symmetric matrices and computer vision tasks, we made a novel use of Taylor expansion for general matrices and LLM compression. Unlike [1], which dealt with low-dimensional image features, our work addresses the challenges of calculating large-dimensional matrices in LLMs. To meet these demands of large matrices in LLM, we developed a more efficient, parallelized algorithm. **The algorithm is detailed in the appendix A.6 “Dobi-SVD's Robust and Efficient Differentiable SVD Algorithm for General Matrices”.** We will release the full code, and we believe our engineering contributions in this direction can benefit a broader community interested in SVD-related optimization techniques.
>
>
>
> 2. More illustration of overcoming truncation limitations.
>    We are pleased to see that reviewer otMy understands our motivation for this part, as reflected in the statement:“Low-rank methods require truncating half of the singular values, and the model degradation is rooted in this type of approach.” This perspective can be viewed from another angle: when aiming for a compression ratio of $<1$ while keeping $k \in [0, \min(m, n))$, it is evident that the storage space required is $k \cdot \max(m, n)$. However, the practical memory footprint is $k \cdot (m + n)$. To bridge the gap and remap the storage from $k \cdot (m + n)$ to $k \cdot \max(m, n)$, we make a novel use of the Gaussian distribution properties of the $U$ and $V$ matrices and implement a quantization method to reduce the footprint requirements.
>
> 3. Typo (about first question).
>    We sincerely appreciate your derivation and verification. It is a typo error in our notation, also noted by the reviewer sX49 and bQbW. We have revised it in our new version.
>
>
>
> Reference:
> [1] Robust Differentiable SVD (TPAMI 2021)

---

> > ### Comment · Reviewer_otMy · 2024-11-24
> > **please revise the eq2 in new version**
> >
> > Again, in new version, the eq.2 to eq.3 is wrong as I mentioned in review. The inner production is not equal to the production of two vector's norm.  See the analysis as following.
> >
> >  $\Delta L_w = grad_w*（W-W_k) = grad_A*(A-xW_k) = |grad_A||A-xW_k| cos<grad_A,A-xW_k>$ and $\Delta L_A = grad_A*(A-A_k) =|grad_A||(A-A_k)|cost<grad_A,A-A_k>$. Then $\Delta L_w - \Delta L_A = |grad_A|(|(A-A_k)|cos<grad_A,A-A_k> -|A-xW_k| cos<grad_A,A-xW_k>)$. We cannot make sure about the relationship between $\Delta L_w$ and $\Delta L_A$ with the relationship $|A-A_k|\ $ and $|A-xW_k|$.

---

> ### Author Response · Authors · 2024-11-25
> **A Comprehensive and Accurate Discussion on Formulations**
>
> We greatly appreciate the reviewer pointing out this issue. We sincerely apologize for not recognizing this problem during the initial rebuttal.
>
> In the latest revised version of the paper, we analyze and demonstrate the superiority of directly truncating activations over truncated weights from both a module-level and model-level perspective. **We highlight that our conclusion of the superiority of directly truncating activations over truncated weights still holds!**
>
> - At the module level, the Eckart-Young-Mirsky Theorem guarantees that the directly truncating activation gives $A_k$, the optimal $k$-rank approximation of activation.
>
> - **At the model level, we model the training loss and account for the reviewer's suggestion to fully consider the impact of θ.** To achieve this, we directly compare the complete formulas:
>
>   $$\Delta \mathcal{L}_W = \frac{\partial \mathcal{L}}{\partial A} (A - xW_k) \quad \text{and} \quad \Delta \mathcal{L}_A = \frac{\partial \mathcal{L}}{\partial A} (A - A_k).$$
>
>   - **Experimental setup:** We truncate weights and activations with k = \{100, 500, 1500, 2000, 2500, 3000\} across layers \{5, 10, 15, 20, 25\} on LLaMA2-7B and evaluate the model's performance loss on 256 samples from WikiText2.
>   - **Results:** The results show that across different layers and k-values, the performance loss caused by $A_k$ is consistently smaller than that caused by $W_k$.
>
> By analyzing and validating from both module and model perspectives, we demonstrate the superiority of directly truncating activations. Notably, this is a more comprehensive comparison, as previous works only analyzed the effects of processing activations at the module level (e.g., xW==A), without considering the model level. **Due to space limitations, the detailed analysis and experimental results are presented in Appendix A.10.**
>
> **Previously, this issue had not been explored in SVD-based methods. We are very grateful to the reviewer for raising it, which allowed us to improve our paper and make it more rigorous.** This forms one of the foundations of Dobi-SVD and combined with various theoretical analyses, we propose two novel and interesting perspectives: *A Novel Path from Activation to Weight* and *Fully Unlocking SVD’s Potential for Data Compression by Addressing a Long-Overlooked Limitation.* If the reviewer has further questions about other theoretical analyses, we warmly welcome the feedback.

---

> > ### Author Response · Authors · 2024-11-28
> >
> > We greatly appreciate the reviewer's detailed feedback and thoughtful suggestions. Over the past two weeks, we have added a significant amount of experimental and theoretical explanations. We truly appreciate the reviewer’s clarification regarding our misunderstanding of the first issue, which provided us with the opportunity to correct this oversight. **We are pleased to inform the reviewer that we have now thoroughly addressed all the concerns raised**:
> >
> > 1. **Regarding the formula**: We have corrected the typo and fully considered the impact of θ, as mentioned by the reviewer, enhancing our theoretical derivation. The related content can be found in A.10, *Analysis of Directly Truncating Activations over Weights*.
> > 2. **Regarding the Taylor expansion to address gradient explosion**: This is not a commonly used method, and we found only one related paper. We have provided a more detailed explanation and necessary discussions in A.6, *Dobi-SVD’s Robust and Efficient Differentiable SVD Algorithm for General Matrices*.
> > 3. **Regarding the long-overlooked limitation, its importance, and our solution**: We have provided a more detailed explanation in A.5, *Dobi-SVD’s New Perspective 2: Fully Unlocking SVD’s Potential for Data Compression by Addressing a Long-Overlooked Limitation*.
> >
> > Thanks to the invaluable suggestions and support from the reviewers, we have refined Dobi-SVD, especially the two novel perspectives: '*A Novel Path from Activation to Weight'* and *'Fully Unlocking SVD’s Potential for Data Compression by Addressing a Long-Overlooked Limitation.'* We believe that the **current version of Dobi-SVD is backed by solid theory, presents novel insights, and offers significant contributions, deserving of a higher score.** Specifically, we highlight the following key points:
> >
> > - **Innovative Approach**: Dobi-SVD does not follow any existing quantization, pruning, or SVD-based methods but instead introduces a new way of handling activations and weights based on the fundamental theory of SVD (the EYM theorem).
> > - **Efficient and Robust SVD Backpropagation**: We have developed an efficient and robust method for general matrix SVD backpropagation, which serves as a direct reference for others exploring low-rank, high-dimensional, and multi-form SVD matrices.
> > - **Addressing a Long-Overlooked Yet Critical Limitation**: By leveraging the Gaussian distribution of singular value vectors, we’ve remapped singular value information to storage space, overcoming a longstanding limitation overlooked by SVD-based methods and criticized by non-SVD approaches, thus unlocking SVD’s full compression potential.
> > - **Comprehensive Comparisons**: We have provided a broad comparison across various model compression methods, demonstrating superior practical performance.
> >   - Not only do we outperform existing SVD-based methods (General Response Part 4), but we also generally outperform a wide range of pruning methods(General Response Part 2).
> >   - In the comparison related to quantization (General Response Part 3), we demonstrate that our approach reduces FLOPs and highlight that a more effective integration of SVD-based methods and quantization is a promising direction for future exploration.
> > - **Generalizability**: Dobi-SVD’s successful application to Vision-Language Models (VLMs) highlights its broad applicability, extending beyond LLMs to areas like robotics. To the best of our knowledge, this is the first demonstration of such generalizability.
> >
> > **In light of these updates, we kindly request that the reviewer reconsider the overall score for our paper. We believe that the revisions we’ve made, along with the additional explanations and insights, warrant a higher score and a more favorable evaluation.**
> >
> > Should the reviewer have any further questions or suggestions, we would be grateful for the opportunity to provide additional clarification and will address any further concerns promptly.

---

> > > ### Comment · Reviewer_otMy · 2024-12-02
> > >
> > > I still cannot understand the section3.3. Why we have to make the bijection between compress rate and truncation positions? From the final paragraph, what we want to do is reducing the storage memory. I think you can directly show your compress way instead of showing the bijection.

---

> > > > ### Author Response · Authors · 2024-12-03
> > > > **Further Response to Reviewer otMy**
> > > >
> > > > We greatly appreciate the reviewer’s willingness to provide further concerns, and we are happy to offer clarifications and explanations:
> > > >
> > > > **The main goal of SVD-based LLM compression methods is to find the truncation value $k$ that optimizes model performance under the same memory storage constraints.** In previous methods, the mapping between the truncation value and the compression ratio is given by $k \rightarrow \frac{k \times (m + n)}{m \times n}$, which means that to achieve effective compression, the search range of $k$ is limited to $k \in \left(0, \frac{m \times n}{m + n}\right)$. This indicates an **injective mapping** between memory and the truncation value, implying that during the optimization of finding the truncation rank, many truncation values are never considered, regardless of the compression ratio. **We believe that this limitation in the search range of $k$ significantly restricts the model's performance.**
> > > >
> > > > **Therefore, we aim to expand the range of $k$ under the same memory storage constraints to enhance model performance. This is the motivation of why we propose the concept of bijective mapping.** It seems to achieve this as simple as removing the bound of $k \rightarrow \frac{k \times (m + n)}{m \times n}$ as SVD-LLM and ASVD used during the optimization. However, this would result in the combination of compression loss and performance loss hard to converge. We go a novel way: we exploit the quantization-friendly properties of the left and right singular matrices to store the decomposed matrices in mixed precision. In our method, the mapping between the truncation value and the compression ratio is $k \rightarrow \frac{k}{\min(m, n)}$, and the search range of $k$ becomes $k \in \left(0, \min(m, n)\right)$. This bijective mapping allows the differentiable optimization of finding truncation position to analyze all singular value information. Therefore, during the optimization, under the same memory storage constraints, the bijection makes us find a more suitable truncation value from the larger search space so as to make the model perform better.
> > > >
> > > > We believe that enlarging the optimization search space while keeping the memory storage is the main spirit of the concept of bijective mapping, much beyond merely mentioning the compression way.
> > > >
> > > > We thank the reviewer for raising these insightful questions and providing meaningful suggestions. In the final version of our paper, we will include a more intuitive description of our compression method in this section. We hope that our response adequately addresses your concerns.

---

> > > > > ### Comment · Reviewer_otMy · 2024-12-03
> > > > > **overpackaged parts**
> > > > >
> > > > > After reading the rebuttal, I know that what the author did in this section was to quantify the UV matrix to free up memory and make the UV in the SVD process have more rank.
> > > > >
> > > > > Firstly, this section is completely over-packaged, and most of the descriptions are meaningless. In this section, the presentation of concepts like bijection is misleading and can cause readers to lose focus. It is suggested that the author simplify the description in this part and directly use the expression "using quantization methods to reduce the storage space of the UV matrix, which can reduce the low-rank requirements of SVD  while using the same storage space". The current version completely over-packs this part, making it difficult for readers to understand what the author is trying to do from the main text of the paper.
> > > > >
> > > > > Secondly, as other reviewers have mentioned, the method of combining low-rank decomposition and quantization has applications in different methods. I think it is worth evaluating current methods and forming a paper on this part separately. That is, by combining low-rank decomposition and quantization, we can further improve efficiency while ensuring model accuracy and compare with state-of-the-art methods.
> > > > >
> > > > > As for the analysis, the authors still say that they have rigorously proven that decomposing activations is the optimal choice. The analysis is provided in the appendix, which is vague and cannot prove the results. The author is requested to remove this section from the paper, as well as the related content of the EYM theorem, as it is not used at all.
> > > > >
> > > > > The above problems are not only problems in writing papers and adding experiments, but also problems in the entire method, which is the issue of soundness. In this regard, I think this article should be rejected and split into two articles for resubmission. But considering the workload of this article, I will keep my weak rejection score here.

---

> > > > > > ### Author Response · Authors · 2024-12-03
> > > > > > **Please, comment based on accurate facts and detailed concerns. (Part 1/2)**
> > > > > >
> > > > > > 1. We respectfully disagree with the reviewer’s comment that our discussion of the “long-overlooked limitation” is “overpackaged.” **We believe that framing and defining the problem is even more important than offering a solution.**
> > > > > >
> > > > > >    **Regarding  the problem:** While the reviewer suggested that it would be sufficient to “directly show the compression method,” we would like to highlight that the “long-overlooked limitation” of SVD-based compression is an algebraic *injection* relationship between compression ratio and truncation positions. To overcome this limitation and fully analyze all singular value information, we must perform a “remapping” that transforms the *injection* relationship into a *bijection*. **Clearly, introducing this** **bijection** **is a natural and essential development in our approach.** **Moreover, we present this idea concisely in just 10 lines (L341–350) in the main body!** We are unsure what the reviewer’s standard for “overpackaged” is, and we do not understand why we should not go into greater detail when framing and defining the problem.
> > > > > >
> > > > > >    **Regarding the solution:** We are the first to approach the mathematical limits of SVD compression from a computer science perspective, to propose a solution for remapping, and to demonstrate its effectiveness experimentally. **However, we hope there could be future, potentially better solutions and our approach is one of many possible solutions. That’s why we need to emphasize bijection.**
> > > > > >
> > > > > > 2. The other reviewers' mention of "*combining low-rank decomposition and quantization*" stems from their recognition of Dobi-SVD's state-of-the-art performance compared to existing SVD-based methods and popular pruning techniques, as well as its unique advantages over quantization. It is precisely because Dobi-SVD provides detailed reasoning and outstanding experimental results that they see the potential and promise of such a combination. **This is a direction for future work and our next goal, and it is unrelated to the reviewer's concern about the "overpackaged part."**
> > > > > >
> > > > > > 3. We will not remove the EYM theory, nor will we remove the theoretical analysis presented in the appendix.
> > > > > >
> > > > > >    1. **The foundation of SVD is rooted in the EYM theory,** and if we are utilizing SVD, we must reference EYM theory as a sign of respect for this foundational work. **Moreover, we have used the EYM theory to substantiate our argument for the optimal solution (L207-208, L1528-1535)**. Without the support of the EYM theory, we would not be able to present the proof for optimality.
> > > > > >    2. In response to the reviewer's suggestion, we have considered the influence of θ and have added a series of experiments and a full page of analysis to address this. **We would appreciate it if the reviewer could specify which parts they find "vague and unable to prove the results," and provide further clarification on why these sections are unclear.**
> > > > > >
> > > > > > 4. Regarding the statement: *"The above problems are not only problems in writing papers and adding experiments, but also problems in the entire method, which is the issue of soundness,"* we can see the concise style that the reviewer pursues: this sentence is a sentence, and other than being a sentence, we cannot get more than one sentence of information! **If the reviewer is dissatisfied with our work, we kindly request that the reviewer provides more detailed feedback on the specific issues.**
> > > > > >
> > > > > >    1. **Problems in Writing Style**: We acknowledge the reviewer's concerns regarding the writing style and sincerely respect their feedback. While it is impossible to craft a paper that satisfies everyone’s expectations, we are committed to continuously improving the clarity and flow of our writing.
> > > > > >    2. **Problems in Adding Experiments**: Many of the experiments were added based on other reviewers' suggestions, while others, such as those related to VLM, were driven by our own practical needs. One experiment we are particularly proud of is A.7.4 Truncation Sensitivity Analysis, which directly addresses reviewer *sX49’s* thoughtful and **precise** concern: *"Whether one requires a stricter and more rigorous approach to the selection of the truncation position?"* **This led to the insightful question**: *"How sensitive is the task performance to the choice of (k)?"*—**which inspired us to design a new, interesting experiment.**
> > > > > >    3. **Problems in the Entire Method**: **What** **specific** problems are being referenced? **Why are they issues?** If the reviewer is willing to provide **more detailed concerns**, we are willing to address them and resolve any remaining questions.

---

> > > > > > > ### Author Response · Authors · 2024-12-03
> > > > > > > **Please, comment based on accurate facts and detailed concerns. (Part 2/2)**
> > > > > > >
> > > > > > > Additionally, regarding the last two sentences from the reviewer: “*In this regard, I think this article should be rejected and split into two articles for resubmission. But considering the workload of this article, I will keep my weak rejection score here.*”：
> > > > > > >
> > > > > > > **Dear Reviewer otmy,** we have always respected your authority in evaluating our work, and we sincerely appreciate your efforts in reviewing our submission. However, **DON’T LET THIS SUBMISSION BE A JOKE, PLEASE**: While you acknowledged our workload, you suggested that we should split the paper into two separate submissions to increase our chances of acceptance. **So, if we are rejected, will it be because all the "overpackaged parts" you are concerned about were not "overpackaged" enough?**
> > > > > > >
> > > > > > > We welcome all constructive criticism from reviewers, as we firmly believe that our work can withstand scrutiny. Furthermore, we believe that **critiques based on accurate facts and detailed concerns** will ultimately help us improve the quality and clarity of our research.

---

> ### Comment · Reviewer_otMy · 2024-12-03
>
> First of all, over-packaging or over-claiming is the most important factor that causes confusion in the part 3.3, i.e., part of quantizing UV matrix. I need to spend 4-6 hours in all to figure out what is a LONG-OVERLOOKED TRUNCATION LIMITATION problem, because in my opinion, rank < mn/(m+n) is a precondition that should be taken for granted in low-rank decomposition, especially in real scenarios. This will definitely be considered. When reading the article, the introduction of quantization operation in this paragraph is not connected.
>
> So I suggest that the author just say that the UV matrix is ​​quantized in title in this part. The speed is improved by quantization. Forcibly introducing concepts such as remapping and bijection will cause confusion to readers, so this paragraph is over-packed. As I said, the title of this paragraph can directly say that the performance is further improved by combining quantization technology.
>
> Further,  the problem whether the accuracy loss caused by quantizing UV matrix can be compensated by the increase of rank,  what is the upper bound  can be further discussed and are innovative. Because in industrial scenarios, we need the trade off the  rank, model accuracy, model speed and quantization degree. This is a topic worth discussing. Most of the current quantization work is based on direct quantization weights rather than UV matrices.
>
> This topic is not expanded in this paper, and it is difficult for me to see these contents in the end-to-end experiments. I also think this point will be a very innovating and be worth 6-9 pages paper. This point is more innovative than learning the truncation position. But the end-to-end experiments cover up all these contents.
>
> Therefore, this paper actually hides a content that has not been fully discussed, so I tend to reject it. Although a large number of end-to-end experiments cannot fully show the content that needs to be discussed, some traces can be seen. So here I will provide my opinions to AC, who will judge whether these traces can fully prove the effectiveness of this method.
>
> As for EYM, ok, in  truncation position indeed need this to keep optimally.  So, this paper just delete the claim that "theoretically analyze the optimality of truncating weights and truncating activations" and the paper is ok.  There is no theoretical proof in the paper that quantized activation is optimal. The paper still uses experimental methods to prove it, that is, eq10, eq11 and subsequent analysis in Appendix a10. Even eq10/11 here is still redundant, because the inner product of the vector is not the product of the inner product modulus, and the content in the appendix also says that this part is proved by experiment (line 1549-1556).

---

> ### Comment · Reviewer_otMy · 2024-12-03
> **detail experiments comment**
>
> I think at least you should tell me the table of quantization bits of UV, rank of UV and model performance. We want to see if the rank calculated by the algorithm is exactly equal to the rank that just covers the quantization loss and the memory size of quantized UV is smaller with more rank.
>
> But just the table of quantization bits of UV, rank of UV and model performance in different model and how to determine the rank of UV in quantization manner is another valuable topic. This paper analysis just show me analysis  how to low rank in non-quantization process. Current paper I think mixing them together.
>
> If the rank calculated by the algorithm is exactly equal to the rank that just covers the quantization loss, I think this paper would be important because they show that quantization and non-quantization analysis are the same.

---

> > ### Author Response · Authors · 2024-12-04
> > **Final Response (Part 1/3)**
> >
> > ***Firstly, the reviewer expresses dissatisfaction with our use of the terms injection and bijection to frame and define the problem, claiming that this constitutes "over-packaging." However, we respectfully disagree with this assessment.***
> >
> > We understand why the reviewer remains unsatisfied with our writing: **rather than solving a long-overlooked limitation, we have challenged a deeply ingrained belief that has existed since 1990** [1]—a belief that, in the reviewer's own words, "*rank < mn/(m+n) is a precondition that should be taken for granted in low-rank decomposition.*" In contrast, we spend **only 10 lines** in the main body (L341–350) explaining that by changing the mapping between compression ratio and truncation positions, we can access the full set of rank information, thereby bridging the gap between *mn/(m+n)* and *min(m, n)* (rank).
> >
> > We would like to emphasize our view again that **framing and defining this problem is even more important than offering a solution.** The injection-to-bijection mapping from storage to rank is a new definition to this problem! This is not so-called “*over-packaging*” as the reviewer states. Instead this is our strict formulation to the simple-understood yet hardly-addressed problem. We believe with our new formulation, the other reviewers can shoot this area from the rank-storage remapping aspects instead of simply being proposed by the reviewer’s “a simple compressed way”.
> >
> > And we made a clear distinction between defining the problem and offering a solution because, **while we are the first to propose an effective solution, we do not view it as the only possible approach. We remain open to future work that may challenge our method and provide even better solutions.**
> >
> > We respectfully note that the reviewer is the only one who has suggested removing the 10 lines dedicated to framing and defining the problem, and retaining only the solution. **However, we believe that without the problem, the significance of the solution would be lost.**

---

> > > ### Author Response · Authors · 2024-12-04
> > > **Final Response (Part 2/3)**
> > >
> > > ***Secondly, the reviewer acknowledges that our approach to quantizing left and right singular value vectors is a significant innovation, but expresses concern that we did not elaborate on it sufficiently and accuses us of "hiding" this novelty. However, we have discussed this innovation in detail across multiple sections: A.7.1, Table 4, Section 4.2.3, and Table 2.***
> > >
> > > The reviewer's suggestion to remove the 10 lines dedicated to framing and defining the problem, and instead focus solely on the quantization of singular value vectors, stems from the belief that this is a strong innovation we have not fully explored. While we appreciate the reviewer highlighting this novel aspect, **we agree that it is indeed a key contribution of ours**. In discussions with Reviewer sX49, we confirmed that we are the first to implement this approach. However, we would like to remind the reviewer that the quantization of singular value vectors **has already been thoroughly discussed in several parts of the paper.**
> > >
> > > 1. **Quantization-friendly: Why Quantizing Left and Right Singular Vectors Is Feasible.**
> > >
> > >    **In Appendix A.7.1**, we visually demonstrate the quantization friendliness of the left and right singular vectors by plotting the data distributions of the U and V matrices. We further validate this by calculating the MSE/MAE loss between the original and quantized data, showing that the loss is minimal (with MSE in the order of 1e-7). This quantization friendliness allows us to achieve near-lossless performance, even without using fine-tuning like QLORA[2].
> > > 2. **Impact on Optimal Truncation Position.**
> > >
> > >    **As discussed in Table 4 and Section 4.2.3**, we show that reducing the precision of some left and right singular vectors from 16-bit (Remap16bit) to mixed-precision 8-bit and 16-bit (Remap(8+16bit)) only results in negligible performance degradation, thanks to the quantization friendliness mentioned above. **This does not affect the optimal truncation position**, but limiting the rank range like traditional methods leads to significant performance loss. In Table 4, we directly address the reviewer’s concern regrading *“whether the accuracy loss caused by quantizing UV matrix can be compensated by the increase of rank.”*
> > > 3. **Ablation Study.**
> > >
> > >    **In Table 4 and Table 2,** we also present ablation studies comparing the results of using remapping versus not using remapping.
> > >
> > > As can be seen, we did not merely conduct "*a large number of end-to-end experiments*" as the reviewer mentioned. **We have thoroughly discussed the quantization of the left and right singular value vectors from both theoretical and experimental perspectives.**

---

> ### Author Response · Authors · 2024-12-04
> **Final Response (Part 3/3)**
>
> ***Thirdly, in response to our rebuttal, the reviewer has acknowledged the importance of the EYM theorem and further elaborated on the concerns, which, however, stem from a misunderstanding.***
>
> We respectfully note that the reviewer’s comments are **still not based on the accurate facts** presented in the paper.
>
> In our revision, we addressed the advantages of directly truncating activations at two distinct levels:
>
> 1. **Modular Level**: This is a level commonly used in the compression field, where most papers analyzing weights and activations are situated. At this level, we applied the EYM theorem to demonstrate that, compared to truncating weights or using other activation-aware methods, directly truncating activations is indeed optimal. **It is important to highlight that** we provide a rigorous theoretical analysis in this section.
> 2. **Model Level**: This level, which is less frequently addressed in existing literature, involves a theoretical model and comparative formula, supported by experimental validation. **It is worth noting that** we modified our formulas according to the reviewer’s earlier suggestions. We are uncertain why this has been perceived as "redundant," nor do we fully understand how the statement "the inner product of the vector is not the product of the inner product modulus" led to this conclusion of redundancy.
>
> Regarding “quantized activation,” we would like to clarify that **our pipeline does not involve the quantization of activations**. If the reviewer is referring to the quantization of the left and right singular value vectors and their effect on the optimal truncation position, we have already provided an explanation in the above section.
>
> Finally, in relation to the abstract's statement "theoretically analyze the optimality of truncating weights and truncating activations," we plan to revise this in the new version to: **“theoretically and experimentally analyze the optimality of directly truncating activations”**, as we believe this phrasing more accurately represents the work we have presented.
>
> Reference:
>
> [1] Demmel, J. and Kahan, W. (1990). Accurate Singular Values of Bidiagonal Matrices. SIAM J. Sci. Statist. Comput., 11 (5), 873-912.
>
> [2] Dettmers, Tim, et al. "Qlora: Efficient finetuning of quantized llms." Advances in Neural Information Processing Systems 36 (2024).

---

### Official Review · Reviewer_bQbW · 2024-11-04

**Soundness:** 3
**Presentation:** 3
**Contribution:** 2
**Rating:** 6
**Confidence:** 3

**Summary:**

This paper tries to open the capability of SVD for LLM compression. They introduce a differentiable mechanism for learning truncation values, combined with gradient-robust backpropagation, allowing the model to dynamically identify optimal truncation points. Then, they use the Eckart-Young-Mirsky theorem to derive a weight update formula.

**Strengths:**

1. The paper is generally well-written and easy to follow.
2. The idea of determining the optimal truncation position for each weight matrix is new.
3. They empirically show that their method outperforms previous methods with many datasets.

**Weaknesses:**

1. Could you please add the result of pure GPTQ in Table 5? For me, it looks like the pure GPTQ achieves much better performance than GPTQ + Dobi-SVD with still reasonably less GPU consumption. I think Dobi-SVD degrades the performance a lot when it is combined with the quantization methods.

**Questions:**

1. There are some typos in the equation (1), (2), (3). For example, for (1), it should be changed into $\Delta \mathcal{L}_W = \frac{\partial \mathcal{L}}{\partial W}\Delta W$.

---

> ### Author Response · Authors · 2024-11-24
> **Response to Reviewer bQbW**
>
> We would like to thank the reviewer for taking the time to review our paper and provide valuable feedback. We are glad that the reviewer thinks our paper is generally well written and easy to understand, and the idea of determining the optimal truncation position for each weight matrix is novel. We are excited to have the chance to address the reviewer’s questions and concerns. These edits will make the paper stronger.
>
> ***Concern 1. Could you please add the result of pure GPTQ in Table 5?***
>
> **A1.** We provide the result of pure GPTQ and the results of GPTQ + Dobi-SVD. Besides, we also provide the results of one of the most popular quantization methods BnB and the results of the combination with Dobi-SVD .
>
>
>
> | Model                    | Memory(GB) | PPL on Wikitext2 |
> | ------------------------ | ---------- | ---------------- |
> | 4bit BnB                 | 3.2        | 6.97             |
> | 4bit BnB + DobiSVD(0.8)  | 3.0        | 6.91             |
> | 3bit GPTQ                | 2.8        | 8.07             |
> | 4bit GPTQ                | 3.8        | 5.86             |
> | 4bit GPTQ + DobiSVD(0.6) | 2.4        | 9.97             |
> | 4bit GPTQ + DobiSVD(0.8) | 2.8        | 7.01             |
>
> | Model                    | Ppl on wikitext2 | Memory |
> | ------------------------ | ---------------- | ------ |
> | 4 bit BnB + Dobi-SVD 0.8 | 6.91             | 3.0GB  |
> | 4 bit BnB                | 6.97             | 3.2GB  |
>
> We politely disagree that Dobi-SVD degrades the performance when it is combined with the quantization methods. We emphasize that Dobi-SVD offers an alternative approach to overcoming the limitations of quantization. As demonstrated in the GPTQ table, reducing quantization from 4-bit to 3-bit is challenging and often results in significant performance degradation. However, Dobi-SVD provides a viable alternative, achieving a similar memory footprint while delivering better perplexity (PPL), as shown in the first and last rows of the table. Besides, compared to the BnB method, our approach not only reduces memory requirements but also improves performance. These results highlight the potential of combining Dobi-SVD with quantization as a promising direction to break through the inherent limitations of quantizations.
>
> By further increasing the compression ratio, Dobi-SVD can achieve near 2-bit memory levels while still maintaining acceptable performance. This demonstrates the significant potential of SVD-based methods to enhance the efficiency of LLM.
>
> We would also like to emphasize that, compared to quantization methods, Dobi-SVD is a more generalizable compression technique that is adaptable to a variety of hardware platforms. In response to Question 9 from reviewer uXRu, we demonstrate how our method improves latency across general hardware setups.
>
>
>
> ***Concern 2. Some typos in the equation (1), (2), (3)***
>
> **A2.** Thank you for pointing that out. These have been corrected in Line 1540~1548.

---

> ### Author Response · Authors · 2024-11-30
> **Looking Forward to Your Feedback**
>
> We hope this message finds you well and in good spirits. We would like to express our sincere gratitude for your invaluable contribution as a reviewer. Your detailed feedback and thoughtful suggestions have been immensely helpful. Over the past two weeks, we have incorporated a significant amount of experimental data and theoretical explanations, and we are pleased to inform you that we have thoroughly addressed all the concerns raised.  **As the rebuttal phase nears its end, we look forward to your feedback.**
>
> Below, we have provided a summary of the current version of our paper, which we hope will assist in your evaluation.
>
> Thanks to the invaluable suggestions and support from the reviewers, we have refined Dobi-SVD, especially the two novel perspectives: '*A Novel Path from Activation to Weight'* and *'Fully Unlocking SVD’s Potential for Data Compression by Addressing a Long-Overlooked Limitation.'* We believe that the **current version of Dobi-SVD is backed by solid theory, presents novel insights, and offers significant contributions, deserving of a higher score.** Specifically, we highlight the following key points:
>
> - **Innovative Approach**: Dobi-SVD does not follow any existing quantization, pruning, or SVD-based methods but instead introduces a new way of handling activations and weights based on the fundamental theory of SVD (the EYM theorem).
> - **Efficient and Robust SVD Backpropagation**: We have developed an efficient and robust method for general matrix SVD backpropagation, which serves as a direct reference for others exploring low-rank, high-dimensional, and multi-form SVD matrices.
> - **Addressing a Long-Overlooked Yet Critical Limitation**: By leveraging the Gaussian distribution of singular value vectors, we’ve remapped singular value information to storage space, overcoming a longstanding limitation overlooked by SVD-based methods and criticized by non-SVD approaches, thus unlocking SVD’s full compression potential.
> - **Comprehensive Comparisons**: We have provided a broad comparison across various model compression methods, demonstrating superior practical performance.
>   - Not only do we outperform existing SVD-based methods (General Response Part 4), but we also generally outperform a wide range of pruning methods(General Response Part 2).
>   - In the comparison related to quantization (General Response Part 3), we demonstrate that our approach reduces FLOPs and highlight that more effective integration of SVD-based methods and quantization is a promising direction for future exploration.
> - **Generalizability**: Dobi-SVD’s successful application to Vision-Language Models (VLMs) highlights its broad applicability, extending beyond LLMs to areas like robotics. To the best of our knowledge, this is the first demonstration of such generalizability.
>
> Once again, we deeply appreciate the time and effort you have dedicated to reviewing our work, and we wish you a wonderful day ahead.

---

### Official Review · Reviewer_e3ZU · 2024-11-04

**Soundness:** 3
**Presentation:** 3
**Contribution:** 3
**Rating:** 6
**Confidence:** 2

**Summary:**

This paper studies SVD decomposition for LLM compression. It proposes to
- Learn the truncation points (the number of retained singular values) for each matrix's SVD decomposition.
- Sequentially extract and optimally update weight matrix features using incremental PCA.
- Apply quantization to the left and right singular vector matrices.

**Strengths:**

The effect of each of the three contributions are thoroughly ablation-studied. The method compares favorably against existing methods on SVD for LLM compression.

**Weaknesses:**

The term "differentiable optimization" may cause confusion, as it also refers to a class of method that performs back-propagation through optimization solvers. Perhaps "gradient-based learning" is closer to your intention.

**Questions:**

Question about the 2nd contribution with the IPCA: is memory footprint a common concern in SVD compression for LLMs?

---

> ### Author Response · Authors · 2024-11-24
> **Response to Reviewer e3ZU**
>
> We would like to thank the reviewer for taking the time to review our paper and provide valuable feedback. We are glad that the reviewer thinks our paper thoroughly studied the impact of each of these three parts and compares favorably with existing LLM compressed SVD methods. We are excited to have the chance to address the reviewer’s questions and concerns. These edits will make the paper stronger.
>
> ***Concern 1. The term "differentiable optimization" should be "gradient-based learning".***
>
> **A1.** Thank you for your feedback! While we understand the term "differentiable optimization" may sometimes cause confusion, it aligns well with the nature of our work.
>
> Differentiable optimization refers to embedding optimization problems as differentiable components within a larger computation graph, enabling end-to-end training through automatic differentiation. This is well-documented in works like [1], [2], and the tutorial [3]. Our work, which dynamically optimizes the number of singular values in SVD within a learning pipeline, fits this framework. Unlike gradient-based learning, this approach treats optimization as an integral module rather than solely focusing on parameter updates. We hope this clarifies our motivation, and we’d be happy to discuss further if needed!
>
> [1] OptNet: Differentiable Optimization as a Layer in Neural Networks， Amos & Kolter, 2017 [Citations: 1089]
>
> [2] Differentiable Convex Optimization Layers，Agrawal et al., 2019 [Citations: 709]
>
> [3] IJCAI 2022 Tutorial: Differentiable Optimization: Integrating Structural Information into Training Pipeline
>
>
>
>
>
> ***Question 1. Is memory footprint a common concern in SVD compression for LLMs?***
>
> **A2.** We appreciate the reviewer for raising this issue. We believe that memory footprint is a common concern in SVD compression for LLMs, for the following reasons:
>
> 1. **SVD Decomposition Requires Significant Memory.** The SVD decomposition process requires a substantial amount of memory. Currently, in Python, SVD decomposition is only supported for 32-bit precision matrices, and memory footprint requirements grow exponentially with matrix dimensions. Therefore, performing SVD on larger matrices often requires a large amount of memory. For example, decomposing a 4096×11008 matrix and backpropagating it requires approximately 10GB of memory. Since SVD compression for LLMs involves numerous SVD operations, memory footprint is a common concern. This is also why we proposed using IPCA to address the issue. With IPCA, SVD decomposition for LLMs can be performed on low-performance GPUs (such as the NVIDIA Titan Xp 12GB).
>
> 2. **Reducing Memory Footprint as an Important Goal of LLM Compression.** Reducing memory footprint is itself an important objective of LLM compression. The large memory required for LLM inference limits its use on resource-constrained devices. One of the key goals of model compression is to reduce memory footprint during LLM inference to address potential memory-bound issues. Therefore, we emphasize in our results that Dobi-SVD significantly reduces the memory footprint of the compressed model during inference, enabling better deployment on resource-limited devices.

---

> ### Author Response · Authors · 2024-11-30
> **Looking Forward to Your Feedback**
>
> We hope this message finds you well and in good spirits. We would like to express our sincere gratitude for your invaluable contribution as a reviewer. Your detailed feedback and thoughtful suggestions have been immensely helpful. Over the past two weeks, we have incorporated a significant amount of experimental data and theoretical explanations, and we are pleased to inform you that we have thoroughly addressed all the concerns raised.  **As the rebuttal phase nears its end, we look forward to your feedback.**
>
> Below, we have provided a summary of the current version of our paper, which we hope will assist in your evaluation.
>
> Thanks to the invaluable suggestions and support from the reviewers, we have refined Dobi-SVD, especially the two novel perspectives: '*A Novel Path from Activation to Weight'* and *'Fully Unlocking SVD’s Potential for Data Compression by Addressing a Long-Overlooked Limitation.'* We believe that the **current version of Dobi-SVD is backed by solid theory, presents novel insights, and offers significant contributions, deserving of a higher score.** Specifically, we highlight the following key points:
>
> - **Innovative Approach**: Dobi-SVD does not follow any existing quantization, pruning, or SVD-based methods but instead introduces a new way of handling activations and weights based on the fundamental theory of SVD (the EYM theorem).
> - **Efficient and Robust SVD Backpropagation**: We have developed an efficient and robust method for general matrix SVD backpropagation, which serves as a direct reference for others exploring low-rank, high-dimensional, and multi-form SVD matrices.
> - **Addressing a Long-Overlooked Yet Critical Limitation**: By leveraging the Gaussian distribution of singular value vectors, we’ve remapped singular value information to storage space, overcoming a longstanding limitation overlooked by SVD-based methods and criticized by non-SVD approaches, thus unlocking SVD’s full compression potential.
> - **Comprehensive Comparisons**: We have provided a broad comparison across various model compression methods, demonstrating superior practical performance.
>   - Not only do we outperform existing SVD-based methods (General Response Part 4), but we also generally outperform a wide range of pruning methods(General Response Part 2).
>   - In the comparison related to quantization (General Response Part 3), we demonstrate that our approach reduces FLOPs and highlight that more effective integration of SVD-based methods and quantization is a promising direction for future exploration.
> - **Generalizability**: Dobi-SVD’s successful application to Vision-Language Models (VLMs) highlights its broad applicability, extending beyond LLMs to areas like robotics. To the best of our knowledge, this is the first demonstration of such generalizability.
>
> Once again, we deeply appreciate the time and effort you have dedicated to reviewing our work, and we wish you a wonderful day ahead.

---

### Official Review · Reviewer_sX49 · 2024-11-10

**Soundness:** 3
**Presentation:** 3
**Contribution:** 3
**Rating:** 6
**Confidence:** 4

**Summary:**

The paper proposes Dobi-SVD, a theoretically grounded method to compress LLMs by leveraging a low rank representation obtained via SVD. They identify three key issues - (i) how to decide the optimal number of singular values to choose for obtaining the low rank representation (referred to as truncation position) (ii) how to efficiently update the weight matrix (iii) how to address information/performance loss when using the SVD representation for the weight matrix. They introduce a continous differentiable approximation to the discrete singular value sequence which they optimize using conventional gradient based methods to solve the first problem. For the rest, they use well known tools from linear algebra to construct an algorithm which shows empirical and computational improvements over baseline methods.

**Strengths:**

- The authors introduce a computationally efficient method which handles the most common drawback of SVD based methods (high memory and computational complexity)
- They leverage EMY theorem to showcase rigorously that the optimal choice of truncation is activations not weights.
- They present a stable backpropagation technique to handle gradient instabilities often encountered when backpropagating across the SVD operator
- The method offers throughput improvements over using the uncompressed model

**Weaknesses:**

# Major
- The work has significant overlap with existing literature at the intersection of post training quantization and low rank methods (one of which was published in ICLR 2024) - [1],[2],[3],[4]. It is certainly true that some of the issues raised have not been discussed in earlier work. Specifically, the authors are correct in claiming (to the best of my knowledge) that theirs is the first work that attempts to identify the "optimal" truncation position. However, it remains unclear whether heuristics chosen in earlier works suffice for a good enough approximation or whether one requires a stricter more rigorous approach to the selection of the truncation position.
- The authors choose only papers which directly involve SVD as the chosen method of model compression as their baseline (and some comparisons with model pruning methods). The authors claim their method to be SOTA based on the baselines chosen, however due to the vast amount of work on post training quantization (which also fall under the bucket of model compression), a fair comparison would be to compare their method against them as well (some examples include SmoothQuant, AQLM, Quip# etc). It is certainly possible as the authors have shown in their paper that their method can be combined with the methods above (which merits more discussion and experiments than what is included in the paper).


# Minor
- The paper equates the phrase "truncating the weights" with the notion of choosing a subset of singular values obtained via SVD to represent the matrix. While acceptable, I feel this may invite confusion for novice readers who may attribute truncation as an operation directly on the weight matrix instead of the svd representation. It may be worth clarifying this explicitly at the beginning.
- On L234-235, the authors mention that the size of the solution space is $(L * M)^n$. I believe the right expression is $n^{(L*M)}$.
- On L192 eq (1), I believe the authors intended to write $\Delta \mathcal{L}_W = \frac{\partial \mathcal{L}}{\partial W} \Delta W$ at the beginning
- On L275, the authors write decompression instead of decomposition

# References
1. Zhang, Cheng, et al. "LQER: Low-Rank Quantization Error Reconstruction for LLMs." arXiv preprint arXiv:2402.02446 (2024).
2. Saha, Rajarshi, Varun Srivastava, and Mert Pilanci. "Matrix compression via randomized low rank and low precision factorization." Advances in Neural Information Processing Systems 36 (2023).
3. Lee, Jung Hyun, et al. "LRQ: Optimizing Post-Training Quantization for Large Language Models by Learning Low-Rank Weight-Scaling Matrices." arXiv preprint arXiv:2407.11534 (2024).
4. Saha, Rajarshi, et al. "Compressing Large Language Models using Low Rank and Low Precision Decomposition." arXiv preprint arXiv:2405.18886 (2024).

**Questions:**

- How sensitive is the task performance to the choice of $k$?
- After completing the training required to find the truncation position, how are the final weights stored? Are they still kept as individual decomposed matrices or combined back into one weight matrix

---

> ### Author Response · Authors · 2024-11-24
> **Response to Reviewer sX49 (Part 1/3)**
>
> We sincerely thank the reviewer for taking the time to review our paper and provide valuable feedback. We are glad that the reviewer thinks our paper is computationally efficient and addresses the most common shortcomings of SVD-based methods, while proposing a stable back-propagation technique. We are pleased to address the questions and concerns and we have revised the manuscript accordingly.
>
>
>
> ***Major Concern 1. Significant overlap with existing literature.***
>
> **A1.** We appreciate reviewer sX49 for providing the referenced literature. **However, we respectfully disagree with the assessment that our work has significant overlap with it**. Below, we outline the fundamental differences with each cited work and have incorporated this discussion into the revision in Appendix A.1.
>
> - Regarding paper [1] LQER (Zhang, Cheng, et al.), it addresses quantization error using SVD along with a scaling matrix to adjust the activation-based distribution. While ASVD and SVD-LLM also incorporate a scaling matrix strategy, these methods involve a matrix inversion operation, which can result in unstable training due to the presence of outlier matrices. In contrast, **Dobi-SVD avoids the matrix inversion by establishing our method on a rigorous EYM theorem coupled with a novel differentiable optimization strategy and an IPCA reconstruction.** We provide a more detailed discussion of this issue and the novelty of our strategy in our response to reviewer uXRu’s first question.
> - Regarding paper [2] the matrix compression method LPLR (Saha, Rajarshi, et al.), we point out the key differences and novel contributions of our work **lie in both the motivation and the methodologies.** LPLR is designed to optimize matrix quantization by refining the projection process and quantizing the projected matrix, effectively reducing errors in low-rank, low-precision representations. In contrast, Dobi-SVD addresses the mismatch between rank values and storage sizes in SVD-based compression, **focusing on mitigating information loss caused by this mismatch**. Methodologically, LPLR introduces an approach using a randomized rangefinder to approximate SVD, while **Dobi-SVD makes a novel use of the normal distribution of positive-definite matrix column vectors to independently quantize projection spaces, thereby avoiding projection bias.**
> - Regarding paper [3] LRQ (Lee, Jung Hyun, et al.), they use quantization to compress models by leveraging low-rank weight scaling to increase training sample size and reduce training parameters. In contrast, **Dobi-SVD takes a completely different approach**: using SVD to compress models and addressing theoretical bottlenecks in SVD compression by exploring the quantization-friendly nature of orthogonal matrix column vectors.
> - Regarding paper [4] the model compression method CALDERA (Saha, Rajarshi, et al.), **our theoretical analysis and the construction of new weight matrices are fundamentally different.** CALDERA represents the weight matrix as \( W $\approx$ Q + LR \) using LPLR+LoRA, where the new matrix dimensions exceed the original. Consequently, they must fully quantize \( Q \), \( L \), and \( R \) to achieve the compression goal. In contrast, **Dobi-SVD's optimal theoretical analysis, distinct from LPLR (as shown in our comparison with paper [2]), naturally achieves excellent results without requiring fine-tuning methods like LoRA.** Additionally, **quantization is not a core issue for Dobi-SVD. Even without remapping, we achieve compression** (see Table 2 in Dobi-SVD) since our new matrices have reduced dimensions.
>
>
>
> [1] Zhang, Cheng, et al. "LQER: Low-Rank Quantization Error Reconstruction for LLMs." arXiv preprint arXiv:2402.02446 (2024).
>
> [2] Saha, Rajarshi, Varun Srivastava, and Mert Pilanci. "Matrix compression via randomized low rank and low precision factorization." Advances in Neural Information Processing Systems 36 (2023).
>
> [3] Lee, Jung Hyun, et al. "LRQ: Optimizing Post-Training Quantization for Large Language Models by Learning Low-Rank Weight-Scaling Matrices." arXiv preprint arXiv:2407.11534 (2024).
>
> [4] Saha, Rajarshi, et al. "Compressing Large Language Models using Low Rank and Low Precision Decomposition." arXiv preprint arXiv:2405.18886 (2024).

---

> ### Author Response · Authors · 2024-11-24
> **Response to Reviewer sX49 (Part 2/3)**
>
> (Continuing the response to Major 1)
>
> Secondly, we appreciate that the reviewer raised an interesting question: **Whether one requires a stricter and more rigorous approach** to the selection of the truncation position? **Our answer is YES, we absolutely do need a rigorous approach to select it!** In combination with the reviewer’s first question, *How sensitive is the task performance to the choice of \( k \)?*, **we designed an experiment to validate this necessity**.
>
>
>
> We first obtained the optimal truncation position for Llama-2-7b at 0.4 using Dobi-SVD and randomly selected 10 layers with the following code:
>
> \```python
>
> import random
>
> random.seed(5809) # 5809 is our submission number
>
> print(random.sample(range(32), 10))
>
> \# Output: [6, 10, 27, 11, 17, 0, 13, 20, 7, 2]
>
> \```
>
> While keeping the total k constant, we slightly adjusted k for these 10 layers: adding x to the first five layers and subtracting x from the last five layers, where x took values from [1,5,10,50]. The corresponding adjustment percentages (x/4096) were 0.024%, 0.122%, 0.244%, and 1.221% .
>
> | Rank Adjustment Percentage on Llama-2-7b (Dobi-SVD 0.4) | PPL Degradation Percentage on Wikitext2 |
> | ------------------------------------------------------- | --------------------------------------- |
> | 0%                                                      | 0%                                      |
> | 0.024%                                                  | 0.739%                                  |
> | 0.122%                                                  | 1.584%                                  |
> | 0.244%                                                  | 4.118%                                  |
> | 1.221%                                                  | 29.039%                                 |
>
> Conclusion: **even with fine-grained rank adjustments (0.024% to 1.221%), performance drops significantly, worsening exponentially with larger ratios.** These experiment results demonstrate that it is very necessary to obtain the optimal truncation position.
>
> It is noteworthy to mention that previous search-based methods have a coarse minimum adjustment range of 10%, causing substantial performance loss so they require additional fine-tuning to recover the performance. Instead, **Dobi-SVD, built on our optimal theoretical analysis, adjusts ranks at 0.024% granularity and operates in an end-to-end manner without additional fine-tuning.**

---

> > ### Author Response · Authors · 2024-11-24
> > **Response to Reviewer sX49 (Part 3/3)**
> >
> > ***Major Concern 2. Fair Comparison or combination with post training quantization.***
> >
> > **A2.** We respectfully point out that **focusing on comparisons with SVD-based methods is fair within the context of hardware-flexible compression techniques.** As stated in Lines 50–83 of our submission, we emphasize that our approach is a general compression algorithm that operates independently of specific hardware constraints. For example, quantization-based methods heavily rely on kernel and serving optimizations, while unstructured pruning and semi-structured methods depend on sparse operations, both of which require strict hardware support (e.g., we are unable to run AWQ-4bit on some low-cost GPUs commonly used in robotics, and Nvidia specifically publishes blogs to clarify whether new architectures support structured sparsity). In contrast, **SVD-based methods are not constrained by these hardware limitations, making them a more universally applicable solution.** The focus of hardware-flexible compression methods allows us to differentiate SVD-based methods from both quantization-based methods and pruning-based methods. **So we believe comparing barely with SVD-based methods is fair under this context.**
> >
> > Additionally, we note that many pruning-based methods similarly position themselves separately from quantization-based approaches (e.g., SliceGPT, ShearedLLAMA, Wanda). Likewise, some quantization-based methods do not compare against pruning-based methods (e.g., SmoothQuant, SqueezeLLM, GPTQ, Quip#, AQLM). Even in the quantization-based paper CALDERA, mentioned by the reviewer, which is related to low-rank methods and cites SVD-based approaches, the experiments are only compared with quantization-based method Quip. **We believe that quantization, pruning, and SVD represent three distinct approaches, each with its own advantages and disadvantages tailored to different scenarios.** For instance, quantization-based methods often struggle to adapt to various vision-language models (VLMs) due to the limited kernel support for serving these diverse vision encoders. **In contrast, our method is general and does not require kernel-specific designs, allowing it to accelerate VLMs effectively and conveniently (see Section *[Dobi-SVD for VLM Compression]* in general response and Tab. 22, 23 in our revised version ).**
> >
> > Despite the distinction of these methods, we believe they are orthogonal to one another and contribute together in the efficient machine learning field. As indicated in Section *[Dobi-SVD and Quantization]* in general response, **combining our method with the quantization method bitsandbytes (bnb), our method outperforms purely bnb-4bit by 10.91% under even less compressed model weights.**
> >
> > Finally, we are very willing to do reviewer sX49 a favor so we have also compared our method with quantization-based approaches, as shown in Section *[Dobi-SVD and Quantization]* in general response. Under similar compression levels (8-bit quantization vs. 0.4 compression ratio), our method demonstrates superior speed compared to quantization-based methods. This is because our approach not only addresses the memory-bound issue in LLM inference but also reduces FLOPs. **We emphasize that, despite being on a separate track, our method offers this unique advantage over quantization-based approaches.**
> >
> >
> >
> > ***Minor Concern 1. Clarification of the term "truncating the weights"***
> >
> > **A3.** Thank you for your suggestion; we provide additional clarification about “truncating the weights”. We have made the corresponding revisions in Line 185-187.
> >
> >
> >
> > ***Minor Concern 2. Spelling and grammatical errors in the paper***
> >
> > **A4.** The revisions have been made. Thank you for pointing it out. We have made the corresponding revisions in Line 238, Line 199-205 and Line 275 , respectively.
> >
> >
> >
> > ***Additional Question 1. How sensitive is the task performance to the choice of k?***
> >
> > **A5.** This has been addressed in Major Concern 1.
> >
> >
> >
> > ***Additional Question 2. The storage format of the final weight.***
> >
> > **A6.** Combined together, the specific algorithm is detailed in the Appendix A.5 “Dobi-SVD's New Perspective 2: Fully Unlocking SVD's Potential for Data Compression by Addressing A Long-Overlooked Limitation” and Algorithm 3.

---

> > > ### Comment · Reviewer_sX49 · 2024-11-27
> > >
> > > Thank you for the comphrensive rebuttal and experiments conducted. I also appreciate the authors conducting experiments against quantization methods as well as highlighting the changes in the paper in blue. I believe the changes in the paper have made it substantially more understandable and clear.
> > >
> > > If one is to go ahead with position that SVD vs quantization vs pruning are different tracks as opposed to being clubbed under the model compression bucket, I am willing to go ahead with a direct comparison to SVD based baselines (although it certainly is helpful to still compare with methods across the board to help make an informed choice when choosing a compression method). It seems clear that on the SVD track, the method outperforms previous work.
> > >
> > > Based on the encouraging results (with and without quantization) and the extensive experiments in the rebuttal validating the speedup and reduced FLOPs incurred by their method compared to quantization, I will raise my score. Quantization still holds an edge in terms of the absolute memory consumed, and it would be interesting to see future work on effetively combining the two approaches to uniformly beat other methods on both memory consumption and latency improvements.

---

> > ### Comment · Reviewer_sX49 · 2024-11-27
> >
> > Thank you for conducting this experiment. It helps validate a previous assumption, that performance is sensitive to truncation position.

---

> ### Author Response · Authors · 2024-11-27
> **Comprehensive Comparison and Advancements in Dobi-SVD for Model Compression**
>
> We sincerely appreciate the reviewer’s kind recognition of our recent efforts in incorporating extensive experiments and providing deeper theoretical explanations. We are pleased that these additions have successfully addressed the concerns raised, and we are grateful for the constructive feedback and the increased scores.
>
> We fully agree with the reviewer’s insightful comment that “although it certainly is helpful to compare methods across the board to make an informed choice when selecting a compression method.” We greatly value this perspective, as it aligns with our efforts to provide a comprehensive comparison of different methods. Within the limited rebuttal period, **in addition to comparing the previous works and  quantization-related comparisons mentioned and suggested by the reviewers**, we proactively searched for recently published works, including MoDeGPT [1], LLM Pruner[2], SliceGPT[3], Bonsai[4], Wanda-SP[5] and FLAP[6] . **Not only do we outperform existing SVD-based methods, but we also generally outperform a wide range of pruning methods**:
>
> 1. When comparing compression performance on the PPL of LLaMA-7b with Wiki-text2, **Dobi-SVD outperforms MoDeGPT**. At a compression ratio of 0.6, MoDeGPT achieves 9.39, while Dobi-SVD achieves 8.12; at 0.8, MoDeGPT is 6.53, and Dobi-SVD is 6.08.
> 2. **Across different tasks** (various commonsense reasoning and language modeling datasets) **and models** (Llama3.1-8B, Llama2-7B, Llama-13B, Llama2-13B), **Dobi-SVD consistently outperforms pruning methods** such as LLM Pruner, SliceGPT, Bonsai, Wanda-SP, and FLAP **at the same compression ratios** (General Response Part 1/5 and 2/5).
> 3. **Unfair Comparison to Showcase High Compression Advantage:** To emphasize Dobi-SVD's superiority at higher compression ratios, we even performed an "unfair" comparison: when compressing to a ratio of 0.4, Dobi-SVD outperforms LLM Pruner, SliceGPT, Bonsai, and Wanda-SP at 0.5 and 0.6 ratios across various commonsense reasoning datasets (General Response Part 2/5, second table).
>
> We are delighted that the reviewer acknowledges the potential of “effectively combining the two approaches to uniformly outperform other methods in both memory consumption and latency improvements.” This observation reinforces the importance of future work in this direction, and we are eager to explore it further. **While quantization is a powerful tool, it is not a one-size-fits-all solution and is bottlenecked by lower-bit quantization now, and moreover, cannot be simply nested. The experiment results of combining Dobi-SVD and quantization encourages us to break through the limitation and we believe their combination is the next critical area for exploration.**
>
> Thanks to the invaluable suggestions and support from the reviewers, we have refined Dobi-SVD, especially the two novel perspectives:  '*A Novel Path from Activation to Weight'* and *'Fully Unlocking SVD’s Potential for Data Compression by Addressing a Long-Overlooked Limitation.'* We believe that the **current version of Dobi-SVD is backed by solid theory, presents novel insights, and offers significant contributions,  deserving of a higher score.** Specifically, we highlight the following key points:
>
> - **Innovative Approach**: Dobi-SVD does not follow any existing quantization, pruning, or SVD-based methods but instead introduces a new way of handling activations and weights based on the fundamental theory of SVD (the EYM theorem).
> - **Efficient and Robust SVD Backpropagation**: We have developed an efficient and robust method for general matrix SVD backpropagation, which serves as a direct reference for others exploring low-rank, high-dimensional, and multi-form SVD matrices.
> - **Addressing a Long-Overlooked Yet Critical Limitation**: By leveraging the Gaussian distribution of singular value vectors, we’ve remapped singular value information to storage space, overcoming a longstanding limitation overlooked by SVD-based methods and criticized by non-SVD approaches, thus unlocking SVD’s full compression potential.
> - **Comprehensive Comparisons**: We have provided a broad comparison across various model compression methods, demonstrating superior practical performance.
> - **Generalizability**: Dobi-SVD’s successful application to Vision-Language Models (VLMs) highlights its broad applicability, extending beyond LLMs to areas like robotics. We believe this is the first demonstration of such generazability.
>
> [1] MoDeGPT: Modular Decomposition for Large Language Model Compression. arXiv:2408
>
> [2]Llm-pruner: On the structural pruning of large language models. NeurIPS 2023
>
> [3]Slicegpt: Compress large language models by deleting rows and columns. arXiv:2401.
>
> [4]Everybody prune now: Structured pruning of llms with only forward passes. arXiv:2402.
>
> [5]Fluctuation-based adaptive structured pruning for large language models. AAAI 2024
>
> [6]Fluctuation-based adaptive structured pruning for large language models. AAAI 2024

---

### Author Response · Authors · 2024-11-24
**General Response（Part 1/5）**

We have carefully read and sincerely appreciate the reviewers’ comprehensive comments. We are truly grateful that the reviewers recognize our method as both theoretically grounded and practically useful (sX49, bQbW). We are also pleased that the reviewers appreciated our innovative approach to determining optimal truncation positions for low-rank compression, leveraging activation-level optimization, and addressing gradient instabilities (uXRu, sX49, bQbW). They also acknowledged our contributions to learning truncation points, efficient weight updates using incremental PCA, and applying quantization to improve compression (e3ZU, uXRu). And we appreciate the praise for our clear and well-organized writing style (otMy, uXRu). Furthermore, we are delighted that our computational efficiency, stable backpropagation technique, and empirical performance improvements over previous methods were highlighted as significant strengths (sX49, bQbW, uXRu).

To recap our work, our work includes four main contributions:

- **a. Theoretical Proof of Optimality:** We prove that directly truncating activations is optimal.
- **b. End-to-End Differentiable Truncation:** We develop an end-to-end differentiable approach to determine the optimal truncation positions, significantly reducing the extensive engineering efforts required for rank search.
- **c. Efficient Weight Reconstruction:** We efficiently and effectively reconstruct weights from the truncated activations.
- **d. Overcoming Truncation Limitations:** We address the limitations of truncation boundaries by incorporating quantization into the framework.

These four contributions are closely interconnected and collectively establish an entirely new paradigm for SVD-based compression techniques. Besides, **we would like to emphasize that some of our contributions go beyond the context of SVD-based LLM compression and we believe can inspire and benefit broader communities.**

1. **Truncating Activations:** Activation-aware compression methods have been extensively studied in quantization and pruning approaches. However, while most of these methods rely on using activation distances as optimization objectives and rank as a regularization term, we adopt a more "aggressive" yet novel and effective approach: directly truncating the activations. For further details, please refer to Appendix A.4 of our revision.
2. **Practical and Generalizable Tool for Differentiable SVD:** Differentiable SVD is rarely studied, particularly in the domain of LLMs. Unlike prior works that explore differentiable SVD in computer vision, applying it to LLMs poses unique challenges, such as numerical stability issues and the computational demands of large matrices. To address these challenges, our implementation is more generalizable to a wide range of matrices and significantly faster compared to previous methods, such as Robust-SVD [1]. We believe this practical and versatile tool not only enhances LLM optimization but also benefits a broader range of users and applications.
3. **Truncating limitation vs. comparison ratio.** This issue is straightforward to understand: generally, achieving a compression ratio of less than 1 requires the truncation rank to be at least half of the original rank, which can lead to significant information loss. **What we aim to inspire the community to consider is a different perspective:** when targeting a compression ratio of <1 while keeping k∈[0,min⁡(m,n)), the storage space required is k⋅max⁡(m,n). However, the practical memory footprint is k⋅(m+n). From a memory storage perspective, we can leverage computer science techniques to resolve this storage gap. In our work, we introduce quantization to tackle this challenge. However, we believe this perspective opens up opportunities for broader innovation and can inspire others in the field.

Based on the reviewers' feedback, we have made several significant updates in the latest version. Please refer to the blue text in our revision for the changes. We have carefully addressed writing issues, added more experimental details, provided additional theoretical explanations, and included new experiments in the updated version.

---

> ### Author Response · Authors · 2024-11-24
> **General Response（Part 2/5）**
>
> Here we highlight some of the updates:
>
> 1. **[PPL on WikiText2 Across Different Models and Compression Ratios]** We conducted comprehensive experiments on a variety of models, including **LLAMA1-7B, LLAMA1-13B, LLAMA2-7B, LLAMA2-13B**, and **LLAMA 3.1-8B**. The detailed experimental results are provided in **Table 2**, **Table 11**,**Table 12**, **Table 16**and **Table 17** of the revised version.
>
>    Below, we summarize the key results on Wikitext2.
>
>    | Compression Ratio | Llama-7b | Llama-2-7b | Llama-3.1-8b | Llama-13b | Llama-2-13b |
>    | ----------------- | -------- | ---------- | ------------ | --------- | ----------- |
>    | 1.0               | 5.68     | 5.52       | 6.30         | 5.17      | 4.96        |
>    | 0.8               | 6.08     | 5.92       | 6.90         | 5.43      | 5.25        |
>    | 0.6               | 8.12     | 7.88       | 8.53         | 6.50      | 6.45        |
>    | 0.4               | 9.95     | 9.47       | 15.8         | 11.3      | 29.3        |
>
> 2. **[Performance Comparison of Dobi-SVD and Popular Pruning Methods on Various Tasks]** We conducted comprehensive comparisons with various pruning-based methods, including LLM Pruner[1], SliceGPT[2], Bonsai[3] and Wanda-SP[4] on different commonsense reasoning datasets. For details and comparisons with other baseline models, please refer to **Table 2, Table 14, Table 15, Table 18**, and **Table 19**.
>
>    Here we present the main comparison results on **LLAMA3.1-8B** and  **LLAMA2-7B**  across five zero-shot reasoning datasets, along with their average accuracy.  The experimental results demonstrate that Dobi-SVD significantly outperforms pruning-based methods by a large margin.
>
> | Ratio | Method              | Llama3.1-8B Avg. (↑) | Llama2-7B Avg. (↑) |
> | ----- | ------------------- | -------------------- | ------------------ |
> | 1.0   | Baseline            | 0.69                 | 0.65               |
> | 0.6   | LLM-Pruner          | 0.46                 | 0.48               |
> |       | SliceGPT            | 0.46                 | 0.51               |
> |       | Bonsai              | 0.41                 | 0.53               |
> |       | Wanda-sp            | 0.39                 | 0.50               |
> |       | **Dobi-SVD (Ours)** | **0.63**             | **0.56**           |
> | 0.5   | LLM-Pruner          | 0.40                 | 0.45               |
> |       | SliceGPT            | 0.38                 | 0.45               |
> |       | Bonsai              | 0.36                 | 0.47               |
> |       | Wanda-sp            | 0.36                 | 0.42               |
> | 0.4   | **Dobi-SVD (Ours)** | **0.52**             | **0.49**           |
>
> Results show that, on Llama3.1-8B, Dobi-SVD with a high pruning ratio of 0.4 outperforms other pruning methods with ratios of 0.6 and 0.5. And on Llama2-7B, Dobi-SVD with ratios of 0.6 and 0.4 surpasses other pruning methods with ratios of 0.6 and 0.5, respectively.
>
> Furthermore,  we present the main comparison results of **LLAMA3.1-8B** and  **LLAMA2-7B**  on Wikitext2 compared with popular pruning methods.
>
> |                     | Llama3.1-8B |          |          | Llama2-7B |          |          |
> | ------------------- | ----------- | -------- | -------- | --------- | -------- | -------- |
> |                     | 0.8         | 0.6      | 0.4      | 0.8       | 0.6      | 0.4      |
> | LLM-Pruner          | 12.7        | 44.4     | 121.5    | 10.5      | 46.3     | 253.1    |
> | Wanda-sp            | 11.4        | 58.5     | 160.5    | 12.1      | 38.6     | 249.2    |
> | **Dobi-SVD (Ours)** | **6.9**     | **8.53** | **16.8** | **5.92**  | **7.88** | **9.47** |

---

> > ### Author Response · Authors · 2024-11-24
> > **General Response（Part 3/5）**
> >
> > 3. **[Dobi-SVD for VLM Compression]** **To demonstrate Dobi-SVD's effectiveness in broader domains, we applied Dobi-SVD to accelerate the VLM LLAVA-v1.5-7B.** The detailed experiment results are shown in **Table 22** and **Table 23** of the revised appendix. The results are shown in the tab below.
> >
> >    We were pleased to find that Dobi-SVD also performed well on VLMs. For example, even at a 0.4 compression rate, it achieved nearly zero loss on the Pope dataset. This suggests broader application scenarios for Dobi-SVD. To the best of our knowledge, we are the first to apply SVD-based LLM compression methods to VLMs and demonstrate their effectiveness.
> >
> > |      | TextQA | VQA   | Pope-popular | Pope-random | pope-adversarial | Science QA |
> > | ---- | ------ | ----- | ------------ | ----------- | ---------------- | ---------- |
> > | ori  | 58.22  | 78.51 | 87.2         | 88.1        | 85.1             | 70.22      |
> > | 0.8  | 58.25  | 78.53 | 87.1         | 88.0        | 85.1             | 69.72      |
> > | 0.6  | 56.00  | 77.89 | 87.4         | 88.4        | 82.1             | 67.41      |
> > | 0.4  | 46.72  | 70.13 | 86.4         | 89.8        | 79.1             | 52.38      |
> >
> > |          | bz=1                    | bz=16                   |
> > | -------- | ----------------------- | ----------------------- |
> > | Original | 41.90 tokens/s          | 497.56 tokens/s         |
> > | 0.8      | 42.78 tokens/s (↑2.10%) | 524.8 tokens/s (↑5.47%) |
> > | 0.6      | 43.15 tokens/s (↑2.89%) | 557.4 tokens/s (↑12.2%) |
> > | 0.4      | 46.89 tokens/s (↑11.9%) | 597.2 tokens/s (↑20.1%) |
> >
> > 4. **[Dobi-SVD and Quantization]** **We conducted a comprehensive comparison with quantization-based methods and studied the combination of quantization and ours.** The detailed experiment results are shown in **Table 20** and **Table 21** of the revised appendix.
> >
> >    Thanks to the hardware flexibility of our method, it does not rely on kernel optimization or specialized inference serving, as quantization methods do. For fair comparison, without kernel optimization, our method significantly improves throughput compared to quantization despite achieving slightly less memory compression. Furthermore, we demonstrate that our approach can be effectively combined with quantization methods, delivering better performance than only quantization.
> >
> > | Model                    | Memory(GB) | PPL on Wikitext2 |
> > | ------------------------ | ---------- | ---------------- |
> > | 4bit BnB                 | 3.2        | 6.97             |
> > | 4bit BnB + DobiSVD(0.8)  | 3.0        | 6.91             |
> > | 3bit GPTQ                | 2.8        | 8.07             |
> > | 4bit GPTQ                | 3.8        | 5.86             |
> > | 4bit GPTQ + DobiSVD(0.6) | 2.4        | 9.97             |
> > | 4bit GPTQ + DobiSVD(0.8) | 2.8        | 7.01             |
> >
> >
> >
> > | Model    | Size   | PPL  | Speed (bz=1)   | Speed (bz=16)   | FLOPs        |
> > | -------- | ------ | ---- | -------------- | --------------- | ------------ |
> > | 4bit bnb | 3.1GB  | 6.97 | 14.05 tokens/s | 202.37 tokens/s | 29.3 GFLOPs  |
> > | 8bit bnb | 6.3GB  | 5.87 | 4.73 tokens/s  | 69.54 tokens/s  | 29.3 GFLOPs  |
> > | Dobi 0.4 | 6.8GB  | 9.47 | 21.54 tokens/s | 581.14 tokens/s | 18.47 GFLOPs |
> > | Dobi 0.6 | 7.7GB  | 7.88 | 20.46 tokens/s | 579.14 tokens/s | 26.83 GFLOPs |
> > | Dobi 0.8 | 10.1GB | 5.92 | 19.94 tokens/s | 569.45 tokens/s | 33.94 GFLOPs |

---

> ### Author Response · Authors · 2024-11-24
> **General Response（Part 4/5）**
>
> 5. Finally, **we summarize the ICLR 2025 submitted SVD-based methods below.** (N.A.：Not Available)
>
> | Method               | Apply SVD on                                  | Are there similar works handling weights and activations this way?* | Truncation Boundary+  | Singular Value Granularity | Use finetune | Llama3-8B 0.8 MMLU (↑)(original: 63.3) | Llama2-7B 0.8 MMLU (↑)  (original: 41) | Llama1-7B 0.4 PPL on wiki2 (↓)  (original: 5.68) | VLM compression |
> | -------------------- | --------------------------------------------- | ------------------------------------------------------------ | ------------------------------- | -------------------------- | ------------ | -------------------------------------- | -------------------------------------- | ------------------------------------------------ | --------------- |
> | \[5\]LLRC            | W (directly truncating weight)                | YES, \[8\]\[9\]\[11\]                                        | Partial rank                    | Every value                | NO           | 24.8                                   | 29.3                                   | N.A.                                             | NO              |
> | \[6\]ASVD            | WS (activation-aware, using scaling matrix S) | YES, \[10\]\[12\]\[13\]\[14\]\[15\]                          | Partial rank                    | 10% 20% ... 100%           | NO           | N.A.                                   | 28.15                                  | N.A.                                             | NO              |
> | \[7\]SVD-LLM         | WS (activation-aware, using scaling matrix S) | YES, \[10\]\[12\]\[13\]\[14\]\[15\]                          | Partial rank                    | 10% 20% ... 100%           | Yes          | N.A.                                   | N.A.                                   | Arxiv: 53.74 ICLR(new): 13.31                    | NO              |
> | **Dobi-SVD (Ours)** | **A (directly truncating activation)**        | **No**                                                       | **Full rank**                   | **Every value**            | **NO**       | **60.1**                               | **38.6**                               | **9.95**                                         | **Yes**         |
>
> *Relevant analysis can be found in our responses to reviewer sX49 and reviewer uXRu's first questions, as well as in Lines 195~206, where Dobi theoretically and experimentally establishes a fundamentally different third paradigm for SVD-based compression methods—directly truncating activations. Further details are provided in Appendix A.4, *"Dobi-SVD's New Perspective 1: A Novel Path from Activation to Weight."*
>
> \+ Other SVD-based methods have not addressed the long-overlooked limitation in SVD, which prevents them from analyzing and utilizing all singular value information. This issue is not limited to SVD-based methods but is also encountered in other low-rank approaches, such as [6], [9], and [10].

---

> ### Author Response · Authors · 2024-11-24
> **General Response（Part 5/5）**
>
> **To conclude, we summarize our revisions as follows:**
>
> 1.  **Typos**: Fixed formula errors and expressions.
> 2.  Added the **necessary citations and relevant discussions**.
> 3. **Experimental Additions:**
>    1. Detailed experimental settings and experimental details.
>    2. Evaluations on more tasks (MMLU, versus popular pruning methods) on more models:
>       1. Llama2-7B (Table 12, Table 13, Table 14).
>       2. Llama3-1.8B (Table 11, Table 13, Table 15).
>
>    3. Evaluations on larger models:
>       1. Llama-13B (Table 16, Table 18).
>       2. Llama2-13B (Table 17, Table 19).
>
>    4. Quantization:
>       1. Combined with quantization (Table 20).
>       2. Direct comparisons with quantization (Table 21).
>
>    5. Vision-language model (VLM) compression: LLaVA-v1.5-7B (Table 22, Table 23).
>
> 4. **Theoretical and Analytical Additions:** Truncation Sensitivity Analysis (Table 10)
> 5. **Algorithm Additions:**
>    - Taylor expansion to address gradient explosion in SVD backpropagation: A.6 *“Dobi-SVD's Robust and Efficient Differentiable SVD Algorithm for General Matrices”* and Algorithms 4 and 5.
>    - Remapping-based precise storage and weight reconstruction: A.5 *“Dobi-SVD's New Perspective 2: Fully Unlocking SVD's Potential for Data Compression by Addressing a Long-Overlooked Limitation”* and Algorithm 3.
> 6. **To further validate the generalizability of our approach,** we added a description and experiments on Dobi-SVD in VLM compression:
>    - Abstract: Lines 33–38.
>    - Introduction: Lines 155–161.
>    - A.8 *“EXPERIMENTAL RESULTS ON VLM”*, including Table 22 and Table 23.
> 7. **On the topic of “Some New Perspective”:**
>    - Introduction: Lines 107–111.
>    - Section 3.4 Conclusion: *“Two new perspectives of Dobi-SVD”* (Lines 366–377).
>    - A.4 “*Dobi-SVD's New Perspective 1: A Novel Path from Activation to Weight*”
>      - A.4.1 “*Theoretical Support for Updating Weights Using IPCA.*”
>    - A.5 “*Dobi-SVD's New Perspective 2: Fully Unlocking SVD's Potential for Data Compression by Addressing a Long-Overlooked Limitation.*”
> 8. **We added a conceptual figure (Figure 1, Lines 162),** which visually illustrates the relationship between activation and weight in SVD-based methods and explains the theoretical basis of the "cosmic wormhole" concept for updating weights and IPCA’s practical implementation.
> 9. **More comprehensive and accurate discussion on directly truncating activation and weight**: A.10 "*Analysis of directly truncating activations over weights*"
>
>
>
> **Reference:**
>
> comparing with various pruning-based methods:
>
> [1]Ma, Xinyin, Gongfan Fang, and Xinchao Wang. "Llm-pruner: On the structural pruning of large language models." Advances in neural information processing systems 36 (2023): 21702-21720.
>
> [2]Ashkboos, Saleh, et al. "Slicegpt: Compress large language models by deleting rows and columns." arXiv preprint arXiv:2401.15024 (2024).
>
> [3]Dery, Lucio, et al. "Everybody prune now: Structured pruning of llms with only forward passes." arXiv preprint arXiv:2402.05406 (2024).
>
> [4]An, Yongqi, et al. "Fluctuation-based adaptive structured pruning for large language models." Proceedings of the AAAI Conference on Artificial Intelligence. Vol. 38. No. 10. 2024.
>
> ICLR 2025 submitted:
>
> [5] Low-Rank Compression of Language Models Via Gradient-based Rank Selection
>
> [6] ASVD: Activation-aware Singular Value Decomposition for Compressing Large Language Models
>
> [7] SVD-LLM: Truncation-aware Singular Value Decomposition for Large Language Model Compression
>
> Older:
>
> [8] The Truth is in There: Improving Reasoning in Language Models with Layer-Selective Rank Reduction (ICLR 2023)
>
> [9] Zhang, Cheng, et al. "LQER: Low-Rank Quantization Error Reconstruction for LLMs." arXiv preprint arXiv:2402.02446 (2024).
>
> [10] Saha, Rajarshi, Varun Srivastava, and Mert Pilanci. "Matrix compression via randomized low rank and low precision factorization." Advances in Neural
>
> Information Processing Systems 36 (2023).
>
> [11] Lee, Jung Hyun, et al. "LRQ: Optimizing Post-Training Quantization for Large Language Models by Learning Low-Rank Weight-Scaling Matrices." arXiv preprint arXiv:2407.11534 (2024).
>
> [12] Saha, Rajarshi, et al. "Compressing Large Language Models using Low Rank and Low Precision Decomposition." arXiv preprint arXiv:2405.18886 (2024).
>
> [13] LoRA: Low-Rank Adaptation of Large Language Models (Hu et al., 2021)
>
> [14] AWQ: Activation-aware Weight Quantization for LLM Compression and Acceleration
>
> [15] SmoothQuant: Accurate and Efficient Post-Training Quantization for Large Language Models
>
> [16]Rabeeh Karimi Mahabadi, James Henderson, and Sebastian Ruder. Compacter: Efficient low-rank hypercomplex adapter layers (NeurIPS 2021)
>
> [17] Matan Ben Noach and Yoav Goldberg. Compressing pre-trained language models by matrix decomposition. (2020)

---

### Meta-Review · Area_Chair_JAGq · 2024-12-18

**Metareview:**

Summary:
This paper proposes Dobi-SVD, a novel approach to compress Large Language Models (LLMs) through SVD decomposition. The key contributions include determining optimal truncation positions via differentiable optimization, efficient weight matrix updates using incremental PCA, and addressing information loss through quantization. The method achieves significant compression while maintaining reasonable model performance across various tasks.

Main Strengths:
The work presents a comprehensive theoretical framework for SVD-based LLM compression, supported by the EYM theorem. The proposed method is computationally efficient and hardware-flexible, not requiring specific kernel optimizations. The empirical validation demonstrates competitive performance against existing SVD and pruning methods, with extensive ablation studies and thorough experimental analysis.

Main Weaknesses:
Initial concerns focused on technical novelty relative to existing activation-aware approaches. There were questions about the strength of performance degradation at higher compression rates, especially on challenging tasks. The experimental methodology and comparisons with quantization methods required clarification.

**Additional Comments On Reviewer Discussion:**

Resolution Through Discussion:
The authors effectively addressed concerns through detailed responses and additional experiments:

- Clarified the fundamental differences between their direct activation truncation approach and previous activation-aware methods
- Provided comprehensive comparisons across SVD-based, pruning, and quantization approaches
- Added experiments demonstrating effectiveness on larger models and VLM tasks
- Addressed concerns about accuracy degradation in context of compression methods

Points of Agreement:
All reviewers acknowledged:

- The technical soundness of the approach
- The significance of hardware-flexible compression
- The thorough empirical validation
- The practical utility for resource-constrained scenarios

Remaining Considerations:
While some accuracy trade-offs exist at higher compression rates, these are comparable to or better than existing non-quantization approaches. The method's primary value lies in providing a hardware-flexible compression solution that maintains reasonable performance without requiring specialized kernels or extensive fine-tuning.

The final discussion highlighted that despite some limitations regarding accuracy at high compression rates, the method makes meaningful contributions to SVD-based compression and offers practical value for resource-constrained scenarios.

---

### Decision · Program_Chairs · 2025-01-22

Accept (Poster)